

SciPost Phys. Lect. Notes 97 (2025)

# Hands-on introduction to randomized benchmarking

Ana Silva⋆ and Eliska Greplova

QuTech and Kavli Institute of Nanoscience,
Delft University of Technology, Delft, The Netherlands

⋆ a.c.oliveirasilva@tudelft.nl

## Abstract

Randomized benchmarking techniques have been an essential tool for assessing the performance of contemporary quantum devices. The goal of this tutorial is to provide a pedagogical, self-contained, introduction to randomized benchmarking. With this intention, every chapter is also supplemented with an accompanying Python notebook, illustrating the essential steps of each protocol. In addition, we also introduce more recent trends in the field that bridge shadow tomography with randomized benchmarking, namely through the gate-set shadow protocol.

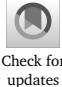

## About this tutorial

The goal of this tutorial is to provide an overview of the main principles behind randomized benchmarking techniques (often abbreviated simply as RB). While there are several comprehensive reviews on the subject (Refs. [1, 2] are two examples), a newcomer to the field still faces the challenge that a considerable amount of background knowledge is required to get familiar with the topic. Our purpose is then to ease this process by aiming at a pedagogical introduction to RB. With this intention, every chapter is supplemented with an accompanying Python notebook, illustrating the essential steps of each protocol.

A myriad of different RB variants are now available in the literature [2], most of which are not covered in this tutorial. Instead, we have chosen to focus on four RB techniques: Standard RB [3] (chapter 1), simultaneous RB [4] (chapter 2), correlated RB [5] (chapter 3), and interleaved RB [6] (chapter 4). The focus on standard RB and interleaved RB was motivated by the fact that these are protocols widely-used in practice [7,8]. Simultaneous RB and correlated RB were chosen for their relevance in the context of characterizing crosstalk errors, which is a pressing issue for multi-qubit systems.

More recently, and in parallel to the development of the RB techniques, classical shadow tomography emerged as powerful tool for characterization of quantum states and processes [9, 10]. These techniques have, in fact, a profound connection to RB. This connection has been formalized via the recently proposed gate-shadow estimation protocol [11]. The method allows for capturing multiple RB variants within the same framework, and serves as a powerful approach for learning several key aspects of a noisy gate-set. Here, we cover it in chapter 5.

Whether you are a student wanting to learn about RB from scratch, or a seasoned experimentalist who wishes to learn more about foundations behind the RB techniques these notes are for you. There are two modalities of using them:

1. You want to understand fundamentally *why* and *how* RB works: Read each section in full, then open an accompanying notebook and walk through the numerical implementation. Since this subject *is* mathematical, we included the key derivations and explanations in full aiming to make them as clear and transparent as possible.

2. You want get up to speed quickly and start using RB in your numerical or physical experiments, and you'd rather skip the math: First section of each chapter is a general description of an RB method we'll be discussing in that chapter. You can read that section and then jump straight to the accompanying notebook. After you have completed it, you can return to the text and look at the Discussion section of the chapter.

# 1 Standard randomized benchmarking

The accompanying notebooks for this section can be found at:
**Standard RB protocol example**: https://gitlab.com/QMAI/papers/rb-tutorial/-/blob/main/StandardRB/StandardRB.ipynb.
**Probing effects of finite sampling in standard RB**: https://gitlab.com/QMAI/papers/rb-tutorial/-/blob/main/StandardRB/RBandSampling.ipynb.
**Benchmarking a real device**: https://gitlab.com/QMAI/papers/rb-tutorial/-/blob/main/StandardRB/Benchmarking_a_real_device.ipynb.

The goal of standard randomized benchmarking (RB) is to quantify the average error rate of a set of quantum gates, as, for example, a Clifford gate-set [3, 12]. The protocol is a widely employed technique, with two features contributing particularly to its success: (1) it allows for constructing estimates in a sample efficient manner, (2) it hosts immunity against state preparation and measurement errors (SPAM errors) [3]. But with its simplicity, also comes several assumptions. These are hidden in the fitting model. In this section, we will give an overview of how the protocol works, as well as what are the key assumptions giving rise to its simple exponential decay model.

## 1.1 General description of standard RB and noise assumptions

When the implemented gate operations deviate from their intended set of transformations, the quantum circuit will collect errors. The goal of standard RB is to provide an estimate for the average error, occurring due to imperfect gates. RB accomplishes this goal by simulating a quantum operation that, in the noise-free case, is globally equivalent to the identity (see fig.(1)). More precisely, imagine selecting at random a set of Clifford gates and implementing them sequentially. Let's say that we choose $m$ of these gates. Then, the set of sequential

**Ideal scenario**

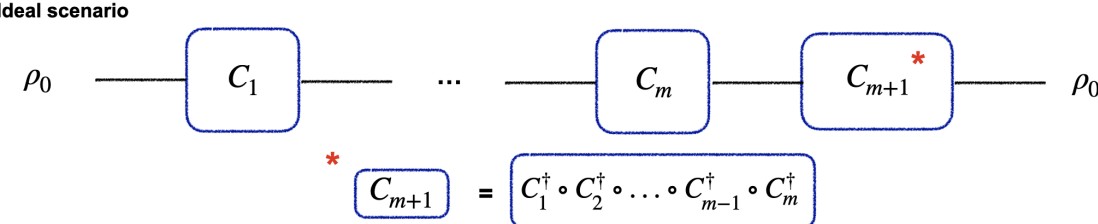

Figure 1: When the gates correspond to their ideal implementations, the RB circuit becomes equivalent to an identity operation.

transformations on the input $\rho$ can be represented as the composition of Clifford operations:

$$\rho \mapsto \left( C_m \circ C_{m-1} \circ \cdots \circ C_1 \right)(\rho). \tag{1}$$

To get back to the initial state, we need to perform an inversion gate at the $m+1$ step, i.e. implement the Clifford gate defined as follows:

$$\begin{aligned} C_{m+1} &= \left( C_m \circ C_{m-1} \circ \cdots \circ C_1 \right)^{\dagger} \\ &= C_1^{\dagger} \circ \cdots \circ C_m^{\dagger}. \end{aligned} \tag{2}$$

Since the Clifford gates form a group [13,14], the gate resulting from the previous composition is also a Clifford gate. However, rather than selected at random like the rest, it is now built specifically to restore the quantum state back to its original form.

When implementing quantum gates in practice, each operation will carry some associated error. This means that instead of the sequence in Eq.(1), one is actually applying some process resembling the following sequence of operations:

$$S_{\mathbf{i_m}} = \bigcirc_{j=1}^{m+1} \Lambda_{i_j,j} \circ C_{i_j}, \tag{3}$$

where $\Lambda_{i_j,j}$ is the error gate associated to each Clifford operation. Hence, $S_{\mathbf{i_m}}$ denotes the precise sequence of transformations on the input state. The index $\mathbf{i_m}$ is an ordered list of numbers that specify the applied Clifford gates. There are in total $m$ randomly selected independent gates,[1] drawn from the Clifford group. Consequently, for every sequence of $m$ selected gates, we can get a different ordered list $\mathbf{i_m} = (i_1, \ldots, i_m)$. An element in this list is labeled as $i_j$: It identifies the specific Clifford gate that is applied to the state, at step $j$ of the sequence. The index $j$ can then be interpreted as a time step. The indexes on the error map imply that the noise channel not only can be gate dependent ($\Lambda$ depends on $i_j$), but also time-dependent ($\Lambda$ depends on $j$). This means that Eq.(3) represents a broad class of noise models. Nevertheless, there is an assumption being made about the noise, namely that it does not depend on the content of previous gate operations. This translates to assuming that the noise is Markovian.

Despite the broad form of Eq. (3), the results in Refs. [3,12] are developed under tighter noise assumptions. Namely, it is often assumed that the errors are both time and gate independent or, equivalently, we assume $\Lambda_{i_j,j} = \Lambda$ in Eq.(3). One could think of this model as a case where the errors deviate negligibly with respect to the their mean value $\Lambda$. This condition may be relaxed to allow for weak gate dependent noise, i.e. small gate-dependent fluctuations around the mean error $\Lambda$. Although weak gate dependent errors are out of the scope of this tutorial, we refer the reader to Refs. [2,15] and references thereof.

---

[1] The last gate is completely determined by the sequence of $m$ Cliffords.

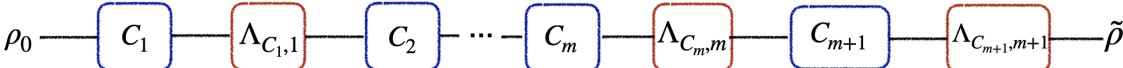

Figure 2: The noisy implementation of the gates is modeled by replacing their ideal action by the composition with an error map $\Lambda_{i_j,j}$. Here $i_j$ labels the specific gate to which the noise channel is applied to, and $j$ denotes the time instant in which it is applied. Note that for the sequential circuit depicted in the figure, the time instance coincides with the position of the operation in the circuit. The input state is denoted by $\rho_0$. The state $\tilde{\rho}$ is the output state, which will differ from $\rho_0$ due to the noisy gates.

Under the assumption that the noise is time and gate independent, the average fidelity as a function of the sequence length acquires a simple formula: $F_{\text{seq}}(m, \psi) = B_0 + A_0 \, p^m$ [3, 12]. This model can then be compared with actual experimental data and, when providing a good fit, allows for estimating the average gate error in the system. The standard RB procedure can be summarized by the following set of steps:

**Step 1:** Randomly choose $m$ Clifford gates from the Clifford group, with each Clifford having an equal probability of being selected (uniform sampling). Construct the $(m+1)$th operation as the inverse operation to the sequence (Eq.(2)).
**Step 2:** For each sequence generated in **step 1**, apply it to the input state $\rho = |\psi\rangle\langle\psi|$.

At the end of the RB sequence, the original state has been transformed by the sequence of $(m+1)$ Clifford operations. The probability of a measurement yielding the same state as the input state can then be assessed. This is expressed in the POVM[2] formalism as:

$$p_{\mathbf{i}_m}(\psi) = \text{Tr}\left(E_\psi S_{\mathbf{i}_m}(\rho)\right), \tag{4}$$

with $E_\psi$ being the POVM element associated with the desired measurement outcome. The POVM element is assumed to take into account possible measurement errors. In the ideal error-free case, $E_\psi = |\psi\rangle\langle\psi|$. Note that the probability defined above will typically depend on the particular gate sequence due to the fact that, in general, the quantum circuit will depart from the identity operation in fig.(1). Thus, in order to estimate the probability of measuring the system to be in the same state as the input state, we need to resort to the average value of $p_{\mathbf{i}_m}(\psi)$. This requires generating a sufficiently large number of random sequences, and evaluating Eq.(4) many times. In Ref. [3], $p_{\mathbf{i}_m}(\psi)$ is referred to as the survival probability.

**Step 3:** Measure the survival probability (Eq.(4)), for each sequence of Clifford gates.
**Step 4:** Repeat steps **1** to **3**, sufficiently many times, keeping the value of $m$ fixed.
**Step 5:** With the collected data, compute the average sequence fidelity, defined as follows:
$$F_{\text{seq}}(m, \psi) = \text{Tr}\left(E_\psi S_m(\rho)\right), \tag{5}$$

---

[2]For a good introduction to the POVM (Positive Operator-Valued Measure) formalism, see Ref. [16], box 2.5, page 91.

where $S_m$ is the average sequence operator,

$$S_m = \frac{1}{|\{\mathbf{i_m}\}|} \sum_{\mathbf{i_m}}^{|\{\mathbf{i_m}\}|} S_{\mathbf{i_m}}.$$ (6)

The index $\mathbf{i_m}$ is going to run over all the sampled sequences, and $|\{\mathbf{i_m}\}|$ corresponds to their total number.

**Step 6:** Repeat steps **1** to **5** for different values of the sequence length $m$.

Given the particular structure of this quantum circuit, plus the assumptions on the noise, it is possible to find an analytic expression for the average sequence fidelity, $F_{\text{seq}}(m, \psi)$. This allows to compare the outcomes of step **6** with a theoretical model, entailing the last step of the RB procedure.

**Step 7:** Fit the results from step **6** to the theoretical predicted model. When the model provides a good fit to the data, use the extracted parameters to retrieve the average error-rate in the system.

Looking ahead, determining the average sequence fidelity requires us to be able to evaluate the average sequence operator $S_m$. It turns out that, due to the simplified error model, and the presence of the inverse sequence gate, this task becomes greatly simplified. As we will see, under the assumption of gate and time independent noise, the circuit in fig.(2) is equivalent to the one in fig.(3). Since the gates are selected from a finite group (namely, the Clifford group), the average over all sequences reduces to an average over the group, and we will be able to express Eq.(6) as:

$$S_m = \Lambda \bigcirc \left( \frac{1}{|\mathcal{C}_n|} \sum_{C \in \mathcal{C}_n} C^\dagger \circ \Lambda \circ C \right)^{\circ m}.$$

The group average of the term $C^\dagger \circ \Lambda \circ C$ is called the twirling channel. This channels takes in an operator (in this case $\Lambda$), and outputs the resulting average operator, arising from the transformation $C^\dagger \circ$ (operator) $\circ C$. In the next section, we will introduce a more formal definition of the twirling channel, starting first from its definition as an average over the set of all unitary operations, and then analysing its restriction to the Clifford group. Taking this more abstract route serves the purpose of highlighting how the average sequence fidelity in RB relates to the broader concept of average gate fidelity: How *well*, on average, a channel approximates a unitary operation. Establishing a direct link between these concepts allows us to have a better grasp on how the RB sequence fidelity can relate to the average error rate over a gate set.

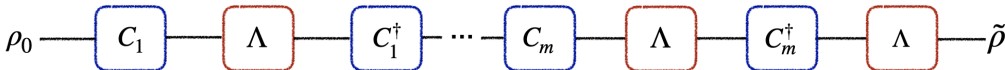

Figure 3: Under the assumption of gate and time independent errors, the RB circuit is equivalent to the one show in the figure above. Because each Clifford gate $C_i$ is followed by its inverse $C_i^\dagger$, the sequence is called "motion reversal sequence" [17].

## 1.2 Average gate fidelity, twirling and the depolarizing channel

When using standard RB, our goal is to estimate the average error-rate. To do so, we have to find a way to compute this error rate from the output of the RB circuits. This is possible because the average sequence fidelity is related to one of the basic quantum information noise models: The depolarizing channel.[3] In this section we provide some background required to establish the link between the sequence fidelity in RB and the emergence of a depolarizing channel.

The channel fidelity is a commonly used measure for gauging the similarity between two quantum channels [12], $\epsilon_1$ and $\epsilon_2$:

$$F_{\epsilon_1,\epsilon_2}(\rho) = \left( \text{Tr}\left( \sqrt{ \sqrt{\epsilon_1(\rho)} \epsilon_2(\rho) \sqrt{\epsilon_1(\rho)} } \right) \right)^2. \tag{7}$$

We assume that the two channels are linear and CPTP (completely positive and trace preserving) maps. Furthermore, let us consider the case where the channels are acting on a pure state $\rho = |\phi\rangle\langle\phi|$, and where $\epsilon_1$ is a unitary operation, while $\epsilon_2$ remains an arbitrary quantum channel,

$$\epsilon_1(\rho) = U(\rho) = \hat{U}|\phi\rangle\langle\phi|\hat{U}^\dagger, \qquad \epsilon_2(\rho) = \epsilon(|\phi\rangle\langle\phi|). \tag{8}$$

In this case, the channel fidelity acquires a much simpler form, and is named gate fidelity [12] (see appendix A for further details):

$$F_{\epsilon_1,\epsilon_2}(\rho) \to F_{U,\epsilon}(|\phi\rangle\langle\phi|) = \langle\phi|\hat{U}^\dagger\epsilon(|\phi\rangle\langle\phi|)\hat{U}|\phi\rangle. \tag{9}$$

If we define a new map $\Lambda$, such that $\Lambda = U^\dagger \circ \epsilon$, then its action on some state $\rho$ reads:

$$\Lambda(\rho) = U^\dagger(\epsilon(\rho))) = \hat{U}^\dagger\epsilon(\rho)\hat{U}, \tag{10}$$

and, therefore, we can re-express the gate fidelity in the following manner:

$$F_{U,\epsilon}(|\phi\rangle) = \langle\phi|\Lambda(|\phi\rangle\langle\phi|)|\phi\rangle = \text{Tr}\big(\Lambda(|\phi\rangle\langle\phi|)\,|\phi\rangle\langle\phi|\big) := F_{\mathcal{I},\Lambda}(\phi), \tag{11}$$

where $\mathcal{I}$ represents the trivial map, i.e. the identity operation. Note that if $\epsilon = U$, then $\epsilon(\rho) = \hat{U}\rho\hat{U}^\dagger$, and, from Eq.(10), the map $\Lambda$ becomes equivalent to $\mathcal{I}$. Hence, the gate fidelity $F_{\mathcal{I},\Lambda}$ can be thought of as a metric quantifying how much $\epsilon$ departs from a true unitary operation.

To be able to relate the gate fidelity with the RB protocol, we need to evaluate the average gate fidelity. This is because RB can only make statements regarding average quantities. The average gate fidelity can be obtained from $F_{\mathcal{I},\Lambda}$ by integrating over all possible pure states [12]:

$$\bar{F}_{\mathcal{I},\Lambda} = \int \text{Tr}\big(\Lambda(|\phi\rangle\langle\phi|)\,|\phi\rangle\langle\phi|\big) \, d|\phi\rangle. \tag{12}$$

Up until now, the quantum map $\Lambda$ has been kept fairly general, but to proceed any further, we need to specify more details regarding this operation. Let us first consider the case where $\Lambda$ is the depolarizing channel. In this scenario, the gate fidelity remains a constant for all possible pure states. As a result, also the average gate fidelity is constant:

$$\Lambda = \Lambda_{\text{dep}} = (1-p)\frac{\mathbb{1}}{d} + p\rho \implies \text{Tr}\big(\Lambda_{\text{dep}}(|\phi\rangle\langle\phi|)\,|\phi\rangle\langle\phi|\big) = \left(p + \frac{1-p}{d}\right)\underbrace{\text{Tr}(|\phi\rangle\langle\phi|)}_{=1}$$

$$\implies \bar{F}_{\mathcal{I},\Lambda} = \left(p + \frac{1-p}{d}\right)\underbrace{\int d|\phi\rangle}_{=1} = p + \frac{1-p}{d}, \tag{13}$$

---

[3]An introduction to the depolarizing channel, along with other elementary noise channels, can be found in Ref. [16], chapter 8, section 8.3.

where $d = 2^N$ is the dimension of the Hilbert space representing our system.

A second case of interest is when $\Lambda$ is replaced by the twirling channel. In the previous section, we briefly introduced the twirling channel as the average operator resulting from the operation: (group element) ∘ (operator) ∘ (inverse group element). There, the group elements were taken explicitly from the Clifford group, but the same concept applies for other groups. In particular, twirling of a quantum channel $\Lambda$ over the unitary group is defined as:

$$\Lambda_T(\rho) = \int d\mu_H(U)\, \hat{U}^\dagger \Lambda\left(\hat{U}\rho\hat{U}^\dagger\right)\hat{U}\,. \tag{14}$$

The integration generalizes the concept of average over group elements in a finite group, and $d\mu_H(U)$ denotes the Haar measure. For more information on the Haar measure see, for example, Ref. [18].

Let us now address what happens to the gate fidelity under the twirling of a quantum channel.

**Statement 1:** The average gate fidelity is invariant under twirling.

Ref. [19] gives a simple and clear proof of this statement. We replicate it in appendix B for the sake of completeness. Note that statement 1 means that learning the average gate fidelity of the channel $\Lambda$ is equivalent to learning the average gate fidelity of its twirling channel $\Lambda_T$. In fig.(3), we have illustrated the fact that, under the simplified noise assumptions, the RB circuits are equivalent to motion reversal sequences. When averaging over all these sequences, the result is the composition of $m$ twirling channels. But these twirlings are defined over the Clifford group, and statement 1 is made in the context of the larger unitary group. This brings us to the second statement:

**Statement 2:** The $n-$qubit Clifford group, $\mathcal{C}_n$, is a unitary 2-design, meaning:

$$\frac{1}{|\mathcal{C}_n|}\sum_{i=1}^{|\mathcal{C}_n|}\hat{C}_i^\dagger \Lambda\left(\hat{C}_i\rho\hat{C}_i^\dagger\right)\hat{C}_i = \int d\mu_H(U)\, \hat{U}^\dagger \Lambda\left(\hat{U}\rho\hat{U}^\dagger\right)\hat{U}\,, \tag{15}$$

where $\hat{C}_i$ is the matrix representation of an element in the Clifford group, and $|\mathcal{C}_n|$ stands for the order of the group, i.e. the total number of distinct elements in $\mathcal{C}_n$.

An explicit derivation of this result can be found, for example, in Ref. [20]. We will not attempt to prove this mathematical statement. Instead, let us provide some further motivation on the meaning of unitary designs.

To simplify notation, let us think of the integration term $\hat{U}^\dagger \Lambda\left(\hat{U}\rho\hat{U}^\dagger\right)\hat{U}$ as some polynomial function $f\left(\hat{U}, \hat{U}^\dagger\right)$. The integral we are trying to compute can be seen as assessing the average value of the function, over the space of unitaries. Suppose we would really like to compute this integral, but did not know how to solve it analytically. We could try to think of a suitable discretization that would allow us to evaluate it numerically, for example:

$$\int d\mu_H(U) f\left(\hat{U}, \hat{U}^\dagger\right) \mapsto \frac{1}{K}\sum_{i=1}^{K} f\left(\hat{U}_i, \hat{U}_i^\dagger\right)\,, \tag{16}$$

where $K$ would be the total number of unitary matrices contributing to the sum. This is the spirit of unitary t-designs [20]: How to best approximate the average value of a polynomial function, defined over the entire space of unitaries, by a finite sum, using only a sub-set of

unitaries. In fact, unitary t-designs are more ambitious than the previous description - to be a unitary t-design[4] means that the finite sum reproduces the average value of the intended function exactly [20]. Still, we need to know how to choose the correct sub-set of unitary matrices, such that this endeavor is possible. It is here that the Clifford group comes into play.

The Clifford group is a subgroup of the unitary group [13,14]. Hence, it not only provides a possible sub-set of unitaries, but also endows them with a group structure. It turns out that the Clifford group is a successful choice of sub-set for the goal of constructing a unitary 2-design. This is indeed the statement made in Eq.(15).

The last bit that is required for the RB procedure is the direct link between twirling and the depolarizing channel, namely:

> **Statement 3:** The twirling of a quantum channel is a depolarizing channel,
>
> $$\Lambda_T(\rho) = p\,\rho + \frac{(1-p)}{d}\,\mathbb{1}\,. \tag{17}$$

The proof of this statement can also be found in Ref. [19], and we have also reproduced it in appendix D for completeness.

The combined implications of statement 2 and 3 are that the twirling channels emerging from averaging the sequences in fig.(3) yield simple depolarizing channels. Since the average gate fidelity does not change under twirling (statement 1), the average fidelity over our gate-set is given by Eq.(13), and the average error rate of the set becomes simply $r = 1 - \bar{F}_{\text{dep}}$ [12].

The goal of this section was to provide more context on the origins of some of the analytical results used in the RB procedure. In the next section, it will become clear how all the three statements give rise to RB's fitting model.

## 1.3 Average sequence fidelity and the definition of the average error rate

In the absence of gate and time-independent errors, the error map simplifies to $\Lambda_{i_j,j} = \Lambda$. In this case, step 1 of the RB procedure leads to the sequence:

$$S_{\mathbf{i_m}} = \Lambda \circ C_{i_{m+1}} \circ \Lambda \circ C_{i_m} \cdots \circ \Lambda \circ C_{i_1}\,. \tag{18}$$

We want to eventually translate the operator $S_{\mathbf{i_m}}$ into something resembling a twirling map, since we know that in that case there is a clear relation with the depolarizing channel. To help in this task, a new set of gates, denoted $D_{i_j}$, are introduced. These are constructed from the Clifford operators recursively, as follows:

$$\begin{aligned} D_{i_1} &= C_{i_1}\,, \\ D_{i_{j+1}} &= C_{i_{j+1}} \circ D_{i_j}\,. \end{aligned} \tag{19}$$

Given that the Clifford gates form a group, and the composition of Cliffords yields another Clifford, the new set of gates $D_{i_j}$ are also elements of the Clifford group. Note that we are at liberty of introducing any number of compositions with the trivial map, $\mathcal{I}$, in the sequence $S_{\mathbf{i_m}}$. This map is defined as:

$$\mathcal{I} = C_{i_j} \circ C_{i_j}^{\dagger} \quad \Longrightarrow \quad \mathcal{I}(\rho) = \left( C_{i_j} \circ C_{i_j}^{\dagger} \right)(\rho) = C_{i_j} C_{i_j}^{\dagger} \rho\, C_{i_j} C_{i_j}^{\dagger} = \rho\,. \tag{20}$$

---

[4]The $t$, in unitary t-design, has to do with the degree of the polynomial, which needs to be $\leq t$. More precisely, in this context, it means that $f(.)$ is a polynomial of degree at most 2 in the matrix elements of both $\hat{U}$ and $\hat{U}^{\dagger}$.

It is also possible to write the trivial map in terms of the $D$ gates. This statement is obvious for $D_{i_1}$. For the remaining $D$ gates, it follows by induction:

$$
\begin{aligned}
D_{i_2} \circ D_{i_2}^\dagger &= C_{i_2} \circ D_{i_1} \circ D_{i_1}^\dagger \circ C_{i_2}^\dagger = C_{i_2} \circ C_{i_2}^\dagger = \mathcal{I}, \\
D_{i_3} \circ D_{i_3}^\dagger &= C_{i_3} \circ D_{i_2} \circ D_{i_2}^\dagger \circ C_{i_3}^\dagger = C_{i_3} \circ C_{i_3}^\dagger = \mathcal{I}, \\
&\vdots \\
D_{i_{k+1}} \circ D_{i_{k+1}}^\dagger &= C_{i_{k+1}} \circ D_{i_{k-1}} \circ D_{i_{k-1}}^\dagger \circ C_{i_{k+1}}^\dagger = C_{i_{k+1}} \circ C_{i_{k+1}}^\dagger = \mathcal{I}.
\end{aligned}
\tag{21}
$$

With this in mind, we can re-write the sequence as a series of composition of terms of the form $D_{i_j}^\dagger \circ \Lambda \circ D_{i_j}$:

$$
\begin{aligned}
S_{\mathbf{i_m}} &= \Lambda \circ C_{i_{m+1}} \circ \Lambda \circ C_{i_m} \cdots \circ \Lambda \circ C_{i_1} \\
&= \Lambda \circ \cdots \circ C_{i_2} \circ \underbrace{C_{i_1} \circ C_{i_1}^\dagger}_{\mathcal{I}} \circ \Lambda \circ D_{i_1} \\
&= \Lambda \circ \cdots \circ \underbrace{C_{i_2} \circ C_{i_1}}_{D_{i_2}} \circ \left( D_{i_1}^\dagger \circ \Lambda \circ D_{i_1} \right) \\
&= \Lambda \circ \cdots \circ C_{i_4} \circ \Lambda \circ C_{i_3} \circ \underbrace{D_{i_2} \circ \left( D_{i_2}^\dagger \circ \Lambda \circ D_{i_2} \right)}_{\mathcal{I}} \circ \left( D_{i_1}^\dagger \circ \Lambda \circ D_{i_1} \right) \\
&= \Lambda \circ \cdots \circ C_{i_4} \circ \Lambda \circ \underbrace{C_{i_3} \circ D_{i_2}}_{D_{i_3}} \circ (D_{i_2}^\dagger \circ \Lambda \circ D_{i_2}) \circ \left( D_{i_1}^\dagger \circ \Lambda \circ D_{i_1} \right) \\
&\quad \vdots \\
&= \Lambda \circ \left( D_{i_m}^\dagger \circ \Lambda \circ D_{i_m} \right) \circ \cdots \circ \left( D_{i_1}^\dagger \circ \Lambda \circ D_{i_1} \right).
\end{aligned}
\tag{22}
$$

Step 5 in the RB procedure requires us to compute the average sequence operator, $S_m$ (see Eq.(6)):

$$
S_m = \frac{1}{|\{\mathbf{i_m}\}|} \sum_{\mathbf{i_m}}^{|\{\mathbf{i_m}\}|} \left( \Lambda \circ \left( D_{i_m}^\dagger \circ \Lambda \circ D_{i_m} \right) \circ \cdots \circ \left( D_{i_1}^\dagger \circ \Lambda \circ D_{i_1} \right) \right).
\tag{23}
$$

This sum can be further simplified in the limit where a *large* set of sequences are generated. To be precise, ideally, the total number should be such that we cover all possible ways of generating a sequence of $m$ Clifford elements. Recalling that the Clifford group has in total $|\mathcal{C}_n|$ elements, the total amount of distinct random sequences (of length $m$) we can generate is $|\mathcal{C}_n|^m$.

$$
\frac{\mathcal{C}_n}{i_1} \frac{\mathcal{C}_n}{i_2} \cdots \frac{\mathcal{C}_n}{i_m} \implies \text{Total Number} = \underbrace{|\mathcal{C}_n| \times |\mathcal{C}_n| \times \cdots \times |\mathcal{C}_n|}_{m \text{ times}} = |\mathcal{C}_n|^m.
\tag{24}
$$

Suppose $m = 1$. Then, the total number of distinct ways of selecting a single Clifford element at random is the same as the total number of elements in the group. When we want to generate a sequence of $m > 1$ elements, at each position of the sequence we always start fresh and have the same amount of elements to choose from. Therefore, the total amount of distinct sequences becomes the total number of different ways we can generate $m$ independent sequences of

length $m = 1$, as depicted in Eq.(24). This allows us to write $S_m$ as:

$$
\begin{aligned}
S_m &= \frac{1}{|\mathcal{C}_n|^m} \sum_{\mathbf{i_m}} \left( \Lambda \circ \left( D_{i_m}^\dagger \circ \Lambda \circ D_{i_m} \right) \circ \cdots \circ \left( D_{i_1}^\dagger \circ \Lambda \circ D_{i_1} \right) \right) \\
&= \Lambda \circ \underbrace{\left( \frac{1}{|\mathcal{C}_n|} \sum_{k=1}^{|\mathcal{C}_n|} D_k^\dagger \circ \Lambda \circ D_k \right) \circ \cdots \circ \left( \frac{1}{|\mathcal{C}_n|} \sum_{k=1}^{|\mathcal{C}_n|} D_k^\dagger \circ \Lambda \circ D_k \right)}_{m \text{ times}},
\end{aligned}
\tag{25}
$$

$$
S_m = \Lambda \circ \Lambda_T \circ \cdots \circ \Lambda_T \quad \Longleftrightarrow \quad \boxed{S_m = \Lambda \circ \Lambda_T^{\circ m},}
$$

where we have set

$$
\Lambda_T = \frac{1}{|\mathcal{C}_n|} \sum_{k=1}^{|\mathcal{C}_n|} D_k^\dagger \circ \Lambda \circ D_k.
\tag{26}
$$

Recall that, in the previous section, we referred to $\Lambda_T$ as the twirling operator of a map $\Lambda$. There, we defined it with respect to its action on an arbitrary state $\rho$ (see Eq.(14)), but we could had equivalently wrote it as $\Lambda_T = \int d\mu_H(U)\, U^\dagger \circ \Lambda \circ U$, where $U$ denotes a representation of the unitary group. Since $U$ acts on a state $\rho$ as $U(\rho) = \hat{U}\rho\hat{U}^\dagger$, the two expressions are interchangeable. The same is true for the elements of the Clifford group (a subgroup of the unitary group). Note that we had previously stated that the Clifford group is a unitary 2-design (Eq.(15)). Hence, $\Lambda_T$ is indeed a twirling map. In particular, this means that the action of $\Lambda_T$ is equivalent to the depolarizing channel (see Eq.(17)). Hence, the composition of $m$ twirls is the same as the composition of $m$ depolarizing channels:

$$
\begin{aligned}
\Lambda_T(\rho) &= \Lambda_{\text{dep}}(\rho) = p\rho + (1-p)\frac{\mathbb{1}}{d}, \\
\Lambda_T^{\circ 2}(\rho) &= \Lambda_{\text{dep}}\left( \Lambda_{\text{dep}}(\rho) \right) = p^2\rho + (1-p^2)\frac{\mathbb{1}}{d}, \\
&\vdots \\
\Lambda_T^{\circ m}(\rho) &= \left( circ_{j=1}^m \Lambda_{\text{dep}} \right)(\rho) = p^m\rho + (1-p^m)\frac{\mathbb{1}}{d}.
\end{aligned}
\tag{27}
$$

The last stage in step 5 entails the computation of the average sequence fidelity, which we are now able to express as:

$$
\begin{aligned}
F_{\text{seq}}(m, \psi) &= \text{Tr}\left( E_\psi S_m(\rho) \right) \\
&= \text{Tr}\left( E_\psi \Lambda \left( \Lambda_T^{\circ m}(\rho) \right) \right) \\
&= \text{Tr}\left( E_\psi \left( p^m \Lambda(\rho) + (1-p^m)\Lambda\left(\frac{\mathbb{1}}{d}\right) \right) \right) \\
&= p^m \underbrace{\text{Tr}\left( E_\psi \Lambda\left( \rho - \frac{\mathbb{1}}{d} \right) \right)}_{=A_0} + \underbrace{\text{Tr}\left( E_\psi \Lambda\left( \frac{\mathbb{1}}{d} \right) \right)}_{=B_0},
\end{aligned}
\tag{28}
$$

$$
\boxed{F_{\text{seq}}(m, \psi) = A_0\, p^m + B_0.}
$$

This equation is at the heart of the RB protocol, and in practice, together with Eq.(17), it is all that is needed to estimate the average error rates in our gate-set. The result in Eq.(28) holds exactly in the limit where the number of generated sequences $K$ is $K = |\mathcal{C}_n|^m$. However, the Clifford group is quite *large*. For instance, for a system of just 2 qubits, the group has already 11520 elements [14]. Eq.(28) should then be interpreted as the limit where the number of

sequences $K \to \infty$, and one expects to observe a convergence to a similar behaviour trend in the data for $K$ *sufficiently* large. To distinguish between the average sequence fidelity obtained in practice and the theoretical prediction in Eq.(28), the latter is referred to as $F_g(m, \psi)$,

$$\lim_{K \to \infty} F_{\text{seq}}(m, \psi) = F_g(m, \psi) = A_0 \, p^m \, + \, B_0 \,. \tag{29}$$

But how large is large, so that $F_{\text{seq}}$ is a reasonable estimator of $F_g$? It is possible to obtain an estimate of $K$ that guarantees a desired accuracy, and is independent of $n$ (the number of qubits) [12,21,22]. In Ref. [12], Hoeffding's variance-independent inequality was used to estimate the number of sampled sequences ($K$), compatible with a given level of accuracy for $F_{\text{seq}}$. The estimated lower bound for $K$ still would require evaluating a larger number of sequences than typical RB experiments do [12,21]. Tighter bounds for $K$ are now available and provide a proof that a reduced number of sample sequences is needed for a fixed level of accuracy in $F_{\text{seq}}$ [22]. It is crucial to note that none of the aforementioned bounds depend directly on the number of qubits in the system. These results support RB as an efficient experimental protocol for characterizing the average noise level in a gate set.

Let us return to the form of $F_g$. The definition of the POVM element $E_\psi$ is set to encode possible measurement errors. State preparation errors can occur if $\rho$ deviates from the intended input state. As it is clear from Eq.(28), all terms that can be affected by state preparation or measurement errors (called SPAM errors) are defining the constants $A_0$ and $B_0$. Thus, $F_g$ is said to be independent of SPAM errors. This can be understood by the fact that, as long as $A_0 \neq 0$ and $p$ lies within the interval $p \in \,]0,1[$, the parameter $p$ can be estimated from the fitting model, regardless of the specific values taken by $A_0$ and $B_0$ [12].

We have covered almost all the crucial ingredients in the standard RB protocol. Step 6 consists of a repetition of the previous steps to obtain a functional relation between the average sequence fidelity and the sequence length $m$. The final step rests in comparing the attained functional dependency with $F_g$. From that comparison, it is possible to estimate the value of the parameter $p$. The final observation we need to make has to do with how to actually estimate the average error rate, commonly denoted by $r$ [3,12]. Since the average gate fidelity is given as a composition of $m$ twirlings, the average error rate is defined to be the error per twirling. As we have seen in the last section, the twirling yields the depolarizing channel, and the average gate fidelity is invariant under twirling. Hence, $r$ is defined to be:

$$r = 1 - \bar{F}_{\text{dep}} \underset{Eq.(13)}{=} 1 - \left( p + \frac{1-p}{d} \right) \quad \Leftrightarrow \quad \boxed{r = \frac{(1-p)(d-1)}{d} \,.} \tag{30}$$

Estimating $p$ is then all that is required in order to evaluate the average error rate. The average error rate, its relation to the average gate fidelity and the RB sequence fidelity (Eq.(28)) are key results for the RB protocol.

## 1.4 Benchmarking a real device

When benchmarking a real device, the most unbiased assumption we can make is that we do not know what the underlying error sources are. This will be true both for the errors affecting the gates, as well as for SPAM errors. As we have seen in this chapter, the reliability of the standard RB protocol depends on the gate errors being at least approximately the same for all possible gate operations in our gate-set.[5] However, even if there are good reasons to assume

---

[5]Here we have used the gate-independent noise assumption in the derivation of the RB sequence fidelity (see Eq.(28)), yet it is known that the resulting decaying profile holds true even in the presence of gate dependent errors, whenever the error rates are bounded to be small. A formal proof of this statement can be found, for example, in Ref. [2]. In this seance, as long as the device is not too noisy, the RB framework remains valid even for more general noise cases.

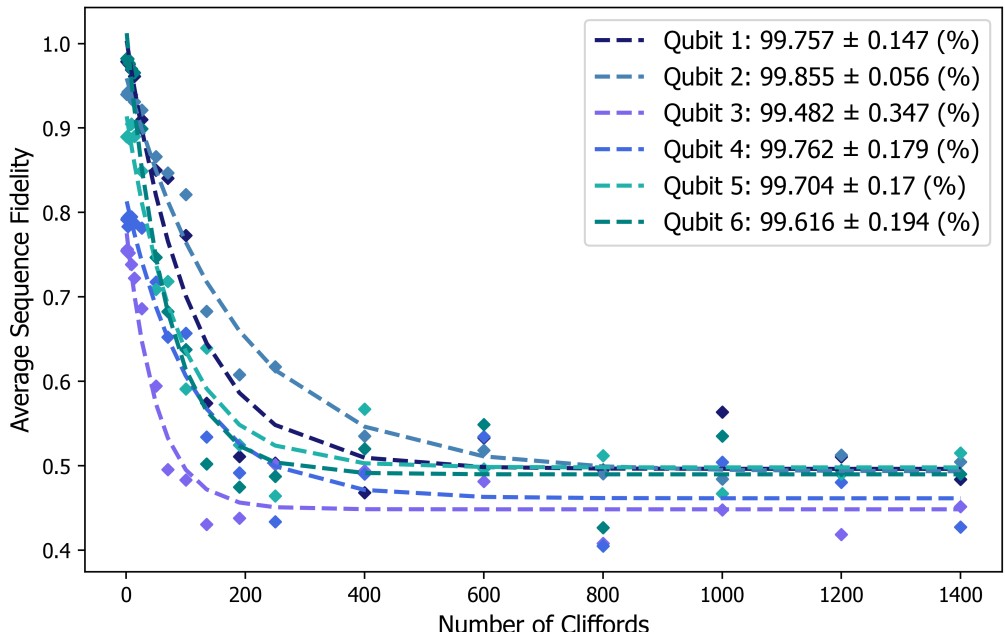

Figure 4: Average sequence fidelities as a function of the number of Clifford gates for all six individual qubits. Each decay profile corresponds to the result of the standard RB applied to benchmark the single-qubit elementary gates applied to one of the qubits in the system, as indicated in the legend. The legend also includes the corresponding single-qubit average fidelity for the operations implemented on the corresponding qubit together with the $2\sigma$ margin of error of each fidelity estimate.

this statement to hold true, there are still a wide range of different error models fulfilling this requirement. In the absence of a reliable error model, we rely solely on statistical tools to judge the quality of our fidelity estimates. In this section, we outline what the post-processing phase, i.e. the fitting procedure, would look like for real device data. The link to the accompany Python notebook can be found at the beginning of this chapter.

Spin qubits formed in semiconductor quantum dots are one of the currently available ways to realise quantum processors [23]. In particular, we will be looking at RB sequence fidelities collected from a 6-qubit linear quantum dot array in a silicon quantum well embedded in a Si/SiGe heterostructure. This device is described in [24]. The goal is to benchmark the elementary single-qubit operations, forming the building blocks for more complex single-qubit operations in the device. More concretely, these are given by single-qubit $\pi$ and $\pi/2$ rotations about the X and Y axis[6] (see table 1). Recall that for each random circuit, we measure the fidelity with respect to the input state (also known as the survival probability), and for the same circuit length we repeat the previous step multiple times (step 3 and 4 in section 1.1). We then proceed with computing the average sequence fidelity for every experimentally tested circuit length (step 5 in section 1.1). The final outcome is then a set of data points describing how the sequence fidelity decays as a function of the circuit length: Often in practice, as a function of the number of implemented random Clifford gates. For this experiment, each data point represents the average of 15 survival probabilities, each obtained from a different random circuit of fixed length. We can then employ a non-linear fitting algorithm [2] to retrieve an estimate for our desired fidelity. For a particular experiment on the 6-qubit device [24],

---

[6]These axes are the same as in the Bloch sphere: The positive $Z$ axis aligns with the qubit state $|0\rangle$, its negative orientation aligns with the qubit state $|1\rangle$, while the positive orientation of the $X$ axis aligns with the qubit state $\frac{|0\rangle+|1\rangle}{\sqrt{2}}$, etc.

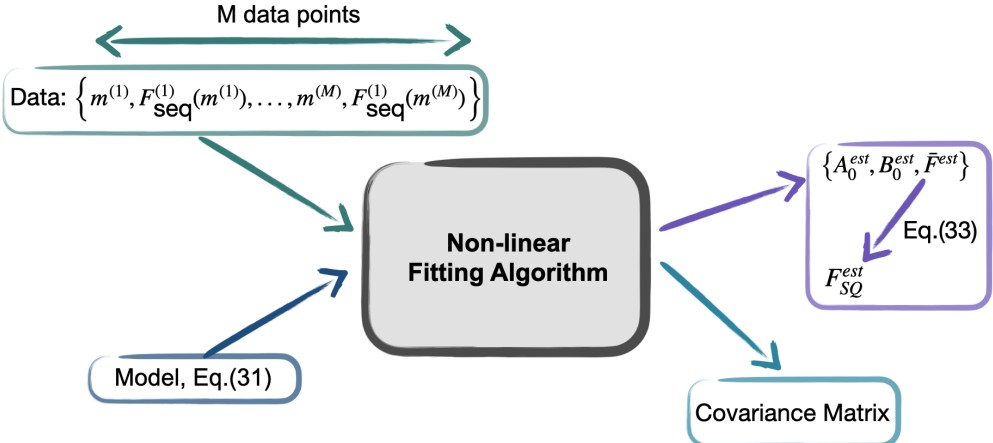

Figure 5: Schematic illustration of the RB fitting procedure. A non-linear fitting algorithm can be used to extract directly the gate-set sequence fidelity, by fitting the data to Eq.(31). Such algorithms typically provide an estimate for the covariance matrix, which can then be used to provide error bounds for the estimated value of the single-qubit fidelity, as described in the text.

performing standard RB on each individual qubit results in the RB decay profiles presented in figure 4. Note that the data scatters around the best-fit curves more widely, compared to our simulated RB data in the standard RB notebook. In that case, we considered a SPAM-free scenario and our average sequence fidelities were the outcome of averaging over more samples. Finite sampling can lead to increased uncertainty in the fidelity estimates, but also the variance of the data is affected by SPAM [22]. Note that this does not contradict the statement that RB is robust to SPAM errors, just that they play a role in the margin of error for our estimates and might justify more or less sampling. The figures of merit here are the single qubit fidelities, listed in the legend of figure 4. In Ref. [24], the reported single-qubit fidelities were between $99.77 \pm 0.04\%$ and $99.96 \pm 0.01\%$, which are superior to our estimates. Even though our data was taken from experiments on the same device, they were not measured under the same conditions as those reported in Ref. [24], which illustrates the importance of environment control for high performance quantum processors.

Let us return to the fitting procedure that renders the estimates in figure 4. From Eq.(28), the sequence fidelity is expressed in terms of the effective depolarizing parameter $p$. However, $p$ is not typically the quantity reported in RB experiments. Instead, it is customary to directly report the fidelity. As established through Eq.(13) and Eq.(17), these two quantities are directly related, so estimating one automatically leads to an estimate of the other. Alternatively, we may re-write the sequence fidelity directly with respect to the average fidelity of the gate-set, and use that instead as our fitting model:

$$F_{\text{seq}}(m) = A_0\big(2\bar{F} - 1\big)^m + B_0\,. \tag{31}$$

The set of average gate-set fidelities $\bar{F}$ are still not quite yet the fidelities shown in figure 4 (and also not those reported in Ref. [24]). Recall that the goal here is to benchmark the elementary single-qubit operations. These are given by the single-qubit rotations listed in table 1. Inspection of this table reveals that each Clifford requires at least one single-qubit rotation and a maximum of three single-qubit rotations, but the majority of single-qubit Cliffords are implemented using two single-qubit rotations. If we then explicitly calculate the average number of single qubit gates required per Clifford, $\bar{n}_{SQC}$, we arrive at an average that is indeed close to

two operations per Clifford:

$$\bar{n}_{SQC} = \frac{7}{24} + 2\,\frac{13}{24} + 3\,\frac{4}{24} = 1.875\,. \tag{32}$$

In the calculation of $\bar{n}_{SQC}$ what we are doing is a weighted average. We see from table 1 that of all possible 24 Clifford operations, 7 of them require one single-qubit rotation to implement, 13 of them require two single-qubit rotations, and the remaining 4 require 3 single-qubit rotations. Thus, the possible number of single-qubit operations required per Clifford is either 1, 2 or 3. The set $\{1, 2, 3\}$ is our set of outcomes. Each of these outcomes is then weighted by the fraction of the total number of Clifford gates corresponding to that outcome. Note that the number $\bar{n}_{SQC}$ depends on our choice of gate-set, since more complex operations will require adding new operations into our elementary set of single-qubit rotations. Under the assumption that all single qubit gates experience (at least on average) the same error rate, the average error rate over our Clifford gate-set, $r_{C_1}$, directly relates to the average error rate for the elementary single-qubit rotations:

$$r_{C_1} = \bar{n}_{SQC}\, r_{SQ} \quad \implies \quad \bar{F}_{SQ} = 1 - r_{SQ} \quad \Leftrightarrow \quad \bar{F}_{SQ} = 1 - \frac{\left(1 - \bar{F}\right)}{\bar{n}_{SQC}}\,. \tag{33}$$

Hence, the non-linear fitting procedure allows us to get an estimate for $\bar{F}$, and by appropriately re-scaling its value, we arrive at the desired fidelity $\bar{F}_{SQ}$. In figure 5 we have summarized schematically the fitting procedure.

Any of the average fidelities we can estimate is, nevertheless, just an approximate value to an unknown ground-truth fidelity. Since we do not know what this reference value is, we also cannot easily judge how *far-away* we are from it. It is therefore important to be able to estimate a plausible range of values for our fidelity estimates. Typically, non-linear fitting algorithms will also provide an estimate for the covariance matrix, where the diagonal elements represent the variance of the estimated parameters. This allows us to get an error estimate for the gate-set fidelity based on the standard deviation retrieved from the estimated covariance matrix. The error margins reported in figure 4 were obtained following this procedure, and represent $2\sigma$ standard errors. For single exponential decaying models, this approach will generally be good enough, but we note that there exists more robust approaches for estimating confidence intervals, such as the method implemented in the Python library LMFIT.

## 1.5 Discussion

The standard RB procedure employs a quantum circuit that, in the ideal error free case, just provides an identity operation. For this reason, an output not coinciding with the input state signals the imperfect implementations of the intended gates. This mismatch can be inferred from the fidelity, that will tend towards zero the more the output is *dissimilar* to the original input state. The relation between the output of this procedure and the average error rate $r$ is rooted on the mathematical properties of twirling maps, and in particular on their relation with the depolarizing channel. The emergence of a twirling map depends on several aspects. On one hand, it is linked to the way the circuit is built: Applying a set of gates and then undoing them, by tacking the inverse operation, leads to a quantum map that can be directly related to twirling (Eq.(26)), whenever we select the gates uniformly from a set that forms a group. However, this direct relation also relies on the assumptions made on the noise model, namely that it is time and gate-independent, and Markovian. This is, of course, a simplified noise model. We may then wonder if, and how, the validity of the fitting model survives beyond this family of noise models. In general, this remains an open question, but further progress has been made in the case of gate-dependent errors [2,26,27]. When only the gate-independence

Table 1: Decomposition of the single-qubit Clifford group elements as single-qubit rotations (table from Ref. [25]). The notation is as follows: $I$ represents the idle operation, $X$ denotes a $\pi$ rotation around the X axis, i.e. the operation $\exp\left(-i\pi/2\hat{\sigma}_x\right)$, $Y$ denotes a $\pi$ rotation around the Y axis, i.e. the operation $\exp\left(-i\pi/2\hat{\sigma}_y\right)$, $X/2$ is a $\pi/2$ rotation around the X axis, and so on. Note that $\hat{\sigma}_x$ and $\hat{\sigma}_y$ denote the standard Pauli matrices. In practice, these rotations are implemented by $\pi$ ($\pi/2$) pulses around the corresponding axis.

| Clifford Gate | Single-qubit rotations |
|:---:|:---:|
| 1 | $I$ |
| 2 | $Y/2$ & $X/2$ |
| 3 | $-X/2$ & $-Y/2$ |
| 4 | X |
| 5 | -Y/2 & -X/2 |
| 6 | X/2 & -Y/2 |
| 7 | Y |
| 8 | -Y/2 & X/2 |
| 9 | X/2 & Y/2 |
| 10 | X & Y |
| 11 | Y/2 & -X/2 |
| 12 | -X/2 & Y/2 |
| 13 | Y/2 & X |
| 14 | -X/2 |
| 15 | X/2 & -Y/2 & -X/2 |
| 16 | -Y/2 |
| 17 | X/2 |
| 18 | X/2 & Y/2 & X/2 |
| 19 | -Y/2 & X |
| 20 | X/2 & Y |
| 21 | X/2 & -Y/2 & X/2 |
| 22 | Y/2 |
| 23 | -X/2 & Y |
| 24 | X/2 & Y/2 & -X/2 |

restriction is lifted,[7] and the sampling of the gates remains uniform, it is possible to establish that the RB fitting model remains a linear combination of exponentials,[8] and only deviates from it by a small error [2]. This result relies on the condition that the gates implementing the elements of the group are, on average, close to a representation of the ideal gates [2], and so loosely speaking, is valid for weakly gate-dependent errors. While this statement supports the exponential decay profile commonly observed in RB experiments, the direct correspondence between the RB decay rates and the average gate fidelity presented here becomes much more subtle, when gate-dependent errors are present. In particular, the case of gate-dependent errors highlights the gauge dependence of the average gate fidelity. Consequently, this quantity depends on the choice of representation for the gates [15], while the decaying parameters estimated from the RB fitting procedure remain insensitive to the gauge choice. We refer the reader to Refs.( [2, 15, 26, 27]) for a thorough discussion on this issue.

---

[7]It still assumes the noise to be Markovian and time independent.

[8]In this sense, Eq.(28) is a particular realization of a more general RB fitting model of the form $\sum_\lambda \mathrm{Tr}(\hat{A}_\lambda \hat{M}_\lambda^m)$, where the sum is over the irreducible representations of the group, to which our gate set belongs to. See, for example, Ref.( [2]) for a more thorough discussion on this topic.

In addition to questions regarding the assumptions made on the noise model, the group from which the gates are sampled from also plays a prominent role on the type of fitting model that is associated with the RB procedure. Here the fact that the gates were assumed to be in the $n-$qubit Clifford group allowed for a very simple model of the form $A_0 p^m + B_0$. In the next section, we will see that choosing gates from the direct product group between local Clifford groups[9] already yields a model that is a linear combination with more decaying parameters. This example will also illustrate more clearly how the irreducible representations of the group influence the exact type of exponentially decaying profile that is associated to the RB procedure. One question that may then arise is what happens when we wish to benchmark gates outside of the Clifford group. A prototypical example would be the desire to benchmark T gates, since the gate set formed by Clifford+T gate is a commonly used gate-set for achieving universal quantum computing. Benchmarking T-gates requires extending the RB procedure to other finite groups, beyond the $n-$qubit Clifford group, or designing appropriate RB interleaved schemes that can still retain a group structure [28–30].

## 1.6 Key results in chapter 1

Here we summarize the main takeaways for the standard RB protocol.

- The Standard RB protocol provides an estimate of the average error-rate over the gate-set, $r$, and the average gate fidelity, $\bar{F}$. These two quantities are directly related to each other:
$$r = 1 - \bar{F}.$$
The RB estimates are robust to SPAM errors.

- The standard RB protocol assumes that the noise is Markovian, time-independent and weakly gate-dependent. This will also apply to the other variants of RB found in these lecture notes.

- The gates should be chosen from a set of gates that form a unitary 2-design. The Clifford group is the prototypical example.

- Averaging over the RB random gate sequences results in an effective depolarizing channel. For this reason, the sequence fidelity acquires the form:
$$F_{\text{seq}}(m) = A_0 \, p^m \, + \, B_0,$$
where $p$ is the effective depolarizing parameter, and the constants $A_0$ and $B_0$ absorb any dependence on SPAM errors. The effective depolarizing parameter, $p$, is related to the average gate fidelity, $\bar{F}$, allowing this quantity to be estimated directly:
$$\bar{F} = p + \frac{1-p}{d}, \quad \text{with} \quad d = 2^n.$$

- Estimation of the effective depolarization parameter also provides an estimate of the average error rate over the gate set:
$$r = \frac{(1-p)(d-1)}{d}.$$

---

[9]For example, $\mathcal{C}_1 \times \mathcal{C}_1$, where $\mathcal{C}_1$ is the single qubit Clifford group.

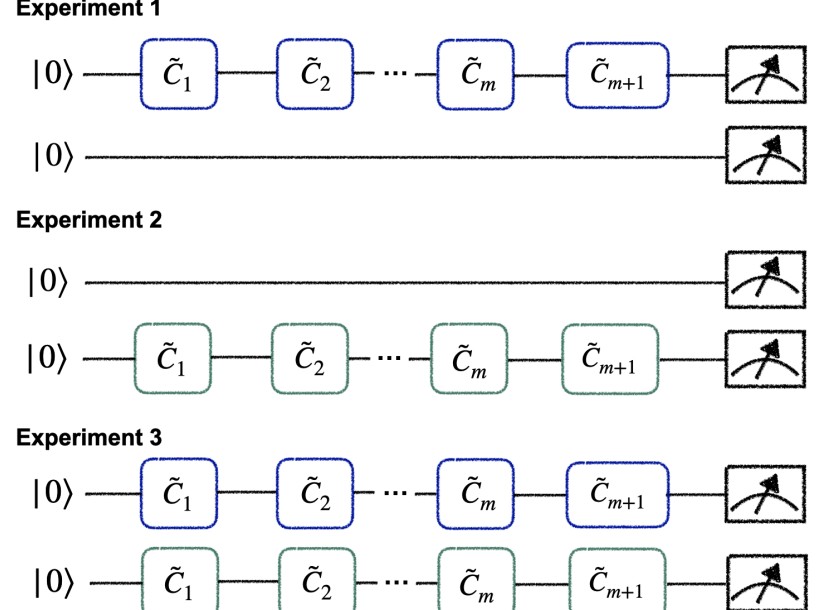

Figure 6: The simultaneous RB protocol applied to two qubits entails three experiments: Single-qubit RB only on qubit 1 (experiment 1), single-qubit RB only on qubit 2 (experiment 2), single-qubit RB on both qubits simultaneously (experiment 3). Comparing the outcomes of these three experiments allows us to extract information on the errors induced by crosstalk [4].

## 2 Simultaneous randomized benchmarking

The accompanying notebook for this section can be found at: https://gitlab.com/QMAI/papers/rb-tutorial/-/blob/main/Simultaneous%20RB/SimRB.ipynb.

In standard RB the goal was to quantify the average error rate. This, nevertheless, does not yield any information on the degree to which crosstalk errors are contributing to it. When a quantum operation is devised, it is often intended to act only on a selected subset of qubits. Furthermore, this operation would ideally be carried out in the exact same manner, regardless of the specific surroundings of the target qubits. However, when interactions between the different qubits remain, the notion of performing a set of local and independent operations on a specific subset of qubits gets spoiled, and leads to the presence of non-local and correlated errors. The ability to efficiently perform quantum operations is then crucially dependent on our capacity to identify and correct crosstalk errors. This imperative is a core motivation for the simultaneous RB method, introduced in Ref. [4]. In this section, we will see how adaptations of standard RB can be made to furnishes us with a more targeted metric for correlated errors.

### 2.1 General description of simultaneous RB

Simultaneous RB requires looking at the system as a collection of subsystems, and addressing how the operations performed in one sector impact the others, and vice-versa. Let us then be more precise with the notion of subsystem. Given a system of $n$ qubits, we can always think of it as the union of distinct parts, each being a unique subset of qubits. For example, if we have a 3—qubit system, we could imagine partitioning it into three subsystems, each of them contain-

ing one of the qubits. A generic $n-$qubit system can be partitioned in many different ways. In simultaneous RB, there is, in theory, quite some freedom in how the smaller subsets of qubits are chosen. This allows the partition to be tailored to the specific needs. In [4], the system is viewed as a bipartite system. The idea is to apply the standard RB protocol to each individual subsystem, and compare it to the case where RB is performed simultaneously on both sectors. This leads to a protocol requiring the execution of the following three experiments:

> **Experiment 1:** Perform standard RB only on the first subsystem, leaving the other un-perturbed. The set of Clifford gates required for this experiment are of the form $\mathcal{C}_1 \times \mathcal{I}$:[a] randomly chosen Clifford gates act solely on the qubits of the first subsystem, while the second subsystem is left untouched. This step is meant to estimate the depolarizing parameter and the error rate intrinsic to subsystem 1.
> **Experiment 2:** Repeat experiment 1, but now for subsystem 2. The chosen set of Clifford gates are then of the form $\mathcal{I} \times \mathcal{C}_1$. The depolarizing parameter and error rate are estimated for subsystem 2.
> **Experiment 3:** Perform standard RB on both subsystems, simultaneously. This now means the chosen set of Clifford gates are of the form $\mathcal{C}_1 \times \mathcal{C}_1$. As we will see, this leads to a fidelity that is a linear superposition of different depolarizing parameters. These can be used to define a measure quantifying the amount of correlations between the two subsystems.
>
> ______________________________
>
> [a]$\mathcal{I}$ represents the trivial group, containing only the identity element.

Additionally, just comparing the behaviour of the survival probabilities, as a function of the sequence length,[10] may be instructive for inferring the presence of crosstalk. This is because, in the absence of crosstalk, the survival probability measured in experiment 3, for example for subsystem 2, should be equivalent to the one measured in experiment 2 [4].

In the following sections, we will delve into the mathematical details that yield the various depolarizing parameters, and on how they can be used to quantify the degree of correlations in the system. We note that all results in Ref. [4] are obtained assuming a time and gate inde-pendent error model, and the same assumption will be made here. At the end of this section, we introduce a simple example meant to illustrate how simultaneous RB aids in detecting the presence crosstalk errors. The numerical implementation of this example can be found in the Python notebook introduced at the beginning of this chapter.

## 2.2 Deriving the sequence fidelity

Simultaneous RB builds on top of the standard RB method. As such, we can import many of the key results of the standard method into simultaneous RB. The biggest difference between the two protocols is in the groups from which each experiment samples the gates from. While in standard RB the gates would be chosen from the $n-$qubit Clifford group $\mathcal{C}_n$, in experiment 3, for example, they are sampled from the direct product group $\mathcal{C}_{n_1} \times \mathcal{C}_{n_2}$. The group $\mathcal{C}_{n_i}$ refers to the $n_i-$Clifford group, where $n_i$ is the total number of qubits in the ith subsystem. Except for the group sampling, the remaining steps follow exactly the standard RB procedure. For

______________________________

[10]Recall that the survival probability corresponds to the probability that the output state is the same as the input state, and that the sequence length is the number of randomly selected Clifford gates executed during the RB protocol.

this reason, we know what the average sequence operator should look like (Eq.(25)):[11]

$$S_m = \Lambda \circ \Lambda_T^{\circ m}, \tag{34}$$

with $\Lambda$ the average error map and $\Lambda_T$ the twirling operator,

$$\Lambda_T = \frac{1}{|\mathcal{G}|} \sum_{C_j \in \mathcal{G}} C_j^\dagger \circ \Lambda \circ C_j. \tag{35}$$

As it will become clearer in this section, the fact that we are now twirling over a different group will alter the type of channel produced by the twirling operator. Recall that, in standard RB, $\Lambda_T$ simply yield the familiar depolarizing channel. This relation will now be slightly different.

The ability to carry out the sum in Eq.(35) benefits from the knowledge of some key results in representation theory of finite groups. Given that we are only considering objects that are linear maps, it is not surprising that suitable representations can be chosen to be matrices. Hence, it will be instructive to express the sequence operator in matrix form. This sets the stage as to why now the notation changes to the Liouville representation of superoperators.[12] An introduction to the Liouville representation can be found in appendix E. There, we also re-express the twirling channel in this new notation, and make use of some key results in appendix C to further simplify it. We will make use of these results to gather further insight on how simultaneous RB works.

## 2.3 Theoretical predictions for the sequence fidelity

### 2.3.1 Experiment 3

During experiment 3, both subsystems are simultaneously assessed via RB. This implies twirling over the tensor product group $\mathcal{C}_{n_1} \times \mathcal{C}_{n_2}$. As discussed previously, the average sequence operator is given as a composition of the error map with $m$ compositions of the twirling channel. In appendix E.0.1, the Pauli transfer matrix associated to the twirling superoperator is shown to decompose as a sum of projectors:

$$\boxed{\hat{\mathcal{R}}_{\Lambda_T} = \sum_j \frac{\mathrm{Tr}\left(\hat{\mathcal{R}}_\Lambda \mathbb{P}_j\right)}{\mathrm{Tr}\left(\mathbb{P}_j\right)} \mathbb{P}_j.} \tag{36}$$

These projectors map onto the invariant subspaces that define the irreducible representations of the group. This require us to address what are the invariant subspaces in this context. Once those are specified, we have all the information needed to explicitly write down the form of the twirling superoperator, construct the average sequence operator as a superoperator, and finally obtain the corresponding fidelity.

Suppose we were twirling over the $n-$qubit Clifford group, $\mathcal{C}_n$. Given our normalized Pauli basis for $\mathcal{H}_{d^2}$, and recalling that the action of the Clifford group is simply a (signed) permutation of the elements in the Pauli group (see Eq.(E.11)), it follows that the action of $\mathcal{C}_n$ over our basis vectors fragments $\mathcal{H}_{d^2}$ into two distinct subspaces:

(i) Subspace $W_0$, spanned by the set of vectors $\{|\sigma_0\rangle\rangle\} \implies 1-$dimensional vector space.

(ii) Subspace $W_1$, spanned by the set of vectors $\{|\boldsymbol{\sigma}\rangle\rangle\} \implies (d^2 - 1)-$dimensional vector space.

---

[11]In Eq.(35), $\mathcal{G}$ denotes the group from which the gates are selected from. In the case of performing simultaneous RB in a 2-qubit system, $\mathcal{G}$ can either be $\mathcal{C}_1 \times \mathcal{I}$ (experiment 1), $\mathcal{I} \times \mathcal{C}_1$ (experiment 2) or $\mathcal{C}_1 \times \mathcal{C}_1$ (experiment 3).

[12]This was not a necessity in standard RB, since the relation between the twirling map and the depolarizing channel followed directly from the result that the $n-$qubit Clifford group was a unitary 2-design. Consequently, there was never a need to explicitly perform the sum in Eq.(35).

This can be understood as follows. The identity always gets mapped onto itself by the action of the Clifford group, and consequently also its vectorized version $|\sigma_0\rangle\rangle$. No other element in the Pauli group (that is not a multiple of the identity) gets mapped to the identity under the action of $\mathcal{C}_n$. It follows then that $W_0$ is an invariant subspace, since the action of $\hat{\mathcal{R}}_{\mathcal{C}}W_0 = W_0$. It is also easy to see that, in this case, $W_0$ does not contain any further non-trivial subspaces of its own. Hence, the subrepresentation of $\hat{\mathcal{R}}_{\mathcal{C}}$ on $W_0$ yields an irreducible representation. For $W_1$, we start by clarifying the meaning of the notation $\{|\boldsymbol{\sigma}\rangle\rangle\}$: This refers to the set of all the vectors that can be generated from $\frac{1}{\sqrt{d}}\{\mathbb{1}, \hat{X}, \hat{Y}, \hat{Z}\}^{\otimes n} \setminus \{\hat{\sigma}_0\}$ through vectorization. This leads to a vector subspace spanned by $(d^2 - 1)$ vectors. The action of the Clifford group on Pauli elements, other than the identity, is to re-shuffle them (apart from the addition of phase factors). Therefore, the vectors in $W_1$ always get sent back to some other vector in the same subspace. It follows then that also $W_1$ is an invariant subspace, and the subrepresentation of $\hat{\mathcal{R}}_{\mathcal{C}}$ in this subspace provides the other irreducible representation. Hence, there are two irreducible representations: A one-dimensional representation, which we denote by $\hat{\mathcal{R}}_{\mathcal{C}}^{(0)} = 1$, and a $(d^2 - 1)$ matrix representation, which we label by $\hat{\mathcal{R}}_{\mathcal{C}}^{(1)}$ [4].

For the direct product group $\mathcal{C}_{n_1} \times \mathcal{C}_{n_2}$ (with $n_1 + n_2 = n$), we can construct representations for the corresponding Pauli transfer matrices by taking the tensor products between the irreducible representations of $\mathcal{C}_{n_1}$ and $\mathcal{C}_{n_2}$. As in Ref. [4], we will limit ourselves to the specific case of a two-qubit system, and so $\mathcal{C}_{n_1} = \mathcal{C}_{n_2} = \mathcal{C}_1$, with $\mathcal{C}_1$ the one-qubit Clifford group. Since the irreducible representations of $\mathcal{C}_1$ are $\{\hat{\mathcal{R}}_{\mathcal{C}_1}^{(0)}, \hat{\mathcal{R}}_{\mathcal{C}_1}^{(1)}\}$, we can construct the set of 4 representations from it: $\{\hat{\mathcal{R}}_{\mathcal{C}_1}^{(0)} \otimes \hat{\mathcal{R}}_{\mathcal{C}_1}^{(0)}, \hat{\mathcal{R}}_{\mathcal{C}_1}^{(0)} \otimes \hat{\mathcal{R}}_{\mathcal{C}_1}^{(1)}, \hat{\mathcal{R}}_{\mathcal{C}_1}^{(1)} \otimes \hat{\mathcal{R}}_{\mathcal{C}_1}^{(0)}, \hat{\mathcal{R}}_{\mathcal{C}_1}^{(1)} \otimes \hat{\mathcal{R}}_{\mathcal{C}_1}^{(1)}\}$. These new found representations turn out to be the irreducible representations of the problem. This statement can be motivated along the same lines as before. Because only the identity gets mapped onto itself by all elements of the Clifford group, the vector space spanned by the single vector $\{|\sigma_0 \otimes \sigma_0\rangle\rangle\}$ is an invariant subspace. Let us point out that, in the superoperator language, the tensor product of linear maps can be written in the standard way (Eq.(E.6)) with a corresponding Pauli transfer matrix given by the tensor product of the individual $\hat{\mathcal{R}}$ matrices. Thus, the representation $\hat{\mathcal{R}}_{\mathcal{C}_1}^{(0)} \otimes \hat{\mathcal{R}}_{\mathcal{C}_1}^{(0)}$ is an irreducible representation (also equivalent to 1). Next, we consider the representation $\hat{\mathcal{R}}_{\mathcal{C}_1}^{(0)} \otimes \hat{\mathcal{R}}_{\mathcal{C}_1}^{(1)}$. Recall that the tensor product between two arbitrary matrices, $\hat{A}$ and $\hat{B}$, with corresponding dimension $n \times m$ and $p \times q$, yields a new matrix with dimension $np \times mq$. Therefore, $\hat{\mathcal{R}}_{\mathcal{C}_1}^{(0)} \otimes \hat{\mathcal{R}}_{\mathcal{C}_1}^{(1)}$ is a matrix representation with dimensions $1 * (d_1^2 - 1) \times 1 * (d_1^2 - 1) = 3 \times 3$ (with $d_1 = 2$). Note that an element of the group $\mathcal{C}_1 \times \mathcal{C}_1$ maps a Pauli element of the form $\sigma_i \otimes \sigma_j$ to $(\hat{C}\hat{\sigma}_i\hat{C}^\dagger) \otimes (\hat{C}\hat{\sigma}_j\hat{C}^\dagger)$. This just follows from the property discussed in Eq.(C.15). Let us then consider the subspace spanned by the set of vectors $\{|\sigma_0 \otimes \boldsymbol{\sigma}\rangle\rangle\}$, which we call $W_1$. It is clear that the action of the group $\mathcal{C}_1 \times \mathcal{C}_1$ should map it onto itself, because $\sigma_0 \to \sigma_0$ and the set $\boldsymbol{\sigma} \to \boldsymbol{\sigma}$ ($\mathcal{C}_1$ can only do sign permutations between the elements in $\boldsymbol{\sigma}$). Therefore, $W_1$ is an invariant subspace, of dimensionality $(d_1 - 1) = 3$. A matrix representation that is only defined in $W_1$ is then an irreducible representation, and will necessarily be a $3 \times 3$ matrix. Thus, we arrive at the irreducible representation $\hat{\mathcal{R}}_{\mathcal{C}_1}^{(0)} \otimes \hat{\mathcal{R}}_{\mathcal{C}_1}^{(1)}$. Similar arguments hold for the invariant subspace $W_2 = \text{Span}(|\boldsymbol{\sigma} \otimes \sigma_0\rangle\rangle)$, but now leading to the $3 \times 3$ irreducible representation $\hat{\mathcal{R}}_{\mathcal{C}_1}^{(1)} \otimes \hat{\mathcal{R}}_{\mathcal{C}_1}^{(0)}$. Finally, the subspace $W_3 = \text{Span}(|\boldsymbol{\sigma} \otimes \boldsymbol{\sigma}\rangle\rangle)$ is also mapped onto itself by $\mathcal{C}_1 \times \mathcal{C}_1$. This invariant subspace has now dimensionality 9, and so its corresponding irreducible representation will be a $9 \times 9$ matrix, respecting the underlying group structure: $\hat{\mathcal{R}}_{\mathcal{C}_1}^{(1)} \otimes \hat{\mathcal{R}}_{\mathcal{C}_1}^{(1)}$.

Having identified the irreducible representations, we have all the information required to

explicitly determine $\hat{\mathcal{R}}_{\Lambda_T}$. By Eq.(36), we can readily write:

$$\hat{\mathcal{R}}_{\Lambda_T} = \sum_{j=0}^{3} \frac{\text{Tr}(\hat{\mathcal{R}}_\Lambda \mathbb{P}_j)}{\text{Tr}(\mathbb{P}_j)} \mathbb{P}_j = \mathbb{P}_0 + \alpha_1 \mathbb{P}_1 + \alpha_2 \mathbb{P}_2 + \alpha_3 \mathbb{P}_3,\tag{37}$$

with

$$\begin{aligned}
1 &= \text{Tr}(\hat{\mathcal{R}}_\Lambda \mathbb{P}_0), &&\text{and} && \mathbb{P} = |\sigma_0 \otimes \sigma_0\rangle\rangle\langle\langle\sigma_0 \otimes \sigma_0|,\\
\alpha_1 &= \frac{\text{Tr}(\hat{\mathcal{R}}_\Lambda \mathbb{P}_1)}{3}, &&\text{and} && \mathbb{P}_1 = \sum_{\sigma_j=\{\sigma_x,\sigma_y,\sigma_z\}} |\sigma_0 \otimes \sigma_j\rangle\rangle\langle\langle\sigma_0 \otimes \sigma_j|,\\
\alpha_2 &= \frac{\text{Tr}(\hat{\mathcal{R}}_\Lambda \mathbb{P}_2)}{3}, &&\text{and} && \mathbb{P}_2 = \sum_{\sigma_j=\{\sigma_x,\sigma_y,\sigma_z\}} |\sigma_j \otimes \sigma_0\rangle\rangle\langle\langle\sigma_j \otimes \sigma_0|,\\
\alpha_3 &= \frac{\text{Tr}(\hat{\mathcal{R}}_\Lambda \mathbb{P}_3)}{9}, &&\text{and} && \mathbb{P}_3 = \sum_{\sigma_i,\sigma_j=\{\sigma_x,\sigma_y,\sigma_z\}} |\sigma_i \otimes \sigma_j\rangle\rangle\langle\langle\sigma_i \otimes \sigma_j|.
\end{aligned}\tag{38}$$

Note that the Pauli transfer matrix for the error map, $\hat{\mathcal{R}}_\Lambda$, is given by:

$$\begin{aligned}
(\hat{\mathcal{R}}_\Lambda)_{ij,kl} = \text{Tr}\left((\sigma_i^\dagger \otimes \sigma_j^\dagger)\Lambda(\sigma_k \otimes \sigma_l)\right) \implies (\hat{\mathcal{R}}_\Lambda)_{00,00} &= \langle\langle\sigma_0 \otimes \sigma_0|\hat{\mathcal{R}}_\Lambda|\sigma_0 \otimes \sigma_0\rangle\rangle\\
&= \text{Tr}(\hat{\mathcal{R}}_\Lambda \mathbb{P}_0)\\
&= \text{Tr}\left((\sigma_0^\dagger \otimes \sigma_0^\dagger)\Lambda(\sigma_0 \otimes \sigma_0)\right)\\
&= \text{Tr}(\Lambda(\sigma_0 \otimes \sigma_0)) = 1,
\end{aligned}\tag{39}$$

for a trace preserving map.

In the case where the errors in the system are truly uncorrelated, the error map should be given as the tensor product $\Lambda_1 \otimes \Lambda_2$, with $\Lambda_1$ only affecting the qubits of the first subsystem, and likewise for $\Lambda_2$. In this regime, the three $\alpha$ coefficients simplify to:

$$\begin{aligned}
\text{Tr}(\hat{\mathcal{R}}_\Lambda \mathbb{P}_1) &= \sum_{\sigma_j}\langle\langle\sigma_0 \otimes \sigma_j|\hat{\mathcal{R}}_\Lambda|\sigma_0 \otimes \sigma_j\rangle\rangle = \sum_j (\hat{\mathcal{R}}_\Lambda)_{0j,0j}\\
&= \sum_{\sigma_j}\text{Tr}\left((\sigma_0^\dagger \otimes \sigma_j^\dagger)(\Lambda_1 \otimes \Lambda_2)(\sigma_0 \otimes \sigma_j)\right) = \sum_{\sigma_j}\text{Tr}\left((\sigma_0^\dagger \otimes \sigma_j^\dagger)(\Lambda_1(\sigma_0) \otimes \Lambda_2(\sigma_j))\right)\\
&= \sum_{\sigma_j}\text{Tr}\left((\sigma_0^\dagger \Lambda_1(\sigma_0)) \otimes (\sigma_j^\dagger \Lambda_2(\sigma_j))\right) = \sum_{\sigma_j}\underbrace{\text{Tr}(\sigma_0^\dagger \Lambda_1(\sigma_0))}_{=1} \text{Tr}(\sigma_j^\dagger \Lambda_2(\sigma_j))\\
&\implies \boxed{\alpha_1 \underset{\Lambda=\Lambda_1\otimes\Lambda_2}{=} \sum_{\sigma_j} \frac{\text{Tr}(\sigma_j^\dagger \Lambda_2(\sigma_j))}{3}},
\end{aligned}$$

$$\begin{aligned}
\text{Tr}(\hat{\mathcal{R}}_\Lambda \mathbb{P}_2) &= \sum_{\sigma_i}\langle\langle\sigma_i \otimes \sigma_0|\hat{\mathcal{R}}_\Lambda|\sigma_i \otimes \sigma_0\rangle\rangle = \sum_i (\hat{\mathcal{R}}_\Lambda)_{i0,i0}\\
&= \sum_{\sigma_i}\text{Tr}\left((\sigma_i^\dagger \otimes \sigma_0^\dagger)(\Lambda_1 \otimes \Lambda_2)(\sigma_i \otimes \sigma_0)\right) = \sum_{\sigma_i}\text{Tr}\left((\sigma_i^\dagger \otimes \sigma_0^\dagger)(\Lambda_1(\sigma_i) \otimes \Lambda_2(\sigma_0))\right)\\
&= \sum_{\sigma_i}\text{Tr}\left((\sigma_i^\dagger \Lambda_1(\sigma_i)) \otimes (\sigma_0^\dagger \Lambda_2(\sigma_0))\right) = \sum_{\sigma_i}\underbrace{\text{Tr}(\sigma_0^\dagger \Lambda_2(\sigma_0))}_{=1} \text{Tr}(\sigma_i^\dagger \Lambda_1(\sigma_i))\\
&\implies \boxed{\alpha_2 \underset{\Lambda=\Lambda_1\otimes\Lambda_2}{=} \sum_{\sigma_i} \frac{\text{Tr}(\sigma_i^\dagger \Lambda_1(\sigma_i))}{3}},
\end{aligned}$$

$$\text{Tr}(\hat{\mathcal{R}}_\Lambda \mathbb{P}_3) = \sum_{\sigma_i,\sigma_j} \langle\langle \sigma_i \otimes \sigma_j | \hat{\mathcal{R}}_\Lambda | \sigma_i \otimes \sigma_j \rangle\rangle = \sum_{i,j} (\hat{\mathcal{R}}_\Lambda)_{ij,ij}$$

$$= \sum_{\sigma_i,\sigma_j} \text{Tr}\big((\sigma_i^\dagger \otimes \sigma_j^\dagger)(\Lambda_1 \otimes \Lambda_2)(\sigma_i \otimes \sigma_j)\big) = \sum_{\sigma_i,\sigma_j} \text{Tr}\big((\sigma_i^\dagger \otimes \sigma_j^\dagger)(\Lambda_1(\sigma_i) \otimes \Lambda_2(\sigma_j))\big)$$

$$= \sum_{\sigma_i,\sigma_j} \text{Tr}\big((\sigma_i^\dagger \Lambda_1(\sigma_i)) \otimes (\sigma_j^\dagger \Lambda_2(\sigma_j))\big) = \sum_{\sigma_i,\sigma_j} \text{Tr}\big(\sigma_j^\dagger \Lambda_2(\sigma_j)\big) \text{Tr}\big(\sigma_i^\dagger \Lambda_1(\sigma_i)\big)$$

$$\implies \boxed{\alpha_3 \underset{\Lambda=\Lambda_1\otimes\Lambda_2}{=} \sum_{\sigma_i,\sigma_j} \frac{\text{Tr}\big(\sigma_i^\dagger \Lambda_1(\sigma_i)\big)\text{Tr}\big(\sigma_j^\dagger \Lambda_2(\sigma_j)\big)}{9} = \alpha_1\alpha_2,} \tag{40}$$

which means that any deviation from the uncorrelated regime can be assessed through:

$$\boxed{\delta\alpha = \alpha_3 - \alpha_1\alpha_2.} \tag{41}$$

Thus, the quantity $\delta\alpha$ works as a proxy to quantify the level of correlations between the two subsystems. We should be able to estimate it from the outcomes of experiment 3. This is one of the main results of the simultaneous RB protocol.

To arrive at the theoretical prediction for the sequence fidelity, it is convenient to explicitly compute the twirling superoeperator. Given the decomposition in Eq.(37), $\hat{\Lambda}_T$ assumes the same form

$$\hat{\Lambda}_T = \mathbb{P}_0 + \alpha_1\mathbb{P}_1 + \alpha_2\mathbb{P}_2 + \alpha_3\mathbb{P}_3. \tag{42}$$

Recall that our average sequence superoperator, and thus the resulting fidelity, consists of a composition of $m$ twirling maps. This composition becomes a $m-$times matrix multiplication of twirling superoperators. However, the product of any two distinct projectors is zero $\mathbb{P}_i\mathbb{P}_j = 0$, $i \neq j$, and thus:

$$S_m = \Lambda \circ \Lambda_T^{\circ m} \quad \to \quad \hat{S}_m = \hat{\Lambda} \underbrace{\hat{\Lambda}_T \hat{\Lambda}_T \cdots \hat{\Lambda}_T}_{m \text{ times}}$$

$$\hat{S}_m = \hat{\Lambda}\,\mathbb{P}_0 + \alpha_1^m\,\hat{\Lambda}\,\mathbb{P}_1 + \alpha_2^m\,\hat{\Lambda}\,\mathbb{P}_2 + \alpha_3^m\,\hat{\Lambda}\,\mathbb{P}_3, \tag{43}$$

which leads to the average sequence fidelity

$$\begin{aligned}
F_{\text{seq}}(m, E, \rho_0) &= \text{Tr}\big(E_{\rho_0} S_m(\rho)\big) \\
&= \langle\langle E_{\rho_0} | \hat{S}_m | \rho_0 \rangle\rangle \\
&= \langle\langle E_{\rho_0} | \underbrace{\hat{\Lambda}\,\hat{\Lambda}_T^m}_{=\langle\langle\tilde{E}|} | \rho_0 \rangle\rangle \\
&= \langle\langle\tilde{E}|\rho_0^{(W_0)}\rangle\rangle + \alpha_1^m\langle\langle\tilde{E}|\rho_0^{(W_1)}\rangle\rangle + \alpha_2^m\langle\langle\tilde{E}|\rho_0^{(W_2)}\rangle\rangle + \alpha_3^m\langle\langle\tilde{E}|\rho_0^{(W_3)}\rangle\rangle \\
&\Leftrightarrow \quad \boxed{F_{\text{seq}}\big(m, E_{\rho_0}, \rho_0\big) = A_1\alpha_1^m + A_2\alpha_2^m + A_3\alpha_3^m + B_0.}
\end{aligned} \tag{44}$$

Just as in standard RB, the sequence fidelity in Eq.(44) is central to the protocol, as it represents the new fitting model from which we can extract estimates fro the three parameters $\alpha_1$, $\alpha_2$ and $\alpha_3$.

We have defined $|\rho_0\rangle\rangle$ to be the initial state, and used its decomposition onto the invariant subspaces, as follows

$$
\begin{aligned}
|\rho_0\rangle\rangle &= \sum_{\sigma_i, \sigma_j \in \{\sigma_0\} \cup \boldsymbol{\sigma}} \mathrm{Tr}\left(\left(\sigma_i^\dagger \otimes \sigma_j^\dagger\right)\rho_0\right)|\sigma_i \otimes \sigma_j\rangle\rangle \\
&= \underbrace{\mathrm{Tr}\left(\left(\sigma_0^\dagger \otimes \sigma_0^\dagger\right)\rho_0\right)}_{=1}\underbrace{|\sigma_0 \otimes \sigma_0\rangle\rangle}_{:=|\rho_0^{(W_0)}\rangle\rangle} + \underbrace{\sum_{\sigma_i \in \boldsymbol{\sigma}} \mathrm{Tr}\left(\left(\sigma_0^\dagger \otimes \sigma_i^\dagger\right)\rho_0\right)|\sigma_0 \otimes \sigma_i\rangle\rangle}_{:=|\rho_0^{(W_1)}\rangle\rangle} \\
&\quad + \underbrace{\sum_{\sigma_i \in \boldsymbol{\sigma}} \mathrm{Tr}\left(\left(\sigma_i^\dagger \otimes \sigma_0^\dagger\right)\rho_0\right)|\sigma_i \otimes \sigma_0\rangle\rangle}_{:=|\rho_0^{(W_2)}\rangle\rangle} + \underbrace{\sum_{\sigma_i, \sigma_j \in \boldsymbol{\sigma}} \mathrm{Tr}\left(\left(\sigma_i^\dagger \otimes \sigma_j^\dagger\right)\rho_0\right)|\sigma_i \otimes \sigma_j\rangle\rangle}_{:=|\rho_0^{(W_3)}\rangle\rangle} \\
&= |\rho_0^{(W_0)}\rangle\rangle + |\rho_0^{(W_1)}\rangle\rangle + |\rho_0^{(W_2)}\rangle\rangle + |\rho_0^{(W_3)}\rangle\rangle .
\end{aligned}
\tag{45}
$$

As pointed out in Ref. [4], the coefficients $\alpha_i$ can be estimated in the usual RB way, conditional to the ability of preparing states, and measuring operators, belonging to a unique invariant subspace. Yet, this approach does not guarantee robustness against SPAM errors [8]. It is possible to recover SPAM independence with modifications to the protocol, as is done in character RB [30].

### 2.3.2 Experiments 1 and 2

Experiments 1 and 2 perform a single subsystem twirl, i.e. twirl over the group $\mathcal{C}_1 \times \mathcal{I}$, or $\mathcal{I} \times \mathcal{C}_1$, respectively. We consider in detail the case $\mathcal{I} \times \mathcal{C}_1$, since the steps required for the other twirl are essentially the same.

Just as in the previous section, explicit computations for the sequence fidelity require us to examine the question regarding invariant subspaces in our Hilbert space. For this task, it is useful to recall the action of the direct product group in the Pauli basis, namely:

$$
C \in \mathcal{I} \times \mathcal{C}_1 : \ C(\sigma_i \otimes \sigma_j) \to \hat{\sigma}_i \otimes (\hat{C}' \hat{\sigma}_j \hat{C}'^\dagger), \quad \hat{C}' \in \mathcal{C}_1 .
$$

Under this action, it is then clear that the 1D subspace $W_0$ and the 3D subspace $W_1$ remain invariant subspaces. Yet, $W_2$ will now host 3 non-trivial invariant subspaces of its own. To see this, consider what happens to the vector $|\sigma_i \otimes \sigma_0\rangle\rangle$ - because only the second qubit is affected by the group in a non-trivial way, the vector remains invariant, regardless of the particular choice for $\sigma_i \in \boldsymbol{\sigma}$. This then means the three possible choices for $\sigma_i$ decompose into three invariant subspaces, since there is no way to move from one sector to the other, using only operations that preserve the group structure. A similar process happens to $W_3$. The issue that now complicates a straightforward use of Eq.(E.22) is that the decomposition into smaller subspaces does not lead to an increase in the number of distinct irreducible representations. Let us motivate this by considering, for instance, the action of all the elements in $\mathcal{I} \times \mathcal{C}_1$ on the set of vectors $\{|\sigma_0 \otimes \sigma_0\rangle\rangle, |\sigma_i \otimes \sigma_0\rangle\rangle\}$. Clearly, all the elements in the group are performing an identity operation for each of the vectors in the set. The only (irreducible) representation that acts in the same way, for all elements in the group, is the trivial representation. This group then contains 4 copies of the trivial irreducible representation. Using a similar argument, it also follows that for the set of vectors $\{|\sigma_0 \otimes \sigma_i\rangle\rangle, |\sigma_j \otimes \sigma_i\rangle\rangle\}$, the group elements perform identical signed permutations of the second qubit for the the two vectors. However, the subspace spanned by $\{|\sigma_0 \otimes \sigma_i\rangle\rangle\}$ is disjoint from that generated by $\{|\sigma_j \otimes \sigma_i\rangle\rangle\}$ $(i,j \neq 0)$. This originates from the fact that the elements in $\mathcal{I} \times \mathcal{C}_1$ can never map $|\sigma_0 \otimes \sigma_i\rangle\rangle$ to $|\sigma_j \otimes \sigma_i\rangle\rangle$. Hence, the group contains 4 copies of the non-trivial representation, and each copy corresponds to a $3D$ subspace. While Eq.(E.22) can be generalized to the case where the same irreducible

representations occurs multiple times, we opt instead to proceed with the approach used in Ref. [4]. This requires looking directly at the matrix elements of the twirling superoperator:

$$
\begin{aligned}
\left(\hat{\mathcal{R}}_{\Lambda_T}\right)_{ij,kl} &= \frac{1}{|\mathcal{I}\times\mathcal{C}_1|}\sum_{\mathcal{C}\in\mathcal{I}\times\mathcal{C}_1}\left(\hat{\mathcal{R}}_{\mathcal{C}^\dagger}\right)_{ij,mn}\left(\hat{\mathcal{R}}_\Lambda\right)_{mn,pq}\left(\hat{\mathcal{R}}_\mathcal{C}\right)_{pq,kl} \\
&= \frac{1}{|\mathcal{C}_1|}\sum_{\hat{C}_1\in\mathcal{C}_1}\Big[\mathrm{Tr}\Big(\big(\hat{\sigma}_i^\dagger\otimes\hat{\sigma}_j^\dagger\big)\big(\hat{\sigma}_m\otimes\hat{C}_1^\dagger\hat{\sigma}_n\hat{C}_1\big)\Big) \\
&\qquad\times \mathrm{Tr}\Big(\big(\hat{\sigma}_m^\dagger\otimes\hat{\sigma}_n^\dagger\big)\Lambda\big(\hat{\sigma}_p\otimes\hat{\sigma}_q\big)\Big)\,\mathrm{Tr}\Big(\big(\hat{\sigma}_p^\dagger\otimes\hat{\sigma}_q^\dagger\big)\big(\hat{\sigma}_k\otimes\hat{C}_1\hat{\sigma}_l\hat{C}_1^\dagger\big)\Big)\Big] \\
&= \frac{1}{|\mathcal{C}_1|}\sum_{\hat{C}_1\in\mathcal{C}_1}\Big[\mathrm{Tr}\Big(\big(\hat{\sigma}_i^\dagger\hat{\sigma}_m\big)\otimes\big(\hat{\sigma}_j^\dagger\hat{C}_1^\dagger\hat{\sigma}_n\hat{C}_1\big)\Big) \\
&\qquad\times \mathrm{Tr}((\hat{\sigma}_m^\dagger\otimes\hat{\sigma}_n^\dagger)\Lambda\big(\hat{\sigma}_p\otimes\hat{\sigma}_q\big))\,\mathrm{Tr}\Big(\big(\hat{\sigma}_p^\dagger\hat{\sigma}_k\big)\otimes\big(\hat{\sigma}_q^\dagger\hat{C}_1\hat{\sigma}_l\hat{C}_1^\dagger\big)\Big)\Big] \\
&= \frac{1}{|\mathcal{C}_1|}\sum_{\hat{C}_1\in\mathcal{C}_1}\Bigg[\underset{(\sigma_i=\sigma_m)}{\mathrm{Tr}\big(\hat{\sigma}_i^\dagger\hat{\sigma}_m\big)}\underset{(\sigma_n=C_1\sigma_j C_1^\dagger)}{\mathrm{Tr}\big(\hat{\sigma}_j^\dagger\hat{C}_1^\dagger\hat{\sigma}_n\hat{C}_1\big)}\underset{(\sigma_p=\sigma_k)}{\mathrm{Tr}\big(\hat{\sigma}_p^\dagger\hat{\sigma}_k\big)} \\
&\qquad\times \underset{(\sigma_q=C_1\sigma_l C_1^\dagger)}{\mathrm{Tr}\big(\hat{\sigma}_q^\dagger\hat{C}_1\hat{\sigma}_l\hat{C}_1^\dagger\big)}\mathrm{Tr}\big(\big(\hat{\sigma}_m^\dagger\otimes\hat{\sigma}_n^\dagger\big)\Lambda\big(\hat{\sigma}_p\otimes\hat{\sigma}_q\big)\big)\Bigg] \\
&= \frac{1}{|\mathcal{C}_1|}\sum_{\hat{C}_1\in\mathcal{C}_1}\mathrm{Tr}\Big(\big(\hat{\sigma}_i^\dagger\otimes\big(\hat{C}_1\hat{\sigma}_j^\dagger\hat{C}_1^\dagger\big)\big)\Lambda\big(\hat{\sigma}_k\otimes\big(\hat{C}_1\hat{\sigma}_l\hat{C}_1^\dagger\big)\big)\Big)\,.
\end{aligned}
\tag{46}
$$

Let us consider what happens when $\sigma_j\neq\sigma_l$, and at least one of them is not the identity. To illustrate this more clearly, let us imagine the case where $\sigma_j=\sigma_x$ and $\sigma_l=\sigma_y$:

$$
\hat{C}_1\hat{\sigma}_x\hat{C}_1^\dagger\rightarrow\begin{cases}\pm\hat{\sigma}_x\,,\\ \pm\hat{\sigma}_y\,,\\ \pm\hat{\sigma}_z\,,\end{cases}\qquad \hat{C}_1\hat{\sigma}_y\hat{C}_1^\dagger\rightarrow\begin{cases}\pm\hat{\sigma}_x\,,\\ \pm\hat{\sigma}_y\,,\\ \pm\hat{\sigma}_z\,.\end{cases}
\tag{47}
$$

The particular sign and Pauli onto which $\sigma_x$ is mapped on, for example, depends on the specific choice for the Clifford element $C_1$. However, as we sum over all elements in the group, all possible sign configurations are generated in an uniform way. This is because all of them preserve the trace and commutation constraints, and so are allowed under the group action. Hence, if $\sigma_j\neq\sigma_l$, the matrix elements in Eq.(46) vanish. The only option is then to have $\sigma_l=\sigma_j$, which results in $\left(\hat{\mathcal{R}}_{\Lambda_T}\right)_{ij,kl}=\delta_{j,l}\left(\hat{\mathcal{R}}_{\Lambda_T}\right)_{ij,kj}$ (see appendix F for further details). For the surviving matrix elements, we can distinguish two separate cases, $j=0$ and $j\neq0$. This distinctness is again due to the fact that the identity is left invariant by the Clifford group, and all remaining Pauli elements are just re-shuffled among themselves. The final form for the matrix elements is then expressed as follows:

$$
\left(\hat{\mathcal{R}}_{\Lambda_T}\right)_{ij,kl}=\begin{cases}\left(\hat{\mathcal{R}}_\Lambda\right)_{i0,k0}=\mathrm{Tr}((\sigma_i\otimes\sigma_0)\Lambda(\sigma_k\otimes\sigma_0))\equiv\alpha_{i,k}^{(0)}\,,\\[2mm] \dfrac{1}{(d_1^2-1)}\sum_{j=1}^3\left(\hat{\mathcal{R}}_\Lambda\right)_{ij,kj}=\dfrac{1}{(d_1^2-1)}\sum_{j=1}^3\mathrm{Tr}\big((\sigma_i\otimes\sigma_j)\Lambda\big(\sigma_k\otimes\sigma_j\big)\big)\equiv\alpha_{i,k}^{(1)}\,,\\[2mm] 0\,,\qquad\text{otherwise.}\end{cases}
\tag{48}
$$

The resulting twirling operator is then

$$\hat{\Lambda}_T = \sum_{\sigma_i,\sigma_k\in\{\sigma_0\}\cup\boldsymbol{\sigma}} \alpha_{i,k}^{(0)} |\sigma_i\otimes\sigma_0\rangle\rangle\langle\langle\sigma_k\otimes\sigma_0| + \sum_{\sigma_i,\sigma_k\in\{\sigma_0\}\cup\boldsymbol{\sigma}} \alpha_{i,k}^{(1)} \sum_{\sigma_l\in\boldsymbol{\sigma}} |\sigma_i\otimes\sigma_j\rangle\rangle\langle\langle\sigma_k\otimes\sigma_j|$$

$$= \sum_{\sigma_i,\sigma_k\in\{\sigma_0\}\cup\boldsymbol{\sigma}} \alpha_{i,k}^{(0)} \; \mathbb{P}_{i,k}^{(0)} + \sum_{\sigma_i,\sigma_k\in\{\sigma_0\}\cup\boldsymbol{\sigma}} \alpha_{i,k}^{(1)} \; \mathbb{P}_{i,k}^{(1)}, \tag{49}$$

written in terms of the linear operators:

$$\mathbb{P}_{i,k}^{(0)} := |\sigma_i\otimes\sigma_0\rangle\rangle\langle\langle\sigma_k\otimes\sigma_0|,$$

$$\mathbb{P}_{i,k}^{(1)} := \sum_{\sigma_j\in\boldsymbol{\sigma}} |\sigma_i\otimes\sigma_j\rangle\rangle\langle\langle\sigma_k\otimes\sigma_j|. \tag{50}$$

The operator $\mathbb{P}_{i,k}^{(l)}$ is responsible for coupling two subspaces, $i$ and $k$, representing the same irreducible representation $l$. Note that when $i = k$, $\mathbb{P}_{i,i}^{(l)}$ can be directly recognized as the projector onto the subspace related to the $i$th copy of the irreducible representation $l$. Furthermore, since the set of vectors $\{|\sigma_k\otimes\sigma_i\rangle\rangle\}$ forms an orthonormal basis, $\mathbb{P}_{i,k}^{(l)}\mathbb{P}_{i,k}^{(l')} = \delta_{l,l'} \, \mathbb{P}_{i,k}^{(l)}$. Thus, the $m$ successive applications of $\hat{\Lambda}_T$ simplify to:

$$\hat{\Lambda}_T^m = \sum_{\sigma_i,\sigma_k\in\{\sigma_0\}\cup\boldsymbol{\sigma}} \left(\alpha_{i,k}^{(0)}\right)^m \; \mathbb{P}_{i,k}^{(0)} + \sum_{\sigma_i,\sigma_k\in\{\sigma_0\}\cup\boldsymbol{\sigma}} \left(\alpha_{i,k}^{(1)}\right)^m \; \mathbb{P}_{i,k}^{(1)}.$$

Twirling over subsystem 2 leads to an average sequence fidelity with possibly many more decaying parameters. Indeed, if before $F_{\text{seq}}(m,E,\rho_0)$ decomposed into 3 decaying parameters, now, we may have as many as 31 terms contributing to the decaying of the average sequence fidelity as a function of the circuit's length ($m$):

$$F_{\text{seq}}(m,E,\rho_0) = \text{Tr}\left(E_{\rho_0} S_m(\rho)\right) = \langle\langle E_{\rho_0}|\hat{S}_m|\rho_0\rangle\rangle$$

$$= \underbrace{\langle\langle E_{\rho_0}|\hat{\Lambda}}_{=\langle\langle\tilde{E}|} \hat{\Lambda}_T^m|\rho_0\rangle\rangle = \langle\langle\tilde{E}|\hat{\Lambda}_T^m|\rho_0\rangle\rangle$$

$$= \sum_{\sigma_i,\sigma_k\in\{\sigma_0\}\cup\boldsymbol{\sigma}} \left(\alpha_{i,k}^{(0)}\right)^m \underbrace{\langle\langle\tilde{E}|\mathbb{P}_{i,k}^{(0)}|\rho_0\rangle\rangle}_{=c_{i,k}^{(0)}} + \sum_{\sigma_i,\sigma_k\in\{\sigma_0\}\cup\boldsymbol{\sigma}} \left(\alpha_{i,k}^{(1)}\right)^m \underbrace{\langle\langle\tilde{E}|\mathbb{P}_{i,k}^{(1)}|\rho_0\rangle\rangle}_{c_{i,k}^{(1)}} \tag{51}$$

$$= c_{0,0}^{(0)} + \sum_{(\sigma_i,\sigma_k\in\{\sigma_0\}\cup\boldsymbol{\sigma})/\sigma_i=\sigma_k=\sigma_0} \left(\alpha_{i,k}^{(0)}\right)^m c_{i,k}^{(0)} + \sum_{\sigma_i,\sigma_k\in\{\sigma_0\}\cup\boldsymbol{\sigma}} \left(\alpha_{i,k}^{(1)}\right)^m c_{i,k}^{(1)},$$

where we have used the fact that $\alpha_{0,0}^{(0)} = \text{Tr}\left((\sigma_0\otimes\sigma_0)\Lambda(\sigma_0\otimes\sigma_0)\right) = \frac{1}{4}\text{Tr}\left(\Lambda(\mathbb{1}\otimes\mathbb{1})\right) = 1$, with $\Lambda$ a trace preserving map. All the other $\alpha_{i,k}^{(l)}$ will, in general, be smaller than one, and therefore will foster a decaying fidelity with the circuit's length. Note that while the functional form of the sequence fidelity appears now more complex, this feature is in line with more general RB models that are not a twirling over the full Clifford group [2].

The sequence fidelity in Eq.(51) defines the fitting model for experiment two. It is therefore of direct practical importance for the implementation of the full protocol.

### 2.3.3 Simultaneous randomized benchmarking: Example

To bring more clarity on how simultaneous RB can shed light on the presence of crosstalk, let us consider an explicit example, based on the following toy model for the error model $\Lambda$:

$$\Lambda(\rho) = \epsilon_1 \left( p_{0,1} (\mathbb{1}\otimes\mathbb{1})\rho(\mathbb{1}\otimes\mathbb{1}) + p_{1,1} (X\otimes\mathbb{1})\rho(X\otimes\mathbb{1}) + p_{2,1} (\mathbb{1}\otimes X)\rho(\mathbb{1}\otimes X) \right)$$

$$+ \epsilon_2 \left( p_{0,2} (\mathbb{1}\otimes\mathbb{1})\rho(\mathbb{1}\otimes\mathbb{1}) + p_{1,2} (\mathbb{1}\otimes X)\rho(\mathbb{1}\otimes X) + p_{2,2} (X\otimes\mathbb{1})\rho(X\otimes\mathbb{1}) \right) \tag{52}$$

$$+ p' (Z\otimes Z)\rho(Z\otimes Z).$$

Note that this simply corresponds to expressing $\Lambda$ in terms of a specific set of Kraus operators, using the Pauli basis.[13] In doing so, we end up with what is known as the process matrix representation, i.e. $\Lambda(\rho) = \sum_{i,j} \chi_{j,k} P_j \rho P_k$ [31]. It's readily visible that the resulting $\chi-$matrix is positive semi-definite.[14] This guarantees that $\Lambda$ is a CP map. The trace preserving (TP) property is fulfilled if the probability distributions satisfy the constraint:

$$\sum_{j=1}^{2} \sum_{i=0}^{2} \epsilon_j \, p_{i,j} = 1 \,.$$

In Eq.(52), both $\epsilon_1$ and $\epsilon_2$ are binary parameters, i.e. $\epsilon_j = \{0, 1\}$, whose choice of values are meant to represent the error map for different experiments:

1. $\epsilon_1 = 1$ and $\epsilon_2 = 0$: Corresponds to an RB experiment only on the first qubit, with the second qubit idle.

2. $\epsilon_2 = 1$ and $\epsilon_1 = 0$: Corresponds to an RB experiment only on the second qubit, with the first qubit idle.

3. $\epsilon_1 = 1$ and $\epsilon_2 = 1$: Simultaneous RB experiment on both qubits.

The choice of notation is the following. For $\epsilon_j \neq 0$ and $\epsilon_i = 0$, the term $p_{0,j}$ is the probability of no errors occurring, $p_{1,j}$ is the probability of errors occurring on qubit $j$, while successfully maintaining qubit $i$ idle. Then, the probability $p_{2,j}$ corresponds to the idle qubit being affected by operations on the target qubit $j$. The term $p'$ represents an always-on interaction term between the two qubits.

Let us assume that our input state is $\rho_0 = |00\rangle\langle 00|$. In the Liouville representation, $\rho_0$ is vectorized and can be expressed as follows:

$$|00\rangle\langle 00| \rightarrow |00\rangle\rangle = \frac{1}{2} \left( |\sigma_0 \otimes \sigma_0\rangle\rangle + |\sigma_0 \otimes \sigma_z\rangle\rangle + |\sigma_z \otimes \sigma_0\rangle\rangle + |\sigma_z \otimes \sigma_z\rangle\rangle \right) \,. \tag{53}$$

Since we have an explicit model for the error map, we have all the information required to evaluate all parameters of the expected average sequence fidelity, as a function of the circuit length. Let us start with computing this object for the case where both qubits are each simultaneously undergoing an RB experiment ($\epsilon_1 = \epsilon_2 = 1$). Recall that this RB experiment entails selecting gates from the direct product group $\mathcal{C}_1 \times \mathcal{C}_1$. As seen previously, the resulting average sequence fidelity is then (see Eq.(44)):

$$\begin{aligned}
F_{\text{seq}}^{(\mathcal{C}_1 \times \mathcal{C}_1)}(m, E, \rho_0) &= \langle\langle \tilde{E}|\rho_0^{(W_0)}\rangle\rangle + \alpha_1^m \langle\langle \tilde{E}|\rho_0^{(W_1)}\rangle\rangle + \alpha_2^m \langle\langle \tilde{E}|\rho_0^{(W_2)}\rangle\rangle + \alpha_3^m \langle\langle \tilde{E}|\rho_0^{(W_3)}\rangle\rangle \\
&\underset{Eq.(53)}{=} \langle\langle E|\Lambda|\rho_0^{(W_0)}\rangle\rangle + \alpha_1^m \langle\langle E|\Lambda|\rho_0^{(W_1)}\rangle\rangle + \alpha_2^m \langle\langle E|\Lambda|\rho_0^{(W_2)}\rangle\rangle + \alpha_3^m \langle\langle E|\Lambda|\rho_0^{(W_3)}\rangle\rangle \\
&= \langle\langle \rho_0|\Lambda(\sigma_0 \otimes \sigma_0)\rangle\rangle + \alpha_1^m \langle\langle \rho_0|\Lambda(\sigma_0 \otimes \sigma_z)\rangle\rangle \\
&\quad + \alpha_2^m \langle\langle \rho_0|\Lambda(\sigma_z \otimes \sigma_0)\rangle\rangle + \alpha_3^m \langle\langle \rho_0|\Lambda(\sigma_z \otimes \sigma_z)\rangle\rangle \,,
\end{aligned}$$

where we have assumed the absence of SPAM errors, so that we may write $E = \rho_0$. This is a simplification that, in general, will not hold true. We have chosen it in the spirit of keeping this example as simple as possible. The resulting average sequence fidelity reads:

$$F_{\text{seq}}^{(\mathcal{C}_1 \times \mathcal{C}_1)}(m, E, \rho_0) = \frac{1}{4} + \frac{c_1}{4} \, \alpha_1^m + \frac{c_2}{4} \, \alpha_2^m + \frac{c_3}{4} \, \alpha_3^m \,, \tag{54}$$

---

[13]For an introduction to Kraus operators, see Ref. [16], chapter 8, section 8.2.3, where there they are referred to as the operator-sum representation.

[14]Our simplified choice for the error map leads to a $\chi-$matrix that is diagonal in the Pauli basis, with all entries being real valued and $\geq 0$.

Table 2: The following table illustrates how $\Lambda$ acts on all possible states $(\sigma_i \otimes \sigma_j)$. The lines denote particular choices for $(\sigma_i \otimes \sigma_j)$, which are then fed into the channel $\Lambda$ to produce the resulting $\Lambda(\sigma_i \otimes \sigma_j)$. These results are displayed in the second column of the table.

| | $\Lambda(.)$ |
|---|---|
| $\sigma_0 \otimes \sigma_0$ | $\sigma_0 \otimes \sigma_0$ |
| $\sigma_0 \otimes \sigma_x$ | $(1-2p')(\sigma_0 \otimes \sigma_x)$ |
| $\sigma_0 \otimes \sigma_y$ | $\left(1-2(\epsilon_2 p_{1,2} + \epsilon_1 p_{2,1} + p')\right)(\sigma_0 \otimes \sigma_y)$ |
| $\sigma_0 \otimes \sigma_z$ | $\left(1-2(\epsilon_1 p_{2,1} + \epsilon_2 p_{1,2})\right)(\sigma_0 \otimes \sigma_z)$ |
| $\sigma_x \otimes \sigma_0$ | $(1-2p')(\sigma_x \otimes \sigma_0)$ |
| $\sigma_y \otimes \sigma_0$ | $\left(1-2(\epsilon_1 p_{1,1} + \epsilon_2 p_{2,2} + p')\right)(\sigma_y \otimes \sigma_0)$ |
| $\sigma_z \otimes \sigma_0$ | $\left(1-2(\epsilon_1 p_{1,1} + \epsilon_2 p_{2,2})\right)(\sigma_z \otimes \sigma_0)$ |
| $\sigma_x \otimes \sigma_x$ | $\sigma_x \otimes \sigma_x$ |
| $\sigma_x \otimes \sigma_y$ | $\left(1-2(\epsilon_2 p_{1,2} + \epsilon_1 p_{2,1})\right)(\sigma_x \otimes \sigma_y)$ |
| $\sigma_x \otimes \sigma_z$ | $\left(1-2(\epsilon_2 p_{1,2} + \epsilon_1 p_{2,1} + p')\right)(\sigma_x \otimes \sigma_x)$ |
| $\sigma_y \otimes \sigma_x$ | $\left(1-2(\epsilon_1 p_{1,1} + \epsilon_2 p_{2,2})\right)(\sigma_y \otimes \sigma_x)$ |
| $\sigma_y \otimes \sigma_y$ | $\left(2(\epsilon_1 p_{0,1} + \epsilon_2 p_{0,2} + p')-1\right)(\sigma_y \otimes \sigma_y)$ |
| $\sigma_y \otimes \sigma_z$ | $\left(2(\epsilon_1 p_{0,1} + \epsilon_2 p_{0,2})-1\right)(\sigma_y \otimes \sigma_z)$ |
| $\sigma_z \otimes \sigma_x$ | $\left(1-2(\epsilon_1 p_{1,1} + \epsilon_2 p_{2,2} + p')\right)(\sigma_z \otimes \sigma_x)$ |
| $\sigma_z \otimes \sigma_y$ | $\left(2(\epsilon_1 p_{0,1} + \epsilon_2 p_{0,2})-1\right)(\sigma_z \otimes \sigma_y)$ |
| $\sigma_z \otimes \sigma_z$ | $\left(2(\epsilon_1 p_{0,1} + \epsilon_2 p_{0,2} + p')-1\right)(\sigma_z \otimes \sigma_z)$ |

where:

$$\begin{aligned}
c_1 &= 1 - 2(p_{1,2} + p_{2,1}), \\
c_2 &= 1 - 2(p_{1,1} + p_{2,2}), \\
c_3 &= 2(p_{0,1} + p_{0,2} + p') - 1.
\end{aligned} \tag{55}$$

To arrive at Eq.(55), we made used of the TP constraint on the probability densities, as well as knowledge on how $\Lambda$ acts on the set of states $(\sigma_i \otimes \sigma_j)$. Below we provide a table summarizing the action of the error map on the states $(\sigma_i \otimes \sigma_j)$: Using table 2, we can compute the set of $\{\alpha\}$ coefficients for the simultaneous RB experiment ($\epsilon_1 = \epsilon_2 = 1$):

$$\begin{aligned}
\alpha_1 &= \frac{1}{3}\mathrm{Tr}\left(\hat{\mathcal{R}}_\Lambda \mathbb{P}_1\right) \\
&= \frac{1}{3} \sum_{j=\{x,y,z\}} \langle\langle \sigma_0 \otimes \sigma_j | \Lambda(\sigma_0 \otimes \sigma_j) \rangle\rangle
\end{aligned} \tag{56}$$

$$\Leftrightarrow \boxed{\alpha_1 = \frac{1}{3}\left(1 + 2(c_1 - 2p')\right),}$$

$$\begin{aligned}
\alpha_2 &= \frac{1}{3}\mathrm{Tr}\left(\hat{\mathcal{R}}_\Lambda \mathbb{P}_2\right) \\
&= \frac{1}{3} \sum_{j=\{x,y,z\}} \langle\langle \sigma_j \otimes \sigma_0 | \Lambda(\sigma_j \otimes \sigma_0) \rangle\rangle
\end{aligned} \tag{57}$$

$$\Leftrightarrow \boxed{\alpha_2 = \frac{1}{3}\left(1 + 2(c_2 - 2p')\right),}$$

$$\alpha_3 = \frac{1}{9}\text{Tr}\big(\hat{\mathcal{R}}_\Lambda \mathbb{P}_3\big)$$

$$= \frac{1}{9}\sum_{i,j=\{x,y,z\}} \langle\langle \sigma_i \otimes \sigma_j | \Lambda(\sigma_i \otimes \sigma_j)\rangle\rangle \tag{58}$$

$$\Leftrightarrow \quad \boxed{\alpha_3 = \frac{1}{9}\big(1 + 2(c_1 + c_2 + 2c_3 - 4p')\big).}$$

Note that if we were to numerically simulate the simultaneous RB experiment, we would be required to fit our average sequence fidelity data to a model with multiple decaying parameters (Eq.(54)). This makes the fitting task much more complex compared to the case of standard RB. Since we are operating under the assumption of no SPAM errors, we could circumvent this challenge by also keeping track of the expectation values $\langle z_1 \rangle := \langle \sigma_z \otimes \sigma_0 \rangle$, $\langle z_2 \rangle := \langle \sigma_0 \otimes \sigma_z \rangle$ and $\langle zz \rangle := \langle \sigma_z \otimes \sigma_z \rangle$. This is fruitful because each of them belongs to a unique irreducible subspace, and thus retains only the $\alpha-$coefficient related to that same subspace, namely:

$$\langle z_1 \rangle = \text{Tr}\Big[z_1 \Lambda\big(\Lambda_T^{\circ m}(\rho)\big)\Big] = \langle\langle \sigma_z \otimes \sigma_0 | \hat{\Lambda}\hat{\Lambda}_T^m | \rho_0 \rangle\rangle = \frac{c_2}{2}\alpha_2^m \,,$$

$$\langle z_2 \rangle = \text{Tr}\Big[z_2 \Lambda\big(\Lambda_T^{\circ m}(\rho)\big)\Big] = \langle\langle \sigma_0 \otimes \sigma_z | \hat{\Lambda}\hat{\Lambda}_T^m | \rho_0 \rangle\rangle = \frac{c_1}{2}\alpha_1^m \,, \tag{59}$$

$$\langle zz \rangle = \text{Tr}\Big[zz \Lambda\big(\Lambda_T^{\circ m}(\rho)\big)\Big] = \langle\langle \sigma_z \otimes \sigma_z | \hat{\Lambda}\hat{\Lambda}_T^m | \rho_0 \rangle\rangle = \frac{c_3}{2}\alpha_3^m \,.$$

We still need to gather information on the parameters for the single subsystem RB experiments. This requires us to evaluate the twirling operator over the groups $\mathcal{C}_1 \times \mathcal{I}$ and $\mathcal{I} \times \mathcal{C}_1$, representing the RB experiments only over qubit 1 and 2, respectively. Let us start with qubit 2, for which we have already constructed $\Lambda_T$ (Eq.(49)). Using Eq.(48), and our knowledge of $\Lambda$, we can directly evaluate all the $\alpha_{k,k'}^{(j)}$ coefficients. Note that due to the simplicity of our error map, $\Lambda(\sigma_i \otimes \sigma_j) \propto (\sigma_i \otimes \sigma_j) \implies \alpha_{k,k'}^{(j)} \propto \delta_{k,k'}$. Moreover, the action of $\Lambda_T^m$ on the input state $|\rho_0\rangle\rangle$ further reduces the number of $\{\alpha\}$ coefficients we need to evaluate. We are left with the set $\{\alpha_{0,0}^{(0)}, \alpha_{3,3}^{(0)}, \alpha_{0,0}^{(1)}, \alpha_{3,3}^{(1)}\}$, where:

$$\alpha_{k,k}^{(0)} = \text{Tr}\Big[(\sigma_k \otimes \sigma_0)\Lambda(\sigma_k \otimes \sigma_0)\Big]$$

$$= \begin{cases} 1 \,, & k = 0 \,, \\ p_{0,2} + p_{1,2} - p_{2,2} + p' \,, & k = 3 \ (\implies \sigma_z) \,, \end{cases} \tag{60}$$

$$\alpha_{k,k}^{(1)} = \frac{1}{3}\sum_{l=1}^{3}\text{Tr}\Big[(\sigma_k \otimes \sigma_l)\Lambda(\sigma_k \otimes \sigma_l)\Big]$$

$$= \begin{cases} \frac{1}{3}\big((1 - 2p') + 2(p_{0,2} - p_{1,2} + p_{2,2})\big) \,, & k = 0 \,, \\ (p_{0,2} - p_{2,2}) - \frac{1}{3}(p' + p_{1,2}) \,, & k = 3 \,. \end{cases} \tag{61}$$

This simplifies the average sequence fidelity to (Eq.(51)):

$$F_{\text{seq}}^{(\mathcal{I} \times \mathcal{C}_1)}(m, E, \rho_0) = c_{0,0}^{(0)} + c_{3,3}^{(0)}\big(\alpha_{3,3}^{(0)}\big)^m + c_{0,0}^{(1)}\big(\alpha_{0,0}^{(1)}\big)^m + c_{3,3}^{(1)}\big(\alpha_{3,3}^{(0)}\big)^m \,, \tag{62}$$

with

$$c_{0,0}^{(0)} = \frac{1}{2}\langle\langle \rho_0 | \Lambda(\sigma_0 \otimes \sigma_0)\rangle\rangle = \frac{1}{4} \,,$$

$$c_{3,3}^{(0)} = \frac{1}{2}\langle\langle \rho_0 | \Lambda(\sigma_z \otimes \sigma_0)\rangle\rangle = \frac{1}{4}(1 - 2p_{2,2}) \,,$$

$$c_{0,0}^{(1)} = \frac{1}{2}\langle\langle \rho_0 | \Lambda(\sigma_0 \otimes \sigma_z)\rangle\rangle = \frac{1}{4}(1 - 2p_{1,2}) \,, \tag{63}$$

$$c_{3,3}^{(1)} = \frac{1}{2}\langle\langle \rho_0 | \Lambda(\sigma_z \otimes \sigma_z)\rangle\rangle = \frac{1}{4}(2p' + 2p_{0,2} - 1) \,.$$

To obtain information only on qubit 2, it would be useful to trace out quibt 1. Recall that the sequence fidelity is nothing more than the (average) probability of retrieving $\rho_0$ as output. Likewise, the quantity $\langle\langle 10|\hat{\Lambda}\hat{\Lambda}_T^m|\rho_0\rangle\rangle$ is the (average) probability of getting the state $|10\rangle\langle 10|$ as output. Thinking in these terms is helpful, because it allows us to effectively trace out qubit 1, according to the law of total probability. Denoting qubit 1 by $Q_1$ and qubit 2 by $Q_2$, we can extract the probability of qubit 2 ending up in the state $|0\rangle\langle 0|$ according to:

$$\Pr(Q_2=\uparrow)=\Pr(Q_1=\uparrow\cap Q_2=\uparrow)+\Pr(Q_1=\downarrow\cap Q_2=\uparrow)=\underbrace{\langle\langle\rho_0|\hat{\Lambda}\hat{\Lambda}_T^m|\rho_0\rangle\rangle}_{=F_{\text{seq}}^{(\mathcal{I}\times\mathcal{C}_1)}(m,E,\rho_0)}+\langle\langle 10|\hat{\Lambda}\hat{\Lambda}_T^m|\rho_0\rangle\rangle$$

$$\Leftrightarrow\quad \Pr(Q_2=\uparrow)=\frac{1}{2}+\underbrace{\frac{(1-2p_{1,2})}{2}}_{:=\tilde{c}_1/2}\left(\alpha_{0,0}^{(1)}\right)^m.$$

(64)

Analogously, this same probability can also be computed in the case where the gates are extracted from the direct product group $\mathcal{C}_1\times\mathcal{C}_1$ (i.e. in experiment 3),

$$\Pr^{(\mathcal{C}_1\times\mathcal{C}_1)}(Q_2=\uparrow)=\frac{1}{2}+\frac{c_1}{2}\alpha_1^m.$$

(65)

Below, we compare the coefficients contributing to the respective probabilities side by side,

Table 3: Coefficients contributing for the probability $\Pr(Q_2=\uparrow)$ for the benchmarking groups $\mathcal{I}\times\mathcal{C}_1$ and $\mathcal{C}_1\times\mathcal{C}_1$.

| $\mathcal{I}\times\mathcal{C}_1$ | $\mathcal{C}_1\times\mathcal{C}_1$ |
|---|---|
| $\tilde{c}_1=(1-2p_{1,2})$ | $c_1=1-2(p_{1,2}+p_{2,1})$ |
| $\alpha_{0,0}^{(1)}=\frac{1}{3}(1+2(b_2-2p')),\quad b_2=1-2p_{1,2}$ | $\alpha_1=\frac{1}{3}(1+2(c_1-2p')),\quad c_1=1-2(p_{1,2}+p_{2,1})$ |

Thus, we expect exactly the same RB decaying profiles for $\Pr(Q_2=\uparrow)$ in both experiments, if $p_{2,1}=0$. Note that this doesn't translate into an absence of crosstalk, since $p'$ contributes to both probabilities in an equal manner. In the context of our noise model, it means that if there is no extra noise on qubit 2 due to operations on qubit 1, we will not be able to differentiate the RB curves for $\Pr(Q_2=\uparrow)$, even if there are other sources of crosstalk in the system.

In a similar way, we can construct $\Pr(Q_1=\uparrow)$, but now using the group $\mathcal{C}_1\times\mathcal{I}$. As before, it is instructive to compare $\Pr(Q_1=\uparrow)$ evaluated from simultaneous RB and when only $Q_1$ is twirled:

Table 4: Coefficients contributing for the probability $\Pr(Q_1=\uparrow)$ for the benchmarking groups $\mathcal{C}_1\times\mathcal{I}_1$ and $\mathcal{C}_1\times\mathcal{C}_1$.

| $\mathcal{C}_1\times\mathcal{I}$ | $\mathcal{C}_1\times\mathcal{C}_1$ |
|---|---|
| $\Pr(Q_1=\uparrow)=\frac{1}{2}+\frac{b_1}{2}\left(\alpha_{0,0}^{(1)}\right)^m$ | $\Pr(Q_1=\uparrow)=\frac{1}{2}+\frac{c_2}{2}\left(\alpha_2\right)^m$ |
| $b_1=1-2p_{1,1}$ | $c_2=1-2(p_{1,1}+p_{2,2})$ |
| $\alpha_{0,0}^{(1)}=\frac{1}{3}\left(1+2(b_1-2p')\right)$ | $\alpha_2=\frac{1}{3}\left(1+2(c_2-2p')\right)$ |

The two RB experiments yield the same RB curve for $\Pr(Q_1=\uparrow)$, if $p_{2,2}=0$. The condition $p_{2,2}=0$ implies that operating on qubit 2 induces no errors on qubit 1. Hence, for both $\Pr(Q_1=\uparrow)$ and $\Pr(Q_2=\uparrow)$, if an interaction term is always present in the background, and this is the dominate source of errors ($p'>>p_{2,1},p_{2,2}$), simply looking at the RB curves is not enough to flag the degree of crosstalk in the system. Yet, the presence of a non-zero $p'$ impacts

the value of the correlation coefficient $\delta\alpha$, and a non-zero value should raise suspicion for possible qubit-qubit interactions as sources of errors. On the other hand, if the dominant errors arise from accidentally driving the idle qubit, then we expect to observe a mismatch between the RB decay profiles, obtained from simultaneous RB and sub-system RB. In this case, such a mismatch should also be accompany by a $\delta\alpha \neq 0$. This illustrates both the usefulness and the limitations of $\delta\alpha$ as a crosstalk metric: While it is simple to evaluate, and instructive as a warning for the presence of crosstalk, it does not provide further insight on the *identity* of the crosstalk errors.

## 2.4 Discussion

Simultaneous RB addresses two limitations of standard RB: The ability to characterize errors in a subset of qubits, and the task of quantifying the degree of cross-talk in the system. For the former, the technique relies on partitioning the full $n-$qubit system into smaller subsystems. These can then be assessed to estimate their intrinsic error metrics [4]. The second goal of simultaneous RB is to provide insight on the issue of crosstalk. To attain this information, standard RB is performed on all individual sub-systems, simultaneously. From the point of view of group twirling, this task requires twirling with the direct product group $\mathcal{C}_{n_1} \otimes \mathcal{C}_{n_2} \otimes \cdots \otimes \mathcal{C}_{n_l}$, where $\mathcal{C}_{n_i}$ denotes the Clifford group of $n_i$ qubits. The result is the prediction of a sequence fidelity with many distinct contributing decaying parameters. As seen here, these can be used to flag the presence of correlations in the system. While this is a clear improvement to the case where only knowledge on the average error rate is retrieved, some hurdles still remain. One of these difficulties is due to the multi-parameter fitting problem that naturally arises when probing all subsystems at the same time. One alternative procedure, is the use of a framework like character RB [30]. Another limitation has to do with the fact that the metric used for signaling correlations does not allow for identifying the weight or locality of these errors. To this end, correlated RB [5] offers a possible route to dissect this information from the simultaneous RB data (this will be discussed in the next section).

It is crucial to be able to examine larger multi-qubit systems. This is specially the case because decomposing the circuit into the smaller native gates, and benchmarking only those, might not be enough to provide an accurate picture on the degree of crosstalk. But doing simultaneous RB in larger systems might not always be a straightforward task. In particular, with larger sub-systems, the number of invariant subspaces will increase. Neglecting possible technical difficulties related to the ability of identifying all irreducible representations, we stumble again with a multi-parameter fidelity. These factors combined corroborate the statement that detecting all possible forms of crosstalk is generally hard, because, as the system size grows, we will unavoidably stumble onto an exponentially growing number of partitions that need to be tested [32].

## 2.5 Key results in chapter 2

Here we summarize the main takeaways for the simultaneous RB protocol.

- Simultaneous RB allows error characterization for a subset of qubits (qubit subsystem) and provides insight into the presence of crosstalk errors.

- The simultaneous RB protocol requires us to define partitions in our qubit system, where we focus on performing qubit operations only on these subsets.

- The simultaneous RB protocol involves three experiments: Running RB in the first qubit subsystem; running RB in the second qubit subsystem; and finally running

RB on both qubit subsystems simultaneously.

- The resulting RB sequence fidelities can be projected onto each qubit subsystem. This allows three separate effective fidelities to be calculated for any desired subsystem, each from one of the three experiments. Comparing the resulting fidelity decay profiles for Experiment 1 and Experiment 3, and then Experiment 2 with Experiment 3, can serve as a witness to crosstalk errors. For example, operational crosstalk errors, where operations in the first qubit subsystem inadvertently lead to unintended operations in the second qubit subsystem, cause the fidelity in the third experiment to decay much faster than in the first.

- For a two-qubit system, the sequence fidelity in the third experiment is given by:

$$F_{\text{seq}}(m) = A_1 \alpha_1^m + A_2 \alpha_2^m + A_3 \alpha_3^m + B_0 \,,$$

where $A_1$, $A_2$ and $B_0$ absorb all SPAM dependent error terms. In this way, simultaneous RB is also robust to SPAM, but requires solving a harder fitting problem to extract reliable estimates of the desired parameters $\alpha_1$, $\alpha_2$ and $\alpha_3$.

- A quantitative crosstalk metric can be constructed from the estimated parameters in experiment 3. For the case of a two-qubit system, this metric, denoted as $\delta\alpha$, is given by:

$$\delta\alpha = \alpha_3 - \alpha_1 \alpha_2 \,.$$

# 3 Correlated randomized benchmarking

Simultaneous RB provides us with a metric capable of flagging the presence of crosstalk ($\delta\alpha$, Eq.(41)). Yet, even for our very simple example, it is not obvious how to use $\delta\alpha$ to distill different sources of crosstalk. In this section, we will explore an extension of simultaneous RB, designed precisely to improve on this issue: Correlated randomized benchmarking [5].

## 3.1 General description of correlated RB

The ultimate goal of correlated RB is to provide a tool for better characterization of crosstalk errors. More concretely, the method aims to measure and classify the average noise in the system, based on weight and locality features. To introduce these concepts, it will be instructive to think in terms of coherent errors: Errors that are described by unitary error channels. For example, we can think of ideal spin qubits that do not interact with each other, unless a pulse implementing a gate is applied. For example, we can picture this pulse as a unitary time evolution operator, $U = e^{-i(\Delta t)/\hbar H_{int}}$, coupling two target spins through a spin-spin interaction Hamiltonian and effectively leading to the execution a particular two-qubit gate operation [33]. However, we can also imagine a less ideal situation in which the spin qubits retain residual interactions with nearest neighbouring spins. This would mean that even if the pulse was implemented perfectly, the result would no longer be an operation that only changes the state of the target spins. Instead, it would also have effects on the state of nearby qubits. Conversely, we can think of ideal spin qubits, but faulty pulses, where the implemented interaction Hamiltonian actually leads to the coupling between a higher number of spins. In all these situations, knowing the number of unintended qubits affected (i.e. weight) and the extent to which this interaction is felt beyond the nearest neighbour qubits (i.e. locality) become crucial diagnostic information to characterize and hopefully mitigate crosstalk errors in the system. The question then becomes how to extract this information.

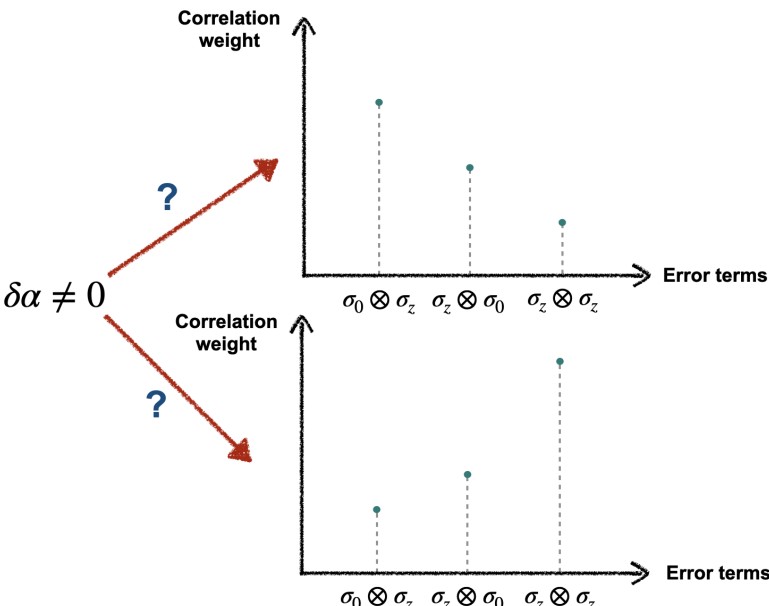

Figure 7: The figure depicts an illustrative example of the difficulty in inferring from $\delta\alpha$ whether correlations are arising due to inadvertently driving the idle qubit ($\sigma_0 \otimes \sigma_z$ or $\sigma_z \otimes \sigma_0$), or due to qubit-qubit interactions ($\sigma_z \otimes \sigma_z$ term).

Let us imagine the simultaneous RB circuit, using the gate-set from the group $G = C_1 \times C_1$. To simplify the discussion, let us constrain the length of this circuit to a single random operation from $G$. Now suppose that instead of the perfect implementation of these gates, their action was plagued by crosstalk errors. If our input state is the tensor product state $|\rho_0 \otimes \rho_0\rangle\rangle$ (with $\rho_0 = |0\rangle\langle0|$), then from the previous discussion it is clear that the output state, right before measurement, can no longer be expressed as another product state[15] [32]. To make this more transparent, consider a measurement with $E = Z \otimes Z$. Following the same notation as in last chapter, the effects of the crosstalk errors will be encoded by the error channel $\Lambda$. After a single random Clifford in $G$, our sequence fidelity (see Eq.(44)) reads:

$$\langle\langle Z \otimes Z|\hat{\Lambda}\hat{\Lambda}_T|\rho_0 \otimes \rho_0\rangle\rangle \underset{|\tilde{\rho}\rangle\rangle = \hat{\Lambda}\hat{\Lambda}_T|\rho_0 \otimes \rho_0\rangle\rangle}{=} \langle\langle Z \otimes Z|\tilde{\rho}\rangle\rangle = \text{Tr}\big(\tilde{\rho}\, Z \otimes Z\big) = \langle Z \otimes Z\rangle, \qquad (66)$$

where the last equality just follows from the definition of expectation value of an observable in quantum mechanics [16]. Note that to simplify the discussion, we have neglected the effect of SPAM errors. The observable $\langle Z \otimes Z\rangle$ will depend on the state of the two-qubits in a non-trivial way, since crosstalk errors induce couplings between qubit states. Suppose instead that the errors in the system preserve locality and independence of operations. For example, the pulse that implements two-qubit gates is faulty, but only results in an imperfect version of the gate on the target qubits, without changing the state of other qubits in the system, or without introducing unintended inter-dependencies in the states of the target qubits. In this case, the same circuit run would yield a product state, and we could express our observable $\langle Z \otimes Z\rangle$ as:

$$\langle Z \otimes Z\rangle = \text{Tr}\big(\tilde{\rho}\, Z \otimes Z\big) = \text{Tr}\big((\tilde{\rho}_1 \otimes \tilde{\rho}_2)\, Z \otimes Z\big) = \text{Tr}\big(\tilde{\rho}_1 Z\big) \text{Tr}\big(\tilde{\rho}_2 Z\big) = \langle Z_1\rangle\langle Z_2\rangle, \qquad (67)$$

where to distinguish the expectation value of the Pauli-$Z$ on each qubit, we label it accordingly. The observable $\langle Z \otimes Z\rangle$ is now a product of two independent expectation values. This factorization was impossible to attain in Eq.(66). Indeed, we could devise the following strategy to flag the presence of crosstalk errors:

$$\langle Z \otimes Z\rangle - \langle Z_1\rangle\langle Z_2\rangle = \begin{cases} 0, & \text{no crosstalk errors,} \\ \neq 0, & \text{crosstalk errors are present.} \end{cases} \qquad (68)$$

The right-hand side of Eq.(68) bares very close resemblance with the definition of covariance between two random variables X and Y:

$$\text{Cov}(X, Y) = \langle XY\rangle - \langle X\rangle\langle Y\rangle. \qquad (69)$$

The covariance measures the linear association between random variables [34], and is zero if $X$ and $Y$ are independent. Hence, like $\langle XY\rangle$, $\langle Z \otimes Z\rangle$ represents a correlator (sometimes also called correlation function), and contains important information for diagnosing crosstalk. Correlated RB uses this feature, but goes beyond the simple estimation of correlators.

Just like the methods presented so far, correlated RB makes use of the Clifford gate-set. It essentially works by borrowing experiment 3 of simultaneous RB, and appending further post-processing steps to it. The system will then be partitioned into different subsystems, each containing a sub-set of qubits. Note that the number of qubits in each subsystem does not need to be the same. For a 3−qubit system, we could consider any of the following partitions: 3−subsystems, with one qubit each; 2−subsystems, where subsystem one contains one qubit and subsystem two contains two qubits, or the reverse. The partition can then be tailored to the specific needs of the problem. The next step is the simultaneous implementation of

---

[15]This is equivalent to saying that the error channel is not a tensor product of single-qubit operations, i.e. $\Lambda \neq \Lambda_1 \otimes \Lambda_2$.

standard RB to each of the subsystems. This stage entails gathering enough data to estimate observables like the averaged sequence fidelity. Recall that this quantity is simply measuring the probability of retrieving the initial state as the final output of the circuit, and depends on the sequence length ($m$) of applied Cliffords. As argued here, in correlated RB the shift changes from directly measuring the fidelity to focusing on measuring Pauli correlators, $\langle \hat{P}_i \rangle$. These two quantities are, nevertheless, intimately connected. In the end stage of the RB protocol, the input state $\rho$ is transformed into a new state, call it $\rho'$, by the composition of $m$ twirling channels. At the same time, the expectation value of any operator, if measured at the end of the RB circuit, will be given by:

$$\langle \hat{P}_i \rangle = \text{Tr}\left( \rho' \hat{P}_i \right) = \langle\langle P_i | \rho' \rangle\rangle , \tag{70}$$

with $\rho' = \Lambda_T^{\circ m} \rho$.[16] As we have seen, simultaneous RB leads to a decomposition of the twirling channel in terms of projectors onto the different irreducible subspaces. Each of these projectors is built from the set of Pauli matrices that are mapped into one another other by the action of the direct product Clifford group. For example, for a 2−qubit system, partitioned into two 1−qubit systems, the sub-set of Pauli matrices $\{\sigma_0 \otimes \sigma_x, \sigma_0 \otimes \sigma_y, \sigma_0 \otimes \sigma_z\}$ was always mapped onto itself by the action of all the elements of the group $\mathcal{C}_1 \times \mathcal{C}_1$; for this reason, it provided a basis for one of the subspace projectors. Furthermore, given that $\hat{P}_i$ is a Pauli operator, it will necessarily belong to one of the invariant subspaces. All these ingredients result in an expectation value that, in the Liouville representation, has the simplified form:

$$\langle \hat{P}_i \rangle = \langle\langle P_i | \hat{\Lambda}_T^m | \rho \rangle\rangle = \sum_S \langle\langle P_i | \alpha_S^m \mathbb{P}_S | \rho \rangle\rangle$$
$$= \sum_S \alpha_S^m \langle\langle P_i | \rho_S \rangle\rangle = \alpha_{S_i}^m \langle\langle P_i | \rho_{S_i} \rangle\rangle \tag{71}$$
$$\Leftrightarrow \quad \boxed{\langle \hat{P}_i \rangle = \alpha_{S_i}^m \gamma_i ,}$$

where $\sum_S$ denotes the sum over invariant subspaces. Thus, just as the sequence fidelity, these correlators will manifest an exponential decay, as a function of the sequence length. Unlike the average sequence fidelity, each correlator is expressed in terms of a single exponential decay. It is, however, sensitive to measurement errors.

By fitting the data to Eq.(71), one can estimate the value of the $\alpha$ parameters. If we were to continue following the simultaneous RB scheme, an extra step would be required to quantify the magnitude of correlations in the system. This would involve comparing the $\alpha$ values, in a similar fashion as in Eq.(41). It is on this step that correlated RB most significantly departures from the simultaneous RB method. In Ref. [5], a new parametrization scheme is proposed, in which the twirling map is re-written as a composition of *fixed-subspace weight error channels*. More concretely,

$$\Lambda_T = \bigcirc_S \Lambda_S ,$$
$$\Lambda_S(\rho) = (1 - \epsilon_S)\rho + \frac{\epsilon_S}{\dim(\mathbb{P}_S) + 1} \left( \rho + \sum_{P \in S} P \rho P^\dagger \right) , \tag{72}$$

and basically translates to re-expressing the original twirling channel as a composition of maps, each *living* on a unique invariant subspace. The reason why this aids in classifying crosstalk,

---

[16]We have argued that the average sequence operator is given by Eq.(25). In Liouville representation, this would suggest the expectation value $\langle \hat{P}_i \rangle$ to be written as $\langle \hat{P}_i \rangle = \langle\langle P_i | \hat{\Lambda} \hat{\Lambda}_T^m | \rho \rangle\rangle$. As was done previously, we could absorb the final error channel into the Pauli operator, and the quantity $\langle\langle \tilde{P}_i | \hat{\Lambda}_T^m | \rho \rangle\rangle$ could be interpreted as evaluating the desired expectation value in the presence of measurement noise. In this sense, the method developed here is not SPAM robust, and a more general approach would be to combine it with a framework like character RB [30].

in terms of weight and (geometric) locality, is because these attributes are already known for each of the invariant sectors. For example, let us again return to the case of a 2−qubit system. Suppose that, after employing this scheme, we would learn that the largest $\epsilon$ coefficient was the one related to the invariant subspace generated by the Paulis $\{\boldsymbol{\sigma} \otimes \boldsymbol{\sigma}'\}$, with $\boldsymbol{\sigma}, \boldsymbol{\sigma}' \in \{\sigma_x, \sigma_y, \sigma_z\}$. This would then imply that weight two terms were dominating in the error map, and were probably the most significant source of crosstalk. In order to determine the coefficients $\epsilon$, one needs to invert Eq.(72), and find an algebraic expression relating the set of $\{\alpha\}$ parameters with the unknown $\{\epsilon\}$ set. Hence, correlated RB can be summarized as follows:

1. Perform simultaneous RB on all the predefined subsystems. This stage includes collecting sufficient data on all relevant Pauli correlators, as a function of the circuit's length $m$. At the end of this step, one should be able to estimate the set of parameters $\{\alpha\}$.

2. From the knowledge of $\{\alpha\}$, determine the unknown $\{\epsilon\}$ parameters.

Let us point out that Eq.(72) assumes that it is always possible to express the twirling map as a composition of fixed-subspace weight error channels. Indeed, this is not a given, and Ref. [5] also provides a toy model as counter example. In the next section, we will discuss how to relate the two sets of parameters, $\{\alpha\}$ and $\{\epsilon\}$. As a main takeaway, we note that Eq.(72) is central to the correlated RB protocol, as it establishes the relationship between the $\{\epsilon\}$ parameters and the characterization of the locality and weight of crosstalk errors.

## 3.2 How to relate the $\{\alpha\}$ and $\{\epsilon\}$ parameters

We start by writing down the Pauli transfer matrix, corresponding to the fixed-subspace weight error channel:

$$
\begin{aligned}
\left(\hat{\mathcal{R}}_{\Lambda_S}\right)_{i,j} &= \mathrm{Tr}\left(\sigma_i^\dagger \Lambda_S(\sigma_j)\right) \\
&= \mathrm{Tr}\left[(1-\epsilon_S)\hat{\sigma}_i^\dagger \hat{\sigma}_j + \frac{\epsilon_S}{\dim(\mathbb{P}_S)+1}\left(\hat{\sigma}_i^\dagger \hat{\sigma}_j + \sum_{\sigma \in S}\hat{\sigma}_i^\dagger \hat{\sigma}\hat{\sigma}_j\hat{\sigma}^\dagger\right)\right] \\
&= (1-\epsilon_S)\delta_{i,j} + \frac{\epsilon_S}{\dim(\mathbb{P}_S)+1}\delta_{i,j} + \frac{\epsilon_S}{\dim(\mathbb{P}_S)+1}\sum_{\sigma \in S}\mathrm{Tr}\left(\hat{\sigma}_i^\dagger \hat{\sigma}\hat{\sigma}_j\hat{\sigma}^\dagger\right) \\
&= \delta_{i,j}\left((1-\epsilon_S)+\frac{\epsilon_S}{\dim(\mathbb{P}_S)+1}\right) + \frac{\epsilon_S}{\dim(\mathbb{P}_S)+1}\sum_{\sigma \in S}(-1)^{\langle\sigma,\sigma_j\rangle}\mathrm{Tr}\left(\hat{\sigma}_i^\dagger \hat{\sigma}_j\right) \\
&= \delta_{i,j}\left((1-\epsilon_S)+\frac{\epsilon_S}{\dim(\mathbb{P}_S)+1}+\frac{\epsilon_S}{\dim(\mathbb{P}_S)+1}\sum_{\sigma \in S}(-1)^{\langle\sigma,\sigma_i\rangle}\right) \\
&= \delta_{i,j}\left[(1-\epsilon_S)+\frac{\epsilon_S}{\dim(\mathbb{P}_S)+1}\left(1+\sum_{\sigma \in S}(-1)^{\langle\sigma,\sigma_i\rangle}\right)\right]
\end{aligned}
\tag{73}
$$

$$
\Leftrightarrow \quad \boxed{\left(\hat{\mathcal{R}}_{\Lambda_S}\right)_{i,j} = \delta_{i,j}\left((1-\epsilon_S)+\epsilon_S\left(R_{\mathrm{depol}}\right)_{i,i}\right).}
$$

Note that we are using the notation whereby $\langle\sigma,\sigma_i\rangle = 0$, if the two commute, or $\langle\sigma,\sigma_i\rangle = 1$, if they anti-commute. A channel composition becomes a matrix product in the Liouville rep-

resentation, which means that Eq.(72) translates to the product of $\hat{\mathcal{R}}_{\Lambda_S}$ matrices:

$$\Lambda_T = \bigcirc_S \Lambda_S \rightarrow \hat{\mathcal{R}}_{\Lambda_T} = \prod_S \hat{\mathcal{R}}_{\Lambda_S}, \tag{74}$$

and since both $\hat{\mathcal{R}}_{\Lambda_T}$ and all the $\hat{\mathcal{R}}_{\Lambda_S}$ matrices are diagonal, the equality becomes an element-wise statement, which allows us to express each $\alpha_i$ coefficient in $\hat{\mathcal{R}}_{\Lambda_T}$ in terms of the $\{\epsilon\}$ parameters as

$$\boxed{\alpha_i = \prod_S \left[ 1 + \epsilon_S \left( \left( R_{\text{depol}} \right)_{i,i} - 1 \right) \right].} \tag{75}$$

Then by inverting Eq.(75), we can determine the $\{\epsilon\}$ parameters, from the set of measured values $\{\alpha\}$. Therefore, Eq.(75) is central to the correlated RB protocol as it provides a practical way to estimate the $\{\epsilon\}$ parameters.

As a concrete example, let us consider the familiar example of a 2–qubit system, partitioned into two 1–qubit subsystems. We already know the resulting form of the twirling Pauli transfer matrix (see Eq.(37)). We will then explicitly compute $\hat{\mathcal{R}}_{\Lambda_S}$ for each invariant subspace. Recall that, in this case, there are four invariant subspaces: $W_0 = \text{Span}\{|\sigma_0 \otimes \sigma_0\rangle\rangle\}$, $W_1 = \text{Span}\{|\sigma_0 \otimes \boldsymbol{\sigma}\rangle\rangle\}$, $W_2 = \text{Span}\{|\boldsymbol{\sigma} \otimes \sigma_0\rangle\rangle\}$ and $W_3 = \text{Span}\{|\boldsymbol{\sigma} \otimes \boldsymbol{\sigma}\rangle\rangle\}$. This then leads to four distinct scenarios for the matrix elements $\left( R_{\text{depol}} \right)_{i,i}$:

$$S = W_0 \quad \Longrightarrow \quad \left( R_{\text{depol}} \right)_{i,i} = \frac{1}{2}\left( 1 + \sum_{\sigma \in W_0} (-1)^{\langle \sigma, \sigma_i \rangle} \right) = 1,$$

$$S = W_1 \quad \Longrightarrow \quad \left( R_{\text{depol}} \right)_{i,i} = \frac{1}{4}\left( 1 + \sum_{\sigma \in W_1} (-1)^{\langle \sigma, \sigma_i \rangle} \right) = \begin{cases} 1, & \sigma_i \in W_0, \\ 0, & \sigma_i \in W_1, \\ 1, & \sigma_i \in W_2, \\ 0, & \sigma_i \in W_3, \end{cases}$$

$$S = W_2 \quad \Longrightarrow \quad \left( R_{\text{depol}} \right)_{i,i} = \frac{1}{4}\left( 1 + \sum_{\sigma \in W_2} (-1)^{\langle \sigma, \sigma_i \rangle} \right) = \begin{cases} 1, & \sigma_i \in W_0, \\ 1, & \sigma_i \in W_1, \\ 0, & \sigma_i \in W_2, \\ 0, & \sigma_i \in W_3, \end{cases} \tag{76}$$

$$S = W_3 \quad \Longrightarrow \quad \left( R_{\text{depol}} \right)_{i,i} = \frac{1}{10}\left( 1 + \sum_{\sigma \in W_3} (-1)^{\langle \sigma, \sigma_i \rangle} \right) = \begin{cases} 1, & \sigma_i \in W_0, \\ -\frac{1}{5}, & \sigma_i \in W_1, \\ -\frac{1}{5}, & \sigma_i \in W_2, \\ \frac{1}{5}, & \sigma_i \in W_3. \end{cases}$$

These results arise because of the commutation relations between the Pauli elements, namely

- $W_0$ only contains the identity, which commutes with every element.

- Every element in $W_1$ commutes with any given element in $W_2$, and vice-versa:

$$(\hat{\sigma}_0 \otimes \hat{\sigma}_i)(\hat{\sigma}_j \otimes \hat{\sigma}_0) = (\hat{\sigma}_j \otimes \hat{\sigma}_i),$$
$$(\hat{\sigma}_j \otimes \hat{\sigma}_0)(\hat{\sigma}_0 \otimes \hat{\sigma}_i) = (\hat{\sigma}_j \otimes \hat{\sigma}_i).$$

- Pick any two elements from $W_1$, they will only commute if they are the same. In every other case, these elements anti-commute. Since there are 3 elements in $W_1$, only one term contributes positively to the sum in $\left( R_{\text{depol}} \right)_{i,i}$, yielding $\sum_{\sigma \in W_1} (-1)^{\langle \sigma, \sigma_i \rangle} = -1$ ($\sigma_i \in W_1$). The same for any two elements belonging to $W_2$.

- Choose an element in $W_2$ and consider its commutation relations with the elements in $W_3$,

$$(\hat{\sigma}_i \otimes \hat{\sigma}_0)(\hat{\sigma}_j \otimes \hat{\sigma}_l) = (\hat{\sigma}_i \hat{\sigma}_j \otimes \hat{\sigma}_l),$$
$$(\hat{\sigma}_j \otimes \hat{\sigma}_l)(\hat{\sigma}_i \otimes \hat{\sigma}_0) = (\hat{\sigma}_j \hat{\sigma}_i \otimes \hat{\sigma}_l).$$

These elements only commute in the case where $\hat{\sigma}_i = \hat{\sigma}_j$. Hence, for any given element belonging to $W_2$, there are 3 elements in $W_3$ commuting with it. There are 9 elements in $W_3$, which leads to $\sum_{\sigma \in W_3}(-1)^{\langle \sigma, \sigma_i \rangle} = 3 - 6 = -3$ ($\sigma_i \in W_2$). The same conclusion holds if we replace $W_2$ with $W_1$.

- For any given element in $W_3$, there is only one element belonging to $W_2$ with which it commutes. The same is true with elements in $W_1$. This results in $\sum_{\sigma \in W_2}(-1)^{\langle \sigma, \sigma_i \rangle} = -1$ ($\sigma_i \in W_3$) and $\sum_{\sigma \in W_1}(-1)^{\langle \sigma, \sigma_i \rangle} = 1 - 1 - 1 = -1$ ($\sigma_i \in W_3$).

- Two elements in $W_3$ only commute if,

$$(\hat{\sigma}_j \otimes \hat{\sigma}_l)(\hat{\sigma}_i \otimes \hat{\sigma}_k) = (\hat{\sigma}_j \hat{\sigma}_i \otimes \hat{\sigma}_l \hat{\sigma}_k),$$
$$(\hat{\sigma}_i \otimes \hat{\sigma}_k)(\hat{\sigma}_j \otimes \hat{\sigma}_l) = (\hat{\sigma}_i \hat{\sigma}_j \otimes \hat{\sigma}_k \hat{\sigma}_l),$$

they are the same, or if $\hat{\sigma}_i \neq \hat{\sigma}_j$ and $\hat{\sigma}_l \neq \hat{\sigma}_k$. For any fixed element in $W_3$, there are then five elements with each it commutes. The resulting sum yields $\sum_{\sigma \in W_3}(-1)^{\langle \sigma, \sigma_i \rangle} = 5 - 4 = 1$ ($\sigma_i \in W_3$).

Using then Eq.(76), it follows that each $\hat{\mathcal{R}}_{\Lambda_S}$ is given as

$$\hat{\mathcal{R}}_{\Lambda_{W_0}} = \mathbb{1},$$

$$\hat{\mathcal{R}}_{\Lambda_{W_1}} = \begin{pmatrix} 1 & 0 & 0 & 0 \\ 0 & (1-\epsilon_1)\mathbb{1}_{W_1} & 0 & 0 \\ 0 & 0 & \mathbb{1}_{W_2} & 0 \\ 0 & 0 & 0 & (1-\epsilon_1)\mathbb{1}_{W_3} \end{pmatrix},$$

$$\hat{\mathcal{R}}_{\Lambda_{W_2}} = \begin{pmatrix} 1 & 0 & 0 & 0 \\ 0 & \mathbb{1}_{W_1} & 0 & 0 \\ 0 & 0 & (1-\epsilon_2)\mathbb{1}_{W_2} & 0 \\ 0 & 0 & 0 & (1-\epsilon_2)\mathbb{1}_{W_3} \end{pmatrix}, \tag{77}$$

$$\hat{\mathcal{R}}_{\Lambda_{W_3}} = \begin{pmatrix} 1 & 0 & 0 & 0 \\ 0 & \left(1-\frac{6\epsilon_3}{5}\right)\mathbb{1}_{W_1} & 0 & 0 \\ 0 & 0 & \left(1-\frac{6\epsilon_3}{5}\right)\mathbb{1}_{W_2} & 0 \\ 0 & 0 & 0 & \left(1-\frac{4\epsilon_3}{5}\right)\mathbb{1}_{W_3} \end{pmatrix},$$

where we have defined $\mathbb{1}_{W_i}$ to be the identity matrix on the reduced subspace, i.e. the identity matrix with dimensions $\dim(\mathbb{P}_i) \times \dim(\mathbb{P}_i)$. The diagonal structure of the matrices $\hat{\mathcal{R}}_{\Lambda_{W_i}}$ allows

us to write:

$$
\hat{\mathcal{R}}_{\Lambda_T} = \hat{\mathcal{R}}_{\Lambda_{W_0}} \hat{\mathcal{R}}_{\Lambda_{W_1}} \hat{\mathcal{R}}_{\Lambda_{W_2}} \hat{\mathcal{R}}_{\Lambda_{W_3}}
$$

$$
\Leftrightarrow
\begin{pmatrix}
1 & 0 & 0 & 0 \\
0 & \alpha_1 \mathbb{1}_{W_1} & 0 & 0 \\
0 & 0 & \alpha_2 \mathbb{1}_{W_2} & 0 \\
0 & 0 & 0 & \alpha_3 \mathbb{1}_{W_3}
\end{pmatrix}
$$

$$
=
\begin{pmatrix}
1 & 0 & 0 & 0 \\
0 & (1-\epsilon_1)\left(1-\frac{6\epsilon_3}{5}\right)\mathbb{1}_{W_1} & 0 & 0 \\
0 & 0 & (1-\epsilon_2)\left(1-\frac{6\epsilon_3}{5}\right)\mathbb{1}_{W_2} & 0 \\
0 & 0 & 0 & (1-\epsilon_1)(1-\epsilon_2)\left(1-\frac{4\epsilon_3}{5}\right)\mathbb{1}_{W_3}
\end{pmatrix}.
$$

From the resulting set of equations, we can begin the task of retrieving the set of $\{\epsilon\}$ parameters. Assuming that all $\alpha_i > 0$,

$$
\underbrace{\frac{\alpha_1}{\alpha_2}}_{:=\gamma_\alpha} = \frac{(1-\epsilon_1)}{(1-\epsilon_2)} \qquad \Leftrightarrow \qquad \epsilon_1 = 1 - \gamma_\alpha(1-\epsilon_2),
$$

$$
1 - \frac{6}{5}\epsilon_3 = \frac{\alpha_2}{1-\epsilon_2} \qquad \Leftrightarrow \qquad 1 - \frac{4}{5}\epsilon_3 = \frac{1}{3} + \frac{2\alpha_2}{3(1-\epsilon_2)},
$$

$$
\alpha_3 = \gamma_\alpha(1-\epsilon_2)^2\left(\frac{1}{3} + \frac{2\alpha_2}{3(1-\epsilon_2)}\right) \qquad \Leftrightarrow \qquad 3\alpha_3 = \gamma_\alpha(1-\epsilon_2)^2 + 2\alpha_2\gamma_\alpha\underbrace{(1-\epsilon_2)}_{:=y_2} \qquad (78)
$$

$$
\Leftrightarrow \qquad y_2^2 + 2\alpha_2 y_2 - 3\frac{\alpha_3}{\gamma_\alpha} = 0
$$

$$
\Leftrightarrow \qquad y_2 = -\alpha_2 \pm \sqrt{\alpha_2^2 + 3\frac{\alpha_3}{\gamma_\alpha}},
$$

$$
\boxed{\epsilon_2 = 1 + \alpha_2 \mp \sqrt{\alpha_2^2 + 3\frac{\alpha_3}{\gamma_\alpha}}, \qquad \epsilon_3 = \frac{5}{6}\left(1 - \frac{\alpha_2}{(1-\epsilon_2)}\right), \qquad \epsilon_1 = 1 - \frac{\alpha_1}{\alpha_2}(1-\epsilon_2).}
$$

So far, we have always been operating under the assumption that all the linear maps are CPTP.[17] In order for this property to also hold true for the fixed-subspace weight error channels, some constraints on the $\{\epsilon\}$ parameters arise:

**(Positivity condition)**: $\langle\phi|\Lambda_S(\rho)|\phi\rangle \geq 0, \qquad \forall \ \{|\phi\rangle\}$,

$$
(1-\epsilon_S)\langle\phi|\rho|\phi\rangle + \frac{\epsilon_S}{\dim(\mathbb{P}_S)+1}\left(\langle\phi|\rho|\phi\rangle + \sum_{P\in S}\underbrace{\langle\phi|P}_{:=\langle\varphi_P|}\rho\underbrace{P^\dagger|\phi\rangle}_{:=|\varphi_P\rangle}\right) \geq 0,
$$

$$
\underbrace{\langle\phi|\rho|\phi\rangle}_{\geq 0}\left((1-\epsilon_S) + \frac{\epsilon_S}{\dim(\mathbb{P}_S)+1}\right) + \frac{\epsilon_S}{\dim(\mathbb{P}_S)+1}\underbrace{\sum_{P\in S}\langle\varphi_P|\rho|\varphi_P\rangle}_{\geq 0} \geq 0
$$

$$
\implies \quad (1-\epsilon_S) + \frac{\epsilon_S}{\dim(\mathbb{P}_S)+1} \geq 0, \quad \text{and} \quad \epsilon_S \geq 0
$$

$$
\Leftrightarrow \quad \boxed{0 \leq \epsilon_S \leq \frac{\dim(\mathbb{P}_S)+1}{\dim(\mathbb{P}_S)},} \qquad (79)
$$

---

[17]CPTP map = Completely Positive and Trace preserving map.

which for the previous example means $\epsilon_1, \epsilon_2 \in [0, \frac{4}{3}]$ and $\epsilon_3 \in [0, \frac{10}{9}]$. The trace condition is automatically satisfied, adding no further constraints,

**(Trace condition)**: $\text{Tr}(\Lambda_S(\rho)) = 1$,

$$(1-\epsilon_S)\text{Tr}(\rho) + \frac{\epsilon_S}{\dim(\mathbb{P}_S)+1}\left(\text{Tr}(\rho) + \underbrace{\sum_{P\in S}\text{Tr}(P\rho P^\dagger)}_{=\,\dim(\mathbb{P}_S)}\right) = 1,$$

$$1 + \frac{\epsilon_S}{\dim(\mathbb{P}_S)+1}\left(-\dim(\mathbb{P}_S) - 1 + 1 + \dim(\mathbb{P}_S)\right) = 1.$$

(80)

We see that the condition in Eq.(79) might lead to cases where it is not possible to find a parameterization of the twirling map in terms of CPTP fixed-subspace weight channels. In those cases, correlated RB cannot offer any further insight beyond what simultaneous RB already provides.

As a final remark, let us return to the 2–qubit example, and examine what happens to the $\epsilon$ parameters in the presence of uncorrelated errors. Recall that this scenario requires $\alpha_3 = \alpha_1\alpha_2$. This implies:

$$(1-\epsilon_1)(1-\epsilon_2)\left(1-\frac{4\epsilon_3}{5}\right) = (1-\epsilon_1)(1-\epsilon_2)\left(1-\frac{6\epsilon_3}{5}\right)^2$$

$$\Leftrightarrow \quad \left(1-\frac{4\epsilon_3}{5}\right) = \left(1-\frac{6\epsilon_3}{5}\right)^2$$

$$\Leftrightarrow \quad \epsilon_3 = 0 \quad \vee \quad \epsilon_3 = \frac{10}{9}.$$

(81)

The solution $\epsilon_3 = 0$ has a clear interpretation. Recall that when the errors are uncorrelated, the matrix elements of $\hat{\mathcal{R}}_{\Lambda_T}$ acquire a simplified form,

$$\left(\hat{\mathcal{R}}_{\Lambda_T}\right)_{ij,kl} = \text{Tr}\left(\left(\sigma_i^\dagger \otimes \sigma_j^\dagger\right)\underbrace{\Lambda}_{=\Lambda^{(1)}\otimes\Lambda^{(2)}}(\sigma_k \otimes \sigma_l)\right)$$

$$= \text{Tr}\left(\left(\sigma_i^\dagger \otimes \sigma_j^\dagger\right)\left(\Lambda^{(1)}(\sigma_k) \otimes \Lambda^{(2)}(\sigma_l)\right)\right)$$

$$= \text{Tr}\left(\left(\sigma_i^\dagger\right)\Lambda^{(1)}(\sigma_k)\right)\text{Tr}\left(\left(\sigma_j^\dagger\right)\Lambda^{(2)}(\sigma_l)\right)$$

$$= \left(\hat{\mathcal{R}}_{\Lambda_T^{(1)}}\right)_{i,k}\left(\hat{\mathcal{R}}_{\Lambda_T^{(2)}}\right)_{j,l},$$

where we denoted $\Lambda^{(i)}$ the error map in subsystem $i$. We then see that $\hat{\mathcal{R}}_{\Lambda_T} = \hat{\mathcal{R}}_{\Lambda_T^{(1)}} \otimes \hat{\mathcal{R}}_{\Lambda_T^{(2)}}$. The solution $\epsilon_3 = 0$ allows us to similarly express $\hat{\mathcal{R}}_{\Lambda_T}$ as the tensor product of the weight-1 error channels, in the 1–qubit basis

$$\hat{\mathcal{R}}_{\Lambda_{W_i}}^{(1\text{-qubit})} = |\sigma_0\rangle\rangle\langle\langle\sigma_0| + (1-\epsilon_i)\sum_i |\sigma_i\rangle\rangle\langle\langle\sigma_i|,$$

$$\hat{\mathcal{R}}_{\Lambda_{W_2}}^{(1\text{-qubit})} \otimes \hat{\mathcal{R}}_{\Lambda_{W_1}}^{(1\text{-qubit})} = |\sigma_0\otimes\sigma_0\rangle\rangle\langle\langle\sigma_0\otimes\sigma_0| + (1-\epsilon_1)\sum_i |\sigma_0\otimes\sigma_i\rangle\rangle\langle\langle\sigma_0\otimes\sigma_i|$$

$$+ (1-\epsilon_2)\sum_i |\sigma_i\otimes\sigma_0\rangle\rangle\langle\langle\sigma_i\otimes\sigma_0|$$

$$+ (1-\epsilon_1)(1-\epsilon_2)\sum_{i,j} |\sigma_i\otimes\sigma_j\rangle\rangle\langle\langle\sigma_i\otimes\sigma_j|,$$

(82)

$$\hat{\mathcal{R}}_{\Lambda_T} = \hat{\mathcal{R}}_{\Lambda_{W_2}}^{(1\text{-qubit})} \otimes \hat{\mathcal{R}}_{\Lambda_{W_1}}^{(1\text{-qubit})},$$

with $\alpha_1 = (1-\epsilon_1)$ and $\alpha_2 = (1-\epsilon_2)$. This clarifies that the twirling channel has no correlations, since it can be expressed in terms of disjoint and uncoupled channels, each with support on a unique subsystem. Thus, reading $\epsilon_3 = 0$ from an experiment in a 2−qubit system allows us to automatically conclude on the absence of correlated noise. Note that, in general, even if the twirling map admits a parametrization in terms of fixed-subspace-weight channels, it will differ from the tensor product of independent error maps. In that case, the extra $\epsilon$ parameters introduce interactions between subsystems, with the weight and locality features specified by the corresponding subspaces.

## 3.3 Discussion

Correlated benchmarking shares many similarities with simultaneous RB, but differs from the latter in that it allows a more direct characterization of crosstalk errors. Attaining this goal requires measuring a series of correlators, of the form given in Eq.(71). If we partition our system into $K$ subsystems, the number of required Pauli correlators to measure scales as $2^K$ [5]. This number corresponds to the total number of invariant subspaces. Since each of these has its own $\alpha$ parameter, we need to measure at least one correlator per invariant subspace. Without the set of all $\{\alpha\}$, solving for the $\{\epsilon\}$ parameters cannot be carried through. Furthermore, there might be cases where the mapping to the $\{\epsilon\}$ set might not be possible, in which case correlated RB will not yield any further insight compared to simultaneous RB [5]. Given the shared steps between correlated and simultaneous RB protocols, both also suffer from the same type of drawbacks. While some of these might be circumvented, the exponential growth (with system size) of the number of subsystems required to evaluate is a persistent problem. However, for characterization of low-weight crosstalk errors, correlated RB is a favourable choice, since it provides further insights on the errors, when compared to simultaneous RB.

## 3.4 Key results in chapter 3

Here we summarize the main takeaways for the correlated RB protocol.

- The aim of correlated RB is to extract information on the weight and locality of crosstalk errors.

- To extract weight and locality information, correlated RB uses a new parameterization of the twirled error channel as a composition of error channels belonging to different irreducible subspaces. Since these different subspaces have distinct weight and locality features, comparing the parameters in different irreducible subspaces allows us to identify where the dominant crosstalk errors lie. Recall from Chapter 1 that the twirling channel preserves average fidelity, so this analysis allows us to identify the most dominant contributions to crosstalk, on average.

- The individual channels contributing to the new parametrization of the twirled error channel are called fixed-subspace-weight channels. The relationship between the original twirling channel and the fixed-subspace-weight channels is defined as:

$$\Lambda_T = \bigcirc_S \Lambda_S \,,$$

$$\Lambda_S(\rho) = (1-\epsilon_S)\rho + \frac{\epsilon_S}{\dim(\mathbb{P}_S)+1}\left(\rho + \sum_{P \in S} P\rho P^\dagger\right).$$

$\Lambda_T$ is the original twirled error channel and each $\Lambda_S$ is a fixed-subspace-weight channel. The different $\Lambda_S$ are expressed in terms of Pauli operators that span

different irreducible subspaces.

- The experimental implementation of correlated RB is similar to the third experiment in simultaneous RB, except for the choice of POVM elements. These are now given by Pauli operators, belonging to different irreducible subspaces. If our qubit system has been decomposed into k subsystems, then we have to choose $2^k$ different Pauli operations from different irreducible subspaces.

- Using the modified third experiment of simultaneous RB allows us to estimate the different $\alpha_i$ parameters. The key insight of correlated RB is then to use these estimates to extract the set of $\epsilon_S$ parameters. This is done by inverting the equation:

$$\alpha_i = \prod_S \left[ 1 + \epsilon_S \left( \left( R_{\text{depol}} \right)_{i,i} - 1 \right) \right],$$

$$\left( R_{\text{depol}} \right)_{i,i} = \frac{1}{\dim(\mathbb{P}_S) + 1} \left( 1 + \sum_{\sigma \in S} (-1)^{\langle \sigma, \sigma_i \rangle} \right),$$

where $\dim(\mathbb{P}_S)$ denotes the dimension of the corresponding irreducible subspace, $\sigma$ belongs to the set of normalized Pauli matrices spanning that particular irreducible subspace, and where $\langle \sigma, \sigma_i \rangle = 0$, if the two normalized Paulis commute, or $\langle \sigma, \sigma_i \rangle = 1$, if they anti-commute.

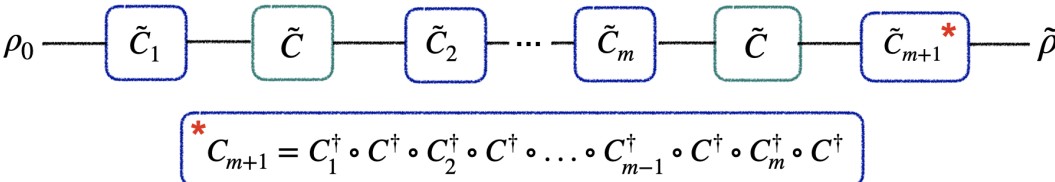

Figure 8: The goal of interleaved RB is to benchmark a specific gate (here denoted by $\tilde{C}$). This protocol requires us to build new sequences, where the gate we wish to characterize is interleaved in between randomly selected gates. We denote by $\tilde{C}$ the noisy implementation of the ideal gate $C$.

# 4 Interleaved randomized benchmarking in a nutshell

The accompanying notebook for this section can be found at: https://gitlab.com/QMAI/papers/rb-tutorial/-/blob/main/Interleaved%20RB/InterleavedRB.ipynb.

In this section we give an overview of the interleaved randomized benchmarking protocol, introduced in Ref. [6]. The essence of this method is to build an estimation of the error rate associated to a particular gate of interest. In practice, it is often the case that benchmarking a specific gate is more relevant than just estimating the average error over the whole gate set [8]. We will see how modifying the standard RB sequences for interleaved sequences (see fig. 8) will be an important strategy for providing us with an error metric for the desired benchmark gate.

## 4.1 General description of interleaved RB

Standard randomized benchmarking tries to quantify the average error rate over a gate set (typically the $n-$qubit Clifford group). As such, it can, at best, provide only a coarse-grained information regarding the errors in the system. The goal of interleaved RB is to gain more detailed information, namely, it is designed to quantify the error of an individual gate of interest. This error characterization is accomplished by considering the following modified sequence operator[18] (see also fig.(8)):

$$S_{\mathbf{i}_m} \underset{\text{Eq.(3)}}{=} \Lambda_{\text{inv}} \circ C_{\text{inv}} \circ \Lambda_{i_m} \circ C_{i_m} \circ \cdots \circ \Lambda_{i_2} \circ C_{i_2} \circ \Lambda_{i_1} \circ C_{i_1} \tag{83}$$

$$\mapsto \boxed{S_{\mathbf{i}_m} = \Lambda_{\text{inv}} \circ C_{\text{inv}} \circ C \circ \Lambda_C \circ \Lambda_{i_m} \circ C_{i_m} \circ \cdots \circ C \circ \Lambda_C \circ \Lambda_{i_2} \circ C_{i_2} \circ C \circ \Lambda_C \circ \Lambda_{i_1} \circ C_{i_1}} \,,$$

where $C$ denotes the particular Clifford gate one wishes to characterize, $\Lambda_C$ denotes its noise channel, and $\Lambda_{i_j}$ is the error channel associated with a randomly chosen Clifford gate. The $S_{\mathbf{i}_m}$ operator emerges from constructing sequences where each randomly selected Clifford gate is interleaved by the specific Clifford $C$ we want to benchmark. More precisely, the first gate is chosen uniformly at random from the Clifford group, and is followed by the specific Clifford gate $C$; next, we again randomly select an element form the Clifford group, and follow it by the deterministic choice for $C$, and proceed in this fashion until a total of $m$ random gates have

---

[18]We are writing the sequence operator with possible gate-dependent errors, but not time-dependent errors.

been implemented. The last step is the implementation of the inverse sequence operator $C_{\text{inv}}$. The gate $C_{\text{inv}}$ is defined, just as before, as:

$$C_{\text{inv}} = \left(C \circ C_{i_m} \circ \cdots \circ C \circ C_{i_2} \circ C \circ C_{i_1}\right)^{\dagger} = C_{i_1}^{\dagger} \circ C^{\dagger} \circ C_{i_2}^{\dagger} \circ C^{\dagger} \circ \cdots \circ C_{i_m}^{\dagger} \circ C^{\dagger}. \tag{84}$$

The full interleaved RB protocol entails two main steps:

(i) Implementation of the standard RB protocol over the suitable $n-$qubit Clifford group. From this experiment, we estimate the depolarizing parameter $p$, associated with the average error map $\Lambda$.

(ii) Implementation of a similar scheme as the one outlined for the standard RB protocol, but now with each sequence defined as in Eq.(83). This allows us to estimate a new depolarizing parameter, $p_{\bar{C}}$, associated to the average error channel $\left(\Lambda_C \circ \Lambda\right)$.

The goal is to retrieve a quantitative metric that estimates the error rate associated with the gate $C$. If $r_C$ denotes the true error rate of $C$, $r_C^{\text{est}}$ refers to its estimate. The latter is given by:

$$r_C^{\text{est}} = \frac{(d-1)\left(1 - \frac{p_{\bar{C}}}{p}\right)}{d}, \tag{85}$$

with $d = 2^n$. In [6], it is proven that the *true* value of $r_C$ is bounded to the interval $[r_C^{\text{est}} - E, r_C^{\text{est}} + E]$, with $E$:

$$E = \min \begin{cases} \frac{(d-1)[(1-p)+|p-(p_{\bar{C}}/p)|]}{d}, \\ \frac{2(d^2-1)(1-p)}{pd^2} + \frac{4\sqrt{1-p}\sqrt{d^2-1}}{p}. \end{cases} \tag{86}$$

Eq.(85) and Eq.(86) are the key results of the interleaved protocol, allowing us to estimate the average error rate and also bound its error. In the next section, we provide a re-derivation of Eq.(85), but refer to [6] for a proof of Eq.(86).

## 4.2 Estimation of $r_C$

We start from the sequence operator $S_{\mathbf{i}_m}$ in Eq.(83). Just as in standard RB, this operator can be more concisely expressed if we preform a change of variables to the new gate set $D_{k_j}$. These are constructed recursively from the *old* gates as:

(i) $D_{k_1} = C_{k_1}$,

(ii) $D_{k_{i+1}} = C_{k_{i+1}} \circ C \circ D_{k_j} \implies D_{k_{i+1}}^{\dagger} = D_{k_j}^{\dagger} \circ D^{\dagger} \circ C_{k_{i+1}}^{\dagger}$.

In terms of the new gates, the sequence operator reads:

$$S_{\mathbf{i}_m} = \Lambda_{\text{inv}} \circ \left( \bigcirc_{j=1}^{m} \left[ D_{k_j}^{\dagger} \circ \Lambda_C \circ \Lambda_{k_j} \circ D_{k_j} \right] \right). \tag{87}$$

Under the assumption of gate-independent noise, $\Lambda_{k_j} = \Lambda$. This allows the average sequence operator (defined in Eq.(6)) to be written in terms of the twirling channel as follows:

$$\begin{aligned} S_m &= \left(\Lambda_C \circ \Lambda\right) \circ \left( \bigcirc_{j=1}^{m} \frac{1}{|\mathcal{C}_n|} \left[ D_{k_j}^{\dagger} \circ \left(\Lambda_C \circ \Lambda\right) \circ D_{k_j} \right] \right) \\ &= \Lambda_{\bar{C}} \circ \left( \bigcirc_{j=1}^{m} \frac{1}{|\mathcal{C}_n|} \left[ D_{k_j}^{\dagger} \circ \Lambda_{\bar{C}} \circ D_{k_j} \right] \right) = \Lambda_{\bar{C}} \circ \left( \Lambda_{\bar{C}}^{\text{Twirl}} \right)^{\circ m}, \end{aligned} \tag{88}$$

where $|\mathcal{C}_n|$ denotes the order of the $n-$qubit Clifford group. The new gates $D_{k_j}$ are equivalent representatives of the Clifford elements. We can then move directly to the superoperator formalism, in which the compositions become matrix products, and where the twirling channel can be explicitly constructed in terms of the irreducible representation of the $n-$qubit Clifford group (see Eq.(36)):

$$\hat{\Lambda}_{\bar{C}}^{\text{Twirl}} = |\sigma_0\rangle\rangle\langle\langle\sigma_0| + p_{\bar{C}} \sum_{i\neq0} |\sigma_i\rangle\rangle\langle\langle\sigma_i| \implies \left(\hat{\Lambda}_{\bar{C}}^{\text{Twirl}}\right)^m = |\sigma_0\rangle\rangle\langle\langle\sigma_0| + p_{\bar{C}}^m \sum_{i\neq0} |\sigma_i\rangle\rangle\langle\langle\sigma_i|. \quad (89)$$

In this notation, the sequence fidelity is expressed as the inner product:

$$\begin{aligned}
F_{\text{seq}}(m,\rho_0) &= \langle\langle E|\hat{\Lambda}_{\bar{C}}\left(\hat{\Lambda}_{\bar{C}}^{\text{Twirl}}\right)^m|\rho_0\rangle\rangle \\
&= \langle\langle\tilde{E}|\left(\hat{\Lambda}_{\bar{C}}^{\text{Twirl}}\right)^m|\rho_0\rangle\rangle \\
&= B_0 + A_0\, p_{\bar{C}}^m,
\end{aligned} \quad (90)$$

which once again demonstrates the exponential decay profile characteristic of RB methods. Hence, we can determine $p_{\bar{C}}$ from the experimental fit, in the usual manner. Since $p$ would be retrieved from performing standard RB in stage 1 of the protocol, all parameters required to evaluate the estimate $r_C^{\text{est}}$ are then defined.

The origin of Eq.(85) arises from extracting a bounded domain for $r_C$. This can be obtained by assessing the difference in the average gate fidelity between the error maps $\Lambda_{\bar{C}} = \Lambda_C \circ \Lambda$ and $\Lambda_{\tilde{C}} = \Lambda_C \circ \Lambda_d$, where $\Lambda_d$ is the depolarizing channel. Recall that the average gate fidelity is defined as:

$$\bar{F}_{\Lambda,I} = \int_{\text{Haar}} d\phi\ \text{Tr}\big(|\phi\rangle\langle\phi|\,\Lambda(|\phi\rangle\langle\phi|)\big).$$

With this definition, plus the fact that $\Lambda_d(|\phi\rangle\langle\phi|) = p|\phi\rangle\langle\phi| + (1-p)\frac{\mathbb{1}}{d}$, it follows that:

$$\bar{F}_{\Lambda_{\tilde{C}},I} = p\bar{F}_{\Lambda_C,I} + \frac{(1-p)}{d}. \quad (91)$$

Let us then denote $E$ as the upper bound of:

$$\frac{|\bar{F}_{\Lambda_{\bar{C}},I} - \bar{F}_{\Lambda_{\tilde{C}},I}|}{p} \leq E, \quad (92)$$

which allows us to write:

$$\frac{\bar{F}_{\Lambda_{\bar{C}},I}}{p} - \frac{(1-p)}{pd} - E \leq \bar{F}_{\Lambda_C,I} \leq \frac{\bar{F}_{\Lambda_{\bar{C}},I}}{p} - \frac{(1-p)}{pd} + E. \quad (93)$$

Note that $\bar{F}_{\Lambda_C,I}$ is the average gate fidelity of the error map for gate $C$, which is precisely what we aim to find. Letting $r_C$ be defined in terms of the corresponding infidelity, i.e. $r_C := 1 - \bar{F}_{\Lambda_C,I}$, and noting that $\bar{F}_{\Lambda_{\bar{C}},I}$ is equivalent to the average gate fidelity of a depolarizing channel, with depolarizing parameter $p_{\bar{C}}$,

$$\bar{F}_{\Lambda_{\bar{C}},I} = p_{\bar{C}} + \frac{1-p_{\bar{C}}}{d}. \quad (94)$$

Eq.(93) can be re-arrange to read:

$$\frac{(d-1)(1-\frac{p_{\bar{C}}}{p})}{d} - E \leq r_C \leq E + \frac{(d-1)(1-\frac{p_{\bar{C}}}{p})}{d},$$

which leads to the definition of the estimate of $r_C$ as in Eq.(85).

## 4.3 Key results in chapter 4

Here we summarize the main takeaways for the interleaved RB protocol.

- The goal of interleaved RB is to be able to estimate the average gate fidelity (or the average error rate) of a particular gate.

- The interleaved protocol requires running two separate experiments: First running a standard RB experiment, and then running a similar experiment, but where the sequences of random gates are modified such that each randomly selected gate is interleaved by the specific gate we want to benchmark.

- Like the standard RB protocol, interleaved RB requires us to select gates from a unitary 2-design, including the gate we want to characterize.

- The estimated error rate for the gate of interest is given by:

$$r_C^{\text{est}} = \frac{(d-1)\left(1 - \frac{p_{\tilde{C}}}{p}\right)}{d},$$

where $d = 2^n$, $p$ is the effective depolarizing parameter estimated from the standard RB experiment and $p_{\tilde{C}}$ is the effective depolarizing parameter estimated from the experiment running the interleaved sequences.

- The error rate of the desired gate is proven in Ref. [6] to be bounded to the interval:

$$r_C \in \left[ r_C^{\text{est}} - E, r_C^{\text{est}} + E \right],$$

$$E = \min \begin{cases} \frac{(d-1)[(1-p)+|p-(p_{\tilde{C}}/p)|]}{d}, \\ \frac{2(d^2-1)(1-p)}{pd^2} + \frac{4\sqrt{1-p}\sqrt{d^2-1}}{p}. \end{cases}$$

# 5 Gate-set shadow protocol

The accompanying notebook for this section can be found at: https://gitlab.com/QMAI/papers/rb-tutorial/-/blob/main/Gate%20set%20shadows/_GateSetShadowProtocol.ipynb.

By now, we have explored four RB variants. All of them were designed to infer particular aspects of the gate noise. They required us to run different experimental protocols.[19] This approach can become inefficient, if we wish to learn multiple different noise aspects.

More recently, classical shadow tomography has emerged as an efficient tool for characterization of quantum states [9]. Its main goal is to efficiently estimate multiple expectation values from the same set of randomized experiments. This efficiency is accomplished by guaranteeing that a desired level of accuracy is attained by collecting a total number of samples that grows proportionally with $\log(M)$, $M$ being the total number of observables we wish to evaluate [9]. The question, then, arises: Can the same philosophy be employed to randomized benchmarking protocols? The gate-set shadow protocol provides an affirmative answer to this question.

The main goal of the gate-set protocol is to provide a method for extracting knowledge on many different observables from the same dataset, in a *sample-efficient* manner. This means that the protocol should yield information on an large number of desired quantities, at the cost of performing measurements on a number of samples, growing only polynomially with the intended number of observables [11]. The advantage of the gate-set method [11] is that it requires a single phase of data collection, from which estimators belonging to different classes of experimental protocols can be gauged. This obliterates the need for running different tailored experiments to learn specific quantities. In the next section, we will describe the different stages of the protocol, and which estimators should be computed from the data.

## 5.1 Generic description of the method

There are two main stages of the gate-set protocol: (1) a data-collection phase, and (2) a post-processing phase. These are sequential, but independent steps. The first step entails the experimental setup, implementing different random circuits. The properties of the circuits,[20] as well as the measurement outcomes provide the data. For a fixed initial state, different realizations of the random circuit will give rise to different outcomes, in a predefined computational basis. More precisely, step (1) encompasses the execution of the following ordered tasks:

(a) Prepare an initial state $\rho$. This could be, for example, the state $\rho = |0\rangle\langle 0|$.[a]

(b) Draw, at random, a sequence of gates from a given group $G$. This sequence is referred to as **g**. The way these gates are selected follows a certain probability distribution. In RB techniques, the default is using a uniform distribution, but in principle we could imagine performing this selection according to a different distribution.

(c) Apply the random sequence to the prepared state $\rho$.

---

[19]With the exception of correlated RB, which simply supplements simultaneous RB with a more targeted post-processing phase to learn correlated errors.

[20]Properties like which random gates are used, in which sequence, and how many gates in total are employed.

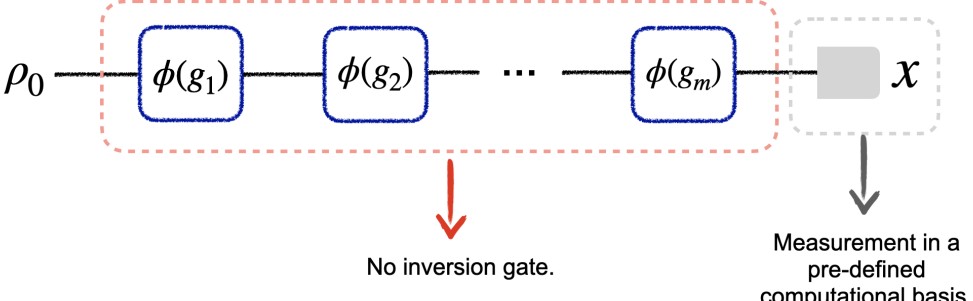

Figure 9: Like standard RB, the gate-set shadow protocol requires us to implement random sequences of gates. In the figure, $\phi(g_i)$ denotes the noisy implementation of the gate $g_i$. Unlike standard RB, the random sequences in the gate-set shadow protocol do not require a sequence inversion gate at the end. Instead, the random gates are followed directly by a measurement in a computational basis. For each random sequence, the protocol will keep track of: Which gates were performed ($\{\phi(g_1), \phi(g_2), \ldots, \phi(g_m)\}$), and what outcome was measured ($x$).

---

(d) Choose a computational basis. Without loss of generality, let us refer to it as $|x\rangle\rangle$, with $x$ a bit string, i.e. $x \in \{0, 1\}^{\otimes n}$.[b] This defines a corresponding set of POVM operators $\{E_x\}_x$, for each possible measurement outcome $x$. Hence, this step entails performing a measurement of the system and registering the observed outcome $x$. By the end of this step, we know which sequence of random gates was performed, and what outcome it yield. We have then produced the tuple $(x, \mathbf{g})$. This tuple is denoted as the gate-set shadow.

(e) Repeat steps (b) to (d) for different gate sequences, chosen randomly from $G$. These sequences should still all have the same length $m$. Each of them will produce a tuple $(x_i, \mathbf{g}_i)$. Suppose then that we undergo steps (b) to (d) $K$ times. By the end of this step, we collected information on a series of tuples $\{x_i, \mathbf{g}_i\}_{i=1}^K$.

(f) Repeat steps (b) to (e) for different sequence lengths.

---

[a]Which in the Liouville representation reads $|\rho\rangle\rangle = |0\rangle\rangle$.

[b]This means $x$ can be any string of $n$ values that can be constructed from choosing n elements from the binary set $\{0, 1\}$.

---

If in total we have measured $S$ samples, then the set of tuples $\{(x_i, \mathbf{g}_i)\}_{i=1}^S$ creates the dataset. Let us point out that the experimental phase of the protocol has a significant overlap with the approach followed in standard RB. Assuming the gates are selected uniformly at random, the major points of contrast are the absence of an inversion gate in each $\mathbf{g}_i$, and the fact that one does not constraint itself to measure exclusively the probability of retrieving the input state as output. What follows next is the post-processing phase, done in a classical platform. In this stage, one computes a series of functions, defined for each quantity we wish to learn,

and based on the data-set collected on stage (1). These functions are defined as follows [11]:

$$
\begin{aligned}
f_A(x, \mathbf{g}) &= \alpha \operatorname{Tr}\left( E_x \sigma(g_m) \prod_{i=1}^{m-1} A\sigma(g_i)(\rho) \right) \\
&= \alpha \langle\langle E_x | \sigma(g_m) \prod_{i=1}^{m-1} A\sigma(g_i) | \rho \rangle\rangle,
\end{aligned}
\tag{95}
$$

where $\alpha$ is a normalization factor, $\sigma(.)$ is an irreducible representation of $G$ and $A$ is a matrix (named probe superoperator). The precise nature of the matrix $A$ depends on which quantity we wish to estimate. In practice, the essence of the postprocessing phase is to construct the *right* correlation functions and take their averages to estimate the quantities we are interested in. Therefore, the evaluation of Eq.(95) is an important step in the protocol.

The functions $f_A(.)$ serve as the building blocks for the (statistical) estimators of the observables we wish to learn. They are referred to as correlation functions and they give rise to the final estimator through the median-of-means:

$$
\hat{k}_A(m) = \operatorname{median}\left[ \frac{1}{N} \sum_{i=J}^{J+N-1} f_A(x_i, \mathbf{g}_i) \,\|\, J \in \{1, N+1, 2N+1, \ldots, (K-1)N+1\} \right],
\tag{96}
$$

where we assume that on stage (1) a total of $S = NK$ samples, with the same length $m$, were measured. Note that $N$ and $K$ are meant to be both integers. This is not the first time we come across correlation functions. Indeed, in section 3.1 we have argued for their usefulness in characterising crosstalk. While that made sense for the specific purpose of correlated RB, correlators have a more broader value in revealing properties of the underlying probability distribution [34]. We can think of the functions $f_A(.)$ as the expectation value of the sequence operator in the computational basis, encoding information on the probability density over the set of possible outcomes. The sequence operator is now given as:[21]

$$
S_{\mathbf{g}_i} = \sigma(g_m) A\sigma(g_{m-1}) A\sigma(g_{m-2}) \cdots A\sigma(g_2) A\sigma(g_1).
\tag{97}
$$

Its expectation value can be seen as a correlator, since for $m > 1$ it depends on an increasing number of product of irreducible representations of the gate-set (which are chosen at random). The resulting probability density depends not only on the gate-set, but also on the choice of probe superoperator $A$. Alternatively, since the range of values for $f_A(.)$ will be bounded, we can view it as random variable, whose probability distribution corresponds to the probability of measuring outcome $x$, as a result of the action of the noisy gate sequence $\mathbf{g}$ over the input state $\rho$. The resulting expectation value of $f_A(.)$ gives rise to the desired estimator $\hat{k}_A$. Later on, we will see how a particular choice for the probe-operator and gate-set $G$ leads to an estimator $\hat{k}_A$ that coincides with the sequence fidelity in standard RB.

In practice, to determine the estimator $\hat{k}_A$, Eq.(96) needs to be assessed for different sequence lengths, and compared to the theoretical model ($k_f$) for the same quantity.[22] Hence, stage (2) of the protocol can be summarized as performing the following set of (ordered) steps:

---

[21]We follow the notation of Ref. [11], but note that this sequence operator has essentially the same structure as the one introduced in Eq.(18) for standard RB. The only difference is that now the sequence interleaves the probe-operator A, instead of the noise channel, and we directly think in terms of the Liouville representation of superoperators, where composition is replaced by multiplication.

[22]Just like in standard RB, where we compare the average sequence fidelity with the theoretical prediction for the same fidelity.

(a) Compute $f_A$ for all $S$ entries $(x_i, \mathbf{g}_i)$ in the dataset, having the same sequence length $m$.

(b) Take the median-of-means over the previously generated set $\{f_A(x_i, \mathbf{g}_i)\}_{i=1}^{S}$ to evaluate the estimator $\hat{k}_A(m)$.

(c) Repeat (a) and (b) for different values of $m$.

(d) Compare the estimators $\hat{k}_A(m)$ with the theoretical model $k_A(m)$, i.e. the theoretical model is fitted to the $\{\hat{k}_A(m_i)\}$ data, in order to estimate the desired parameters in the model. Note that the model is dependent on the quantity we wish to learn.

In [11], one can find a thorough analysis of how the gate-set protocol is sample-efficient. For the purpose of these notes, we choose to focus on how the procedure allows to estimate familiar quantities.

## 5.2 The theoretical model for $\hat{k}_A(m)$

On the actual experimental setup, the implementation of the gates corresponding to elements in the group $G$ are likely to be faulty. Hence, if the idealized operation we want to perform on same state $\rho$ is given as:

$$w(g)(\rho) = U_g \rho U_g^\dagger, \qquad U_g \in G, \tag{98}$$

then the noisy (more realistic) version of the same operation is:

$$\phi(g) = \Lambda_L w(g) \Lambda_R, \tag{99}$$

with the quantum channels $\Lambda_L$ and $\Lambda_R$ representing gate-independent noise channels. The sequence of gate operations is then given by the product of the noisy gate channels in Eq.(99),

$$
\begin{aligned}
\epsilon(\mathbf{g}) &= \prod_{i=1}^{m} \phi(g_i) \\
&= \Lambda_L w(g_m) \underbrace{\Lambda_R \Lambda_L}_{\Lambda := \Lambda_R \Lambda_L} w(g_{m-1}) \Lambda_R \cdots \Lambda_L w(g_2) \Lambda_R \Lambda_L w(g_1) \Lambda_R \\
&= \Lambda_L w(g_m) \left( \prod_{i=1}^{m-1} \Lambda w(g_i) \right) \Lambda_R.
\end{aligned}
\tag{100}
$$

Let us point out that $w(g)$ corresponds to a reducible representation of the group $G$. This is also the case in standard RB, where the Clifford gates are implemented by noisy super-operators, whose matrix versions encode a reducible representation of the Clifford group. In this sense, while the functions $f_A(.)$ are already defined with respect to the irreducible representations of $G$, the sequence operations are kept more generic. In what follows next, we assume that the gates are randomly selected from $G$ with a uniform distribution.

The object $\hat{k}_A(m)$ is estimating the expectation value $\mathbb{E}(f_A)$ from the sampled data, and deviates from it up to a certain accuracy $\varepsilon$ (see appendix in [11], eqs.(41)-(42)):

$$|\hat{k}_A - \mathbb{E}(f_A)| \leq \varepsilon, \tag{101}$$

with the value of $\varepsilon$ in part controlled by the total number of collected samples during the experiment.[23] In the limit of infinite samples, the value of the estimator $\hat{k}_A$ converges to the

---

[23]The magnitude of $\varepsilon$ is also controlled by other parameters that depend on the type of observables we want to measure, but loosely speaking one expects that, keeping the other parameters fixed, the more samples we use, the more $\varepsilon$ decreases.

true expectation value, and it is this relation that defines its theoretical model, $k_A$:

$$k_A = \mathbb{E}(f_A) = \mathbb{E}_{\mathbf{g} \in G} \sum_{x \in \{0,1\}^n} f_A(x, \mathbf{g}) \, p(x, \mathbf{g}), \tag{102}$$

where $p(x, \mathbf{g})$ corresponds to the probability of measuring outcome $x$, as a result of the action of the gate sequence $\mathbf{g}$ over the input state $\rho$. Note that Eq.(102) aligns with the usual definition of expectation value. This is because the probability of obtaining a particular value for $f_A$ is equivalent to the probability of obtaining a specific realization of the tuple $(x, \mathbf{g})$. We can view the latter event as the result of two independent random events: First the selection of $m$ elements from the group $G$, which is assumed to follow a uniform distribution; second, measuring a particular outcome $x$. Given that both state preparation and measurement errors (SPAM) affect the value of the measured outcome, it is reasonable to encode this into the probability of measuring $x$. All these factors combined, lead to the probability distribution:

$$p(x, \mathbf{g}) = \langle\!\langle \tilde{E}_x \mid \epsilon(\mathbf{g}) \mid \tilde{\rho} \rangle\!\rangle, \tag{103}$$

where $\epsilon(\mathbf{g})$ is the sequence operator given in Eq.(100), $\mid \tilde{\rho} \rangle\!\rangle$ stands for a state $\mid \rho \rangle\!\rangle$ with preparation errors, while $\mid \tilde{E}_x \rangle\!\rangle$ is the POVM element, subjected to measurement errors. Inserting Eq.(103) into Eq.(102), it is possible to show that the theoretical model $k_A$ reads (see appendix in [11]):

$$k_A(m) = \text{Tr}\left[ \Theta\left(\{E_x\}_x, \rho\right) \left(\Phi(A, \Lambda)\right)^{m-1} \right], \tag{104}$$

where $\Theta\left(\{E_x\}_x, \rho\right)$ is a matrix that encodes all SPAM dependency, while $\Phi(A, \Lambda)$ encodes the information on the decaying parameters in the model. The latter are the typical features one wishes to estimate from an RB protocol. Note that Eq.(104) defines the practical way to estimate the desired decay parameters: It will be our fitting model. This equation is therefore central to the protocol.

The matrix elements of $\Phi$ are given by:

$$\Phi_{i,j} = \frac{1}{\dim(\mathbb{P}_j)} \text{Tr}\left(\mathbb{P}_i A \mathbb{P}_j \Lambda\right), \tag{105}$$

with $\mathbb{P}_j$ the projector onto the jth irreducible representation contained in $w$, and $\dim(\mathbb{P}_j)$ stands for the dimension of the subspace onto which this projector maps on.

## 5.3 Example: Standard RB as a gate-set shadow

In standard RB, the gates are randomly selected from the $n-$qubit Clifford group. This group has two irreducible representations, corresponding to two invariant subspaces: A trivial one spanned by $\{|\sigma_0\rangle\rangle\}$, and a non-trivial one, spanned by all the normalized and vectorized Pauli operators that do not match the identity. Let us name the trivial and non-trivial subspaces as $W_0$ and $W_1$, respectively. Consider then that $A \in W_1$, namely $A$ is the projector onto this subspace: $A = \mathbb{P}_1$. Then, from Eq.(104), the theoretical model $k_A(m)$ reads:

$$k_{\mathbb{P}_1}(m) = c_1 \left( \frac{\text{Tr}\left(\mathbb{P}_1 \Lambda\right)}{d^2 - 1} \right)^{m-1}, \tag{106}$$

with $c_1$ capturing all the SPAM dependency. Hence, $k_{\mathbb{P}_1}(m)$ has a familiar form: It is intimately related to the fidelity in standard RB. The theoretical prediction for fidelity in the latter protocol

is given by:

$$
\begin{aligned}
\mathcal{F}_{\text{seq}}(\rho) &= \langle\langle E_\rho | \Lambda \Lambda_T^m | \rho \rangle\rangle \\
&= \underbrace{\langle\langle \tilde{E}_\rho | \mathbb{P}_0 | \rho \rangle\rangle}_{:=\tilde{e}_0} + \left( \frac{\text{Tr}(\mathbb{P}_1 \mathcal{R}_\Lambda)}{d^2 - 1} \right)^m \underbrace{\langle\langle \tilde{E}_\rho | \mathbb{P}_1 | \rho \rangle\rangle}_{:=\tilde{e}_1} \\
&= \tilde{e}_0 + \left( \frac{\text{Tr}(\mathbb{P}_1 \mathcal{R}_\Lambda)}{d^2 - 1} \right)^m \tilde{e}_1 ,
\end{aligned}
\tag{107}
$$

where again the pre-factors $\tilde{e}_i$ encode all the SPAM dependency. We recognize the depolarizing parameter as being $\frac{\text{Tr}(\mathbb{P}_1 \mathcal{R}_\Lambda)}{d^2-1}$. Indeed, this is the same as the one in Eq.(106). To make this statement more obvious, recall that the Pauli transfer matrix is defined as:

$$
\text{Tr}(\mathbb{P}_1 \mathcal{R}_\Lambda) = \sum_{\tau \in W_1} \langle\langle \tau | \mathcal{R}_\Lambda | \tau \rangle\rangle = \sum_{\tau \in W_1} (\mathcal{R}_\Lambda)_{\tau,\tau} = \sum_{\tau \in W_1} \text{Tr}(\tau \Lambda(\tau)) = \sum_{\tau \in W_1} \langle\langle \tau | \Lambda | \tau \rangle\rangle = \text{Tr}(\mathbb{P}_1 \Lambda) . \tag{108}
$$

Retrieving the effective depolarizing parameter for standard RB exemplifies of how the gate-set shadow is able to reconstruct the figures of merit of existing RB protocols. However, it is far from making justice to the full potential of this method. The protocol presents a general framework to reconstruct any RB protocol via the appropriate choice of sequence correlation functions $f_A$ [11].

## 5.4 Key results in chapter 5

Here we summarize the main takeaways for the gate-set shadow protocol.

- The gate-set shadow protocol is a versatile and sample efficient protocol aimed at characterizing gate noise.

- The protocol consists of two main phases: A data collection phase followed by a postprocessing phase. The data collection phase consists of an RB-like experiment, except that no inverse gate is applied at the end, and we measure in a computational basis. The data set consists of all pairs $\{(x_i, \mathbf{g}_i)\}_{i=1}^S$, i.e. all the sequences of gates $\mathbf{g}_i$ and corresponding measurement outcomes for all $S$ experiments.

- The versatile quality of the protocol comes from the possibility of estimating many figures of merit in the postprocessing step, from the same data set. This is done by appropriately choosing probe operators, $A$, and using them to construct sequence correlation functions, defined as:

$$
f_A(x, \mathbf{g}) = \alpha \langle\langle E_x | \sigma(g_m) \prod_{i=1}^{m-1} A\sigma(g_i) | \rho \rangle\rangle ,
$$

where $\sigma(g_i)$ corresponds to an irreducible representation of the gate $g_i$, $\alpha$ is a normalization constant, and $E_x$ is the POVM element, measuring the probability of obtaining outcome $x$.

- The figures of merit are the expectation values of the sequence correlation functions, over all $S = NK$ samples:

$$
\hat{k}_A(m) = \text{median}\left[ \frac{1}{N} \sum_{i=J}^{J+N-1} f_A(x_i, \mathbf{g}_i) \,\Big|\, J \in \{1, N+1, 2N+1, \ldots, (K-1)N+1\} \right] .
$$

The median of means is used to prove the sample efficiency of the protocol.

- In principle, we can choose our gates from any gate set $G$. However, different gate sets lead to different sequence correlation functions, so our choice of gate set should be guided by the quantities we want to estimate.

- Each choice of probe operator leads to a different theoretical model for $\hat{k}_A(m)$, which in general will have the form:

$$k_A(m) = \mathrm{Tr}\Big[\Theta(\{E_x\}_x, \rho)\big(\Phi(A, \Lambda)\big)^{m-1}\Big],$$
$$\Phi_{i,j} = \frac{1}{\dim(\mathbb{P}_j)}\,\mathrm{Tr}\big(\mathbb{P}_i A \mathbb{P}_j \Lambda\big),$$

where $\Theta(\{E_x\}_x, \rho)$ is a matrix that encodes all SPAM dependency, while $\Phi(A, \Lambda)$ encodes the information on the decaying parameters in the model. Similar to standard RB, we fit the theoretical model to the sample mean (median of means) over the correlation functions to estimate the decay parameters in the model. The physical interpretation of the decaying parameters depends on the underlying theoretical model and thus on the choice of the probe operator $A$. Like the standard RB protocol, the gate-set shadow protocol produces figures of merit that are robust to SPAM errors.

Table 5: Summary of the main features for the four commonly used RB protocols: Standard RB, simultaneous RB, correlated RB and interleaved RB.

| Protocol | Primary Goal | Main Assumptions | Figure of Merit |
|---|---|---|---|
| Standard RB | Estimate the average error rate of a gate-set. | Gate errors are time independent and weakly gate dependent. Noise is Markovian. Gate-set forms a unitary-2 design. | Average error rate $r$ (Eq.(30), or average gate fidelity $\bar{F} = 1 - r$. |
| Simultaneous RB | Flag the presence of crosstalk errors. | Gate errors are time independent and weakly gate dependent. Noise is Markovian. Protocol's efficiency is reliant on crosstalk errors being low-weight. | Crosstalk metric $\delta\alpha$ (Eq.(41)). Individual subsystem fidelities. |
| Correlated RB | Discern the weight and locality of crosstalk errors. | Same as simultaneous RB. | Estimation of the $\{\epsilon\}$ parameters that relate to the weight of crosstalk errors (Eq.(72)). |
| Interleaved RB | Estimate the error rate of a specific gate. | The errors are time independent and weakly gate dependent. Noise is Markovian. | Error rate associated to the target gate (Eq.(85)). |

# 6  Conclusions

In this tutorial we took hands-on, user-friendly approach to the theory and practice of randomized benchmarking techniques. We only focused on the most foundational, and, at the same time, most frequently used methods covered in Refs. [3–6]. Table 5 summarizes the main features of these protocols. We also contextualized randomized benchmarking in the more contemporary context of shadow tomography [11].

There are many more specialized RB methods, as for example: Leakage RB [35,36], RB beyond the Clifford group [28–30,37,38], RB using approximate sampling of the group [39], cross-entropy benchmarking [40], logical randomized benchmarking [41], direct RB (benchmarking native gates) [42], RB for quantum channel reconstruction [43], RB for estimating the coherent contribution to the error channel [44], RB for benchmarking universal gate sets [45], benchmarking analog quantum simulators [46], or binary RB (scalable protocol that does not rely on the inversion sequence gate) [47]. Similarly, the field at the boundary of shadow tomography and RB is growing fast. Useful references include: Shadow tomography adapted for

a faulty gate-set [48,49], or even the development of shadow process tomography [10]. These works, combined with the approach proposed in the gate-set shadow protocol, offer promising avenues for the characterization of multiple features of a quantum channel. We hope this tutorial can serve as a friendly entry point to the subject of randomized benchmarking, and that, at the same time, it may spark the interest for exploring the new developments in this growing field.

## Acknowledgments

We gratefully acknowledge productive discussions with Jonas Helsen and Christian Kraglund Andersen. We are grateful to Irene Fernandez De Fuentes, who kindly provided the experimental data for the "Benchmarking a real device" section, and whose time in clarifying the data structure and details of the experiment is greatly appreciated. We are grateful to Pieter Eendebak for his advice on implementation of accompanying code for this tutorial.

**Funding information** This publication is part of the 'Quantum Inspire - the Dutch Quantum Computer in the Cloud' project (with project number [NWA.1292.19.194]) of the NWA research program 'Research on Routes by Consortia (ORC)', which is funded by the Netherlands Organization for Scientific Research (NWO). This work is part of the project Engineered Topological Quantum Networks (Project No.VI.Veni.212.278) of the research program NWO Talent Programme Veni Science domain 2021 which is financed by the Dutch Research Council (NWO).

**Code and data availability** Complete code and data to reproduce this tutorial is available at this URL: https://gitlab.com/QMAI/papers/rb-tutorial.

## A Gate fidelity evaluated for pure states

The definition of gate fidelity in Eq.(9) came from the more complex expression giving the channel fidelity, Eq.(7). We would like to provide more clarity on how the channel fidelity simplifies to the gate fidelity, when $\rho$ is a pure state:

$$\left( \text{Tr} \left( \sqrt{ \sqrt{\epsilon_1(\rho)} \epsilon_2(\rho) \sqrt{\epsilon_1(\rho)} } \right) \right)^2 \stackrel{?}{=} \langle \phi | \hat{U} \epsilon(|\phi\rangle\langle\phi|) \hat{U} | \phi \rangle .$$

Recall that we have set $\epsilon_1$ as a unitary operation, while $\epsilon_2$ is remains a general quantum channel that we label by $\epsilon$. Being linear maps, we can choose to represent the two quantum channels by matrices. Additionally, a unitary operation can only map a pure state into another pure state:

$$\epsilon_1(\rho) = \hat{U}|\phi\rangle\langle\phi|\hat{U}^\dagger = |\psi\rangle\langle\psi| . \tag{A.1}$$

Let us then set $\hat{A}$ to be the matrix representation of the state $|\psi\rangle\langle\psi|$. The matrix $\hat{B}$ is said to be the square root of $\hat{A}$ if:

$$\hat{B}\hat{B} = \hat{A} . \tag{A.2}$$

Since $\hat{A}$ needs to be a hermitian matrix in a finite Hilbert space, it is diagonalizable in its eigenbasis: $\hat{A} = \hat{V}\hat{D}\hat{V}^{-1}$, where $\hat{V}$ is the matrix whose columns are the eigenvectors of $\hat{A}$ and $\hat{D}$ is a diagonal matrix storing the eigenvalues of $\hat{A}$. Furthermore, since $\rho$ is a hermitian CP map,

$\hat{A}$ needs to be a hermitian positive semi-definite matrix. This implies that the eigenvalues of $\hat{A}$ are positive and real. All these features reassure us that we can write $\hat{B}$ to be:

$$\hat{B} = \hat{V}\hat{D}^{1/2}\hat{V}^{-1}, \tag{A.3}$$

which automatically satisfies Eq.(A.2). Equivalently, we can express $\hat{B}$ as: $\hat{B} = \sum_i \sqrt{\lambda_i}\, |\lambda_i\rangle\langle\lambda_i|$, with $\{\sqrt{\lambda_i}\}_i$ and $\{|\lambda_i\rangle\}_i$ the set of eigenvalues and eigenvectors, respectively. But, given that $\hat{A}$ is a pure state, it follows that $\lambda_i = 1$ and $|\lambda_i\rangle = |\psi\rangle$, and so for pure states it holds that $\hat{B} = \hat{A}$. This then allows us to simplify Eq.(7) greatly,

$$
\begin{aligned}
\mathrm{Tr}\left(\sqrt{\sqrt{\epsilon_1(\rho)}\epsilon_2(\rho)\sqrt{\epsilon_1(\rho)}}\right) &\underset{\hat{B}=\hat{A}}{=} \mathrm{Tr}\left(\sqrt{\left(|\psi\rangle\langle\psi|\right)\epsilon(|\phi\rangle\langle\phi|)\left(|\psi\rangle\langle\psi|\right)}\right) \\
&= \mathrm{Tr}\left(\sqrt{\langle\psi|\epsilon(|\phi\rangle\langle\phi|)|\psi\rangle\,|\psi\rangle\langle\psi|}\right) \\
&\underset{\hat{B}=\hat{A}}{=} \mathrm{Tr}\left(\sqrt{\langle\psi|\epsilon(|\phi\rangle\langle\phi|)|\psi\rangle}\,|\psi\rangle\langle\psi|\right) \\
&= \sqrt{\langle\psi|\epsilon(|\phi\rangle\langle\phi|)|\psi\rangle}\,\mathrm{Tr}(|\psi\rangle\langle\psi|) \\
&= \sqrt{\langle\psi|\epsilon(|\phi\rangle\langle\phi|)|\psi\rangle} \\
\implies \quad F_{U,\epsilon}(|\phi\rangle\langle\phi|) &= \langle\psi|\epsilon(|\phi\rangle\langle\phi|)|\psi\rangle \\
&= \langle\phi|\hat{U}^\dagger\epsilon(|\phi\rangle\langle\phi|)\hat{U}|\phi\rangle.
\end{aligned}
\tag{A.4}
$$

# B   Average gate fidelity under twirling

We follow closely the argument provided in Ref. [19]. Combining the definition of a twirling channel, Eq.(14), with the definition of average gate fidelity, Eq.(12), we get:

$$
\begin{aligned}
\bar{F}_{\mathcal{I},\Lambda_T} &= \int d|\phi\rangle\,\mathrm{Tr}\big(\Lambda_T(|\phi\rangle\langle\phi|)\,|\phi\rangle\langle\phi|\big) \\
&= \int d|\phi\rangle \int d\mu_H(U)\,\mathrm{Tr}\big(\hat{U}^\dagger\Lambda(\hat{U}|\phi\rangle\langle\phi|\hat{U}^\dagger)\hat{U}\,|\phi\rangle\langle\phi|\big) \\
&= \int d\mu_H(U) \int d|\phi\rangle\,\langle\phi|\hat{U}^\dagger\Lambda(\hat{U}|\phi\rangle\langle\phi|\hat{U}^\dagger)\hat{U}\,|\phi\rangle \\
&\underset{|\psi\rangle=\hat{U}|\phi\rangle}{\overset{24}{=}} \int d\mu_H(U) \int d|\psi\rangle\,\langle\psi|\Lambda(|\psi\rangle\langle\psi|)|\psi\rangle \\
&= \bar{F}_{\mathcal{I},\Lambda} \int d\mu_H(U) = \bar{F}_{\mathcal{I},\Lambda}.
\end{aligned}
\tag{B.1}
$$

Thus, the average gate fidelity is invariant under twirling.

---

[24] The measure $d|\phi\rangle$ defines the integration over pure states. Recall that two quantum states are unique up to a $U(1)$ phase. Thus, the integration measure over quantum states should also reflect this property, and be unitarily invariant. More rigorously, the integration measure $d|\phi\rangle$ is given by the Fubini-Study measure [12]. See Ref. [50] for more technical details.

## C  Poor man's group theory: Short and simplified overview of some useful results

A group $G$ is a set of elements plus a binary operation (denoted generically as ".".[25]) that together fulfill the following requirements [16]:

1. For all $g_1, g_2 \in G$, then $g_1.g_2 \in G$ (closure).

2. For all $g_1, g_2, g_3 \in G$, then $(g_1.g_2).g_3 = g_1.(g_2.g_3)$ (associativity).

3. There exists an element $e \in G$, such that $e.g = g.e = g$ for all $g \in G$ (identity).

4. For all $g \in G$ there exists an element, such that $g.g^{-1} = g^{-1}.g = 1$ (inverse).

For example, the set of all $n \times n$ invertible matrices, in an $n-$dimensional vector space V, forms a group under matrix multiplication, called the general linear group $GL(V)$ [51]. Here matrix multiplication is the binary operation, and this set can be easily understood as forming a group. The product of two invertible matrices yields another invertible matrix $\in V$, hence closure is satisfied; being a set of invertible matrices, the inverse matrix of every element is necessarily contained in the set; the identity matrix is also invertible, hence belongs to the set; the associativity property holds for matrix multiplication. Thus, all group properties are satisfied.

A group homeomorphism is a map, call it $\varphi$, between two groups, $G$ and $H$, that is defined to respect the group structure, i.e. $\varphi : G \rightarrow H$, such that [52]:

1. $\varphi(g_1.g_2) = \varphi(g_1).\varphi(g_2)$, for all $g_1, g_2 \in G$.

2. If $e_G$ is the identity element in $G$, and $e_H$ is the identity element in $H$, $\varphi(e_G) = e_H$.

3. $\varphi(g^{-1}) = \varphi^{-1}(g)$ for all $g \in G$.

The group homomorphism $\varphi$ mapping a group $G$ to $GL(V)$ is called a linear representation [51]. This map provides an implementation of $G$ as a set of linear transformations on $V$. A concrete choice for $\varphi$, in this case, can be to fix it as a matrix representation. Then, to every group element $g \in G$, there corresponds a square matrix $\hat{D}$ in $V$, such that $\hat{D}(g_1.g_2) = \hat{D}(g_1)\hat{D}(g_2)$. Given that these matrices are themselves elements of $GL(V)$, the identity, the existence of inverse, plus the associative property are all necessarily satisfied. In this way, the properties that define $G$ are now reproduced in the vector space $V$, by the set of matrices $\hat{D}$. From now on, we will restrict ourselves to linear representations (most often, concretely using matrix representations), and will refer to them simply as representations.

Two representations, $\hat{D}_1$ and $\hat{D}_2$, are said to be equivalent if one can find a matrix $\hat{A} : V \rightarrow V$, such that [53]:

$$\hat{D}_1(g) = \hat{A}\hat{D}_2(g)\hat{A}^{-1}, \quad \text{for all } g \in G. \tag{C.1}$$

The linear transformation $\hat{D}(g) \rightarrow \hat{A}\hat{D}(g)\hat{A}^{-1}$ is called a similarity transformation. Any set of matrices that differ only by a similarity transformation are valid representations of a group. If this similarity transformation is implemented by a unitary matrix, then $\hat{A}^{-1} = \hat{A}^{\dagger}$.

Suppose our vector space $V$ contains a non-trivial vector subspace $W$.[26] If all the vectors in $W$ are mapped back to $W$ by the action of the representation $\hat{D}$, meaning

$$\hat{D}(g)\mathbf{w}_i = \mathbf{w}_j, \quad \text{for all } g \in G \text{ and all } \mathbf{w}_i, \mathbf{w}_j \in W, \tag{C.2}$$

---

[25]These can be, for example, addition or multiplication.

[26]Meaning that $W$ is vector space containing more than the null vector, and $W$ is not just $V$ itself.

then we can define a subrepresentation $\hat{D}_W$, which corresponds to a lower dimensional matrix (dim(W) × dim(W)), defined only over the subspace $W$. Because $W$ gets mapped back onto itself by $\hat{D}$, it is called an invariant subspace [54]. Note that if $W$ itself also contains a non-trivial subspace, then we are at liberty to define yet another subrepresentation that would be restricted to the new found subspace of $W$ [54]. This sets the stage for the concept of irreducible representations.

Given a vector space $V$ and a representation $\varphi$ of $G$ in $V$, if the only invariant subspaces of $V$ are the trivial ones, i.e.

$$\text{Trivial invariant subspaces of V: } W = \{\mathbf{0}\} \quad \text{and} \quad W = V, \tag{C.3}$$

then $\varphi$ is said to be an irreducible representation [54]. On the other hand, when we do find non-trivial invariant subspaces $W_i \subset V$, then $\varphi$ is said to be a reducible representation. For example, suppose that $V$ could be written as the sum of three disjoint invariant sub-spaces, $V = W_1 + W_2 + W_3$, then we could proceed by creating sub-representations for each of the $W_i$ subspaces. Let us then assume that we cannot further identify any remaining invariant sub-spaces in each of the $W_i$. This would then mean our subrepresentations are actually irreducible representations. For a representation $\hat{D}$, defined over $V$, this representation will be reducible under the previous definition, and only the representations restricted to the subspaces $W_i$ are irreducible. However, $V$ is the larger vector space, so the information regarding the irreducible representations must be encoded in $\hat{D}$. This motivates the following result. Whenever $\hat{D}(g)$ is a reducible matrix representation, it is possible to find a similarity transformation $\hat{A}$ that transforms $\hat{D}(g)$ into block diagonal form, for all $g \in G$ [53]:

$$\hat{A}\hat{D}(g)\hat{A}^{-1} = \begin{pmatrix} \hat{D}_1(g) & 0 & 0 \\ 0 & \hat{D}_2(g) & 0 \\ 0 & 0 & \hat{D}_3(g) \end{pmatrix}, \tag{C.4}$$

where we have limited ourselves to our example of 3 irreducible representations. In general, the block diagonal matrix will contain as many matrices along the diagonal as the total number of irreducible representations, and the same irreducible representation may appear multiple times. Thus, a general reducible representation can be expressed as the direct sum $\hat{D}(g) = n_1\hat{D}_1(g) \oplus n_2\hat{D}_2(g) \oplus \cdots \oplus n_m\hat{D}_m(g)$, with $n_i$ denoting the number of times the irreducible representation $i$ is contained in $\hat{D}$ [53]. When all $n_i = 1$, we refer to $\hat{D}$ as a multiplicity-free representation.

A very important result in group theory is Schur's lemma:

**Schur's lemma (1):** Take $\hat{D}$ and $\hat{\Gamma}$ to be two irreducible representations of $G$ in the corresponding vector spaces $W$ and $\tilde{W}$. If $\hat{A}$ is a linear transformation $\hat{A} : \hat{W} \to \tilde{W}$, obeying:

$$\hat{A}\hat{D}(g) = \hat{\Gamma}(g)\hat{A}, \quad \text{for all } g \in G, \tag{C.5}$$

then two cases apply [54]:

1. If $\hat{D}$ and $\hat{\Gamma}$ are inequivalent representations, $\hat{A} = \hat{0}$.

2. If $\hat{D}$ and $\hat{\Gamma}$ are equivalent representations, $\hat{A} = \lambda \mathbb{1}$ (with $\lambda$ a constant and $\mathbb{1}$ the identity matrix).

**Schur's lemma (2):** If $\hat{D}$ is a reducible representation in $V$, and $\hat{H}$ is some matrix $\hat{H} : V \to V$ that commutes with $\hat{D}$, i.e.

$$\hat{H}\hat{D}(g) = \hat{D}(g)\hat{H}, \quad \text{for all } g \in G, \tag{C.6}$$

then $\hat{H}$ has the form [52]

$$
\hat{H} = \begin{pmatrix} c_1 \mathbb{1}_1 & \hat{0} & \cdots & \hat{0} \\ \hat{0} & c_2 \mathbb{1}_2 & \cdots & \hat{0} \\ \vdots & \vdots & \ddots & \vdots \\ \hat{0} & \hat{0} & \cdots & c_n \mathbb{1}_n \end{pmatrix},
\tag{C.7}
$$

with each $c_i$ a constant and $\mathbb{1}_i$ the identity matrix (on the reduced invariant subspace $i$). The total number of constants equals to the total number of irreducible representations in $V$.

We will not prove Schur's lemma (1) here (see for example Refs. [53,54]), but we will provide a proof for part (2) in the multiplicity free case, since this result is avidly used throughout this tutorial. Let us start by the fact that if $\hat{D}$ is reducible, it can always be written in block diagonal form (Eq.(C.4)). This then means we can decompose the vector space $V$ into the sum $V = W_1 + W_2 + \ldots + W_n$, and we can define a set of projectors $\{\hat{P}_1, \hat{P}_2, .., \hat{P}_n\}$ mapping onto each invariant subspace. Given that $\sum_{i=1}^{n} \hat{P}_i = \mathbb{1}$, we can express the matrix $\hat{H}$ as:

$$
\hat{H} = \mathbb{1}\, \hat{H}\, \mathbb{1} = \sum_{i,j=1}^{n} \hat{P}_i \hat{H} \hat{P}_j \equiv \sum_{i,j=1}^{n} \hat{H}_{i,j}\,.
\tag{C.8}
$$

If we denote $\{|\phi_l^{(i)}\rangle\}$ as the set of orthonormal vectors that span the subspace $W_i$, then they provide a suitable basis for the corresponding projectors. Hence,

$$
\hat{H}_{i,j} = \sum_{l=1}^{\dim(W_i)} \sum_{k=1}^{\dim(W_j)} \langle \phi_l^{(i)} | \hat{H} | \phi_k^{(j)} \rangle\, |\phi_l^{(i)}\rangle \langle \phi_k^{(j)}|\,.
\tag{C.9}
$$

At the same time, $\hat{D}(g)$ being a block diagonal matrix, can be expressed in the same basis as:

$$
\hat{D}(g) = \sum_{m=1}^{n} \hat{P}_m \hat{D}_m(g) \hat{P}_m = \sum_{m=1}^{n} \sum_{l,k} \langle \phi_l^{(m)} | \hat{D}(g) | \phi_k^{(m)} \rangle\, |\phi_l^{(m)}\rangle \langle \phi_k^{(m)}|\,.
\tag{C.10}
$$

We can then explicitly compare $\hat{D}(g)\hat{H}$ with $\hat{H}\hat{D}(g)$, and determine what constraints arise in $\hat{H}$, in order to satisfy the commutation relation.

$$
\begin{aligned}
\hat{D}(g)\hat{H} &= \sum_{m,l,k,i,j,p,q} \langle \phi_l^{(m)} | \hat{D}(g) | \phi_k^{(m)} \rangle \langle \phi_p^{(i)} | \hat{H} | \phi_q^{(j)} \rangle\, |\phi_l^{(m)}\rangle \underbrace{\langle \phi_k^{(m)} | \phi_p^{(i)} \rangle}_{\delta_{m,i}\delta_{k,p}} \langle \phi_q^{(j)}| \\
&= \sum_{m,l,k,j,q} \langle \phi_l^{(m)} | \hat{D}(g) | \phi_k^{(m)} \rangle \langle \phi_k^{(m)} | \hat{H} | \phi_q^{(j)} \rangle\, |\phi_l^{(m)}\rangle \langle \phi_q^{(j)}| \\
&= \sum_{m,j} \sum_{l,q} \left( \sum_{k} \langle \phi_l^{(m)} | \hat{D}(g) | \phi_k^{(m)} \rangle \langle \phi_k^{(m)} | \hat{H} | \phi_q^{(j)} \rangle \right) |\phi_l^{(m)}\rangle \langle \phi_q^{(j)}| \\
&= \sum_{m,j} \sum_{l,q} \left( \sum_{k} \left( \hat{D}_m(g) \right)_{l,k} \left( \hat{H}_{m,j} \right)_{k,q} \right) |\phi_l^{(m)}\rangle \langle \phi_q^{(j)}| \\
&= \sum_{m,j} \sum_{l,q} \left( \hat{D}_m(g) \hat{H}_{m,j} \right)_{l,q} |\phi_l^{(m)}\rangle \langle \phi_q^{(j)}|\,.
\end{aligned}
\tag{C.11}
$$

Note that $\hat{D}_m(g)$ and $\hat{H}_{m,j}$ denote matrices, and not matrix elements. Their product results in a new matrix, from which we extract the matrix element on line $l$, column $q$, $\left( \hat{D}_m(g) \hat{H}_{m,j} \right)_{l,q}$. Likewise,

$$
\hat{H}\hat{D}(g) \sum_{m,j} \sum_{l,q} \left( \hat{H}_{m,j} \hat{D}_j(g) \right)_{l,q} |\phi_l^{(m)}\rangle \langle \phi_q^{(j)}|\,.
\tag{C.12}
$$

Hence, in order for $\hat{D}(g)$ and $\hat{H}$ to commute, we need:

$$\hat{H}_{m,j}\hat{D}_j(g) = \hat{D}_m(g)\hat{H}_{m,j}. \tag{C.13}$$

When $m \neq j$, the above constraint implies an equality relating two distinct irreducible representations. From Schur's lemma, it follows that $\hat{H}_{m,j} = \hat{0}$, for all $m \neq j$. On the other hand, when $m = j$, Schur's lemma tells us that $\hat{H}_{j,j} = c_j \mathbb{1}$, with $c_j$ some constant. Furthermore, $\mathbb{1}$ is the identity matrix in a reduced space, more precisely it is the unitary matrix defined only on the subspace $W_j$. Thus, we conclude that $\hat{H}$ has the diagonal form given in Eq.(C.7). This result will be very handy, when simplifying the twirling map, as we will seen in the next section.

Another useful result from group theory is how to construct direct product groups. If $G$ and $H$ are two groups, then the direct product group $G \times H$ has elements of the form $\{(g, h) \mid g \in G, h \in H\}$. Labeling the binary operation of $G$ and $H$ respectively as "$*$" and "$\star$", then the group $G \times H$ has a binary operation ("$.$") defined as [53]:

$$(g_1, h_1).(g_2, h_2) = (g1 * g_2, h_1 \star h_2). \tag{C.14}$$

Thinking concretely in terms of their linear representations, let us set the binary operation of $G \times H$ to be the Kronecker product $\otimes$, and $*$ and $\star$ to correspond to a matrix multiplication. Additionally, let us denote $\hat{D}(g)$ to be a representation of $G$, and $\hat{M}(h)$ that of $H$. Furthermore, if the representation $\hat{D}$ was previously expressed in terms of the basis vectors $\{|v_i\rangle\} \in V$, while $\hat{M}$ was given in terms of the basis vectors $\{|w_i\rangle\} \in W$, then we can think of a representation for $G \times H$ as being the set of appropriate matrices in the vector space $V \times W$, with basis vectors $\{|v_i\rangle \otimes |w_j\rangle\}$. For this reason, a natural representation becomes $\hat{D} \otimes \hat{M}$, acting in the basis vectors in a similar way as in Eq.(C.14), thus in line with the group structure:

$$\begin{aligned}
\left(\hat{D}(g_1) \otimes \hat{M}(h_1)\right)(|v_l\rangle \otimes |w_k\rangle) &= \left(\hat{D}(g_1)|v_l\rangle\right) \otimes \left(\hat{M}(h_1) \otimes |w_k\rangle\right), \\
\left(\hat{D}(g_2) \otimes \hat{M}(h_2)\right)\left(\hat{D}(g_1) \otimes \hat{M}(h_1)\right)(|v_l\rangle \otimes |w_k\rangle) &= \left(\hat{D}(g_2)\hat{D}(g_1)|v_l\rangle\right) \otimes \left(\hat{M}(h_2)\hat{M}(h_1)|w_k\rangle\right).
\end{aligned} \tag{C.15}$$

If we know the irreducible representations of $G$ and $H$, a way to construct representations for $G \times H$ is to consider all possible Kronecker products between the two sets of representations [53].

The final concept we will borrow from group theory is the definition of a subgroup, and in particular of what characterizes a normal subgroup. This will be import when introducing the action of the Clifford group on the Pauli group.

Given a group $G$, if there is a subset of its elements satisfying all group requirements (1)-(4), then that subset is itself a group, and it is referred to as a subgroup of $G$. For example, the Clifford group is a subgroup of the unitary group, and the Pauli group is a subgroup of the Clifford group. A subgroup $N$ is called a normal subgroup of $G$, if and only if [55]:

$$gng^{-1} \in N, \quad \text{for all } g \in G \text{ and } n \in N. \tag{C.16}$$

Hence, a normal subgroup is mapped onto itself by the action of the group $G$. The above relation does not imply that an element $n \in N$ is sent onto itself by $g$, just that it can be mapped onto any other element in $N$, and it will never be mapped outside that set.

# D  Twirling channel and the depolarizing channel

Let us bring more clarity on how the twirling channel yields a depolarizing channel. We again follow the arguments given in Ref. [19]. Given a unitary matrix $\hat{V}$, and using Eq.(14),

$\hat{V}\Lambda_T(\rho)\hat{V}^\dagger$ can be written as follows:

$$
\begin{aligned}
\hat{V}\Lambda_T(\rho)\hat{V}^\dagger &= \int d\mu_H(U)\,\hat{V}\hat{U}^\dagger\Lambda\big(\hat{U}\rho\hat{U}^\dagger\big)\hat{U}\hat{V}^\dagger \\
&\overset{27}{\underset{\hat{W}=\hat{U}\hat{V}^\dagger}{=}} \int d\mu_H(W)\,\hat{W}^\dagger\Lambda\big(\hat{W}\big(\hat{V}\rho\hat{V}^\dagger\big)\hat{W}^\dagger\big)\hat{W} \\
&= \Lambda_T\big(\hat{V}\rho\hat{V}^\dagger\big)\,.
\end{aligned}
\tag{D.1}
$$

If we define $\hat{P}$ to be a projector onto some subspace, then $\hat{Q} = \mathbb{1} - \hat{P}$ is the projector onto the orthocomplementary space. Since $\hat{V}$ is a unitary matrix, it is a representation of the unitary group. Even if this representation is not irreducible, we can always bring it into block diagonal form by some similarity transformation, with the irreducible representations filling in the diagonal (see Eq.(C.4)). Because the subspaces onto which $\hat{P}$ and $\hat{Q}$ project on are, by construction, disjoint subspaces, the irreducible representations will also split into representations that *live on* one of these subspaces. Therefore, $\hat{V}$ will be a block diagonal matrix with respect to the subspaces of $\hat{P}$ and $\hat{Q}$, without loss of generality.

Taking $\hat{P}$ to be a $1D$ projector, we may write $\hat{V}$ as follows:

$$
\hat{V} = |\phi\rangle\langle\phi| + \sum_{i,j} c_{i,j}|\varphi_i\rangle\langle\varphi_j|,
\tag{D.2}
$$

where $|\phi\rangle$ generates the $1D$ subspace onto which $\hat{P}$ projects on. The (higher dimensional) orthocomplementary subspace is spanned by the set of orthonormal vectors $\{|\varphi_j\rangle\}$. The coefficients are kept general, but in reality they are constrained by the fact that $\hat{V}$ has to be unitary, i.e. $\hat{V}\hat{V}^\dagger = \mathbb{1}$. Since $\langle\phi|\varphi_j\rangle = 0$, $\hat{V}$ is indeed a block diagonal matrix in the basis $\{|\phi\rangle, |\varphi_1\rangle, \ldots, |\varphi_n\rangle\}$. It is then straightforward to check that $\hat{V}$ and $\hat{P}$ commute:

$$
\hat{V}\hat{P}\hat{V}^\dagger = \hat{V}|\phi\rangle\langle\phi|\hat{V}^\dagger = \hat{P}\,.
\tag{D.3}
$$

With this knowledge, plus Eq.(D.1), we can conclude that also $\Lambda_T(\hat{P})$ commutes with $\hat{V}$:

$$
\hat{V}\Lambda_T(\hat{P})\hat{V}^\dagger = \Lambda_T(\hat{V}\hat{P}\hat{V}^\dagger) = \Lambda_T(\hat{P})\,.
\tag{D.4}
$$

Note that $\hat{V}$ can be any arbitrary unitary matrix. Hence, the previous conclusions imply that $\Lambda_T(\hat{P})$ needs to commute with every representation of the unitary group. By Schur's lemma, this means that $\Lambda_T(\hat{P})$ needs to be a constant diagonal matrix (see Eq.(C.7)). This allows us to write:

$$
\Lambda_T(\hat{P}) = \alpha\,\hat{P} + \beta\,\hat{Q} \underset{\hat{Q}=\mathbb{1}-\hat{P}}{=} (\alpha - \beta)\,\hat{P} + \beta\,\mathbb{1} = \gamma\,\hat{P} + \beta\,\mathbb{1}\,.
\tag{D.5}
$$

Note that these results are independent of $\hat{P}$. To see this, imagine picking a different $1D$ projector $\hat{P}' = \hat{U}\hat{P}\hat{U}^\dagger$. Then, by Eq.(D.1),

$$
\Lambda_T(\hat{P}') = \Lambda_T(\hat{U}\hat{P}\hat{U}^\dagger) = \hat{U}\Lambda_T(\hat{P})\hat{U}^\dagger = \gamma\,\hat{U}\hat{P}\hat{U}^\dagger + \beta\,\mathbb{1} = \gamma\,\hat{P}' + \beta\,\mathbb{1}\,.
\tag{D.6}
$$

Given that $\Lambda_T$ is a linear map, we have already all the required information to deduce its form for a generic input state $\rho$:

$$
\begin{aligned}
\rho = \sum_i \rho_i|\psi_i\rangle\langle\psi_i| &\implies \Lambda_T(\rho) = \sum_i \rho_i\,\Lambda_T(|\psi_i\rangle\langle\psi_i|) \\
\Lambda_T(\rho) &= \sum_i \rho_i\big(\gamma|\psi_i\rangle\langle\psi_i| + \beta\mathbb{1}\big) = \gamma\sum_i \rho_i|\psi_i\rangle\langle\psi_i| + \mathbb{1}\,\beta\sum_i \rho_i \\
\Lambda_T(\rho) &= \gamma\,\rho + \beta\,\mathbb{1}\,,
\end{aligned}
\tag{D.7}
$$

---

[27]The Haar measure is both left and right invariant under unitary transformations [18,19].

where we have assumed $\text{Tr}(\rho) = 1$. The values of the coefficients $\gamma$ and $\beta$ are subjected to the constraint that $\Lambda_T$ should be a CPTP map. This means that both $\gamma$ and $\beta$ need to be $\geq 0$, and

$$\text{Tr}(\Lambda_T(\rho)) = 1 \quad \Leftrightarrow \quad \gamma + \beta d = 1 \quad \Leftrightarrow \quad \gamma = 1 - \beta d \quad \Longrightarrow \quad \beta \leq \frac{1}{d}. \tag{D.8}$$

Therefore, we may write $\beta = \frac{(1-p)}{d}$, with $p \in [0,1]$. With this parametrization, we arrive at last to the depolarizing channel:

$$\Lambda_T(\rho) = p\,\rho + \frac{(1-p)}{d}\,\mathbb{1}. \tag{D.9}$$

Hence, statement 3 is fulfilled.

# E   Superoperators in the Liouville representation

The average sequence operator, $S_m$, is a linear operator that acts on the space of density matrices: It takes in a density matrix, performs some linear transformation on it, and returns back some other density matrix. Hence, it maps matrices into matrices. For this reason, $S_m$ can also be referred to as a superoperator, and we can defer the term operator to linear maps that send vectors onto vectors. Being a linear superoperator, it must be possible to express it as a matrix. A common way of providing an explicit matrix representation for superoperators is the Liouville representation, which makes explicit use of the Pauli basis. More precisely, just as we often seek an orthonormal basis in which we can express any vector state $|\psi\rangle$, we can also look for an orthonormal basis that will allow an explicit matrix realization of any given superoperator. For two vectors, the orthonormality condition is expressed in terms of the inner product: $\langle v_i | v_j \rangle = \delta_{i,j}$. The concept of inner product can be extended to operators through the Hilbert-Schmidt inner product [18], $\langle \hat{A}, \hat{B} \rangle = \text{Tr}(\hat{A}^\dagger \hat{B})$. We, therefore, seek an operator basis $\{B_j\}$ for which $\text{Tr}(\hat{B}_i^\dagger \hat{B}_j) = \delta_{i,j}$. The set of normalized $n-$qubit Pauli matrices, $\{\hat{\sigma}_0, \boldsymbol{\sigma}_n\}$, provides an explicit example for a such a basis:[28]

$$\begin{cases} \hat{\sigma}_0 = \frac{1}{\sqrt{d}}\mathbb{1}, \quad d \equiv 2^n, \\ \boldsymbol{\sigma}_n = \frac{1}{\sqrt{d}}\left\{\mathbb{1}, \hat{X}, \hat{Y}, \hat{Z}\right\}^{\otimes n} \setminus \{\hat{\sigma}_0\}, \end{cases} \tag{E.1}$$

as can be seen by the fact that, for any two Pauli matrices in the set $\{\hat{\sigma}_0, \boldsymbol{\sigma}_n\}$, the Hilber-Schmidt inner product satisfies the orthonormal condition,

$$\langle \sigma_i, \sigma_j \rangle = \text{Tr}\left(\sigma_i^\dagger \sigma_j\right) = \delta_{i,j}. \tag{E.2}$$

We can then use this basis to compute an explicit matrix form for a superoperator. To accomplish this, we pick any (normalized) Pauli matrix in the set $\{\hat{\sigma}_0, \boldsymbol{\sigma}_n\}$ and consider its vectorization $\hat{\sigma}_j \mapsto |\sigma_j\rangle\rangle$. For example, in the 1-qubit basis,

$$\hat{\sigma}_x = \frac{1}{\sqrt{2}}\begin{pmatrix} 0 & 1 \\ 1 & 0 \end{pmatrix} \quad \mapsto \quad |\sigma_x\rangle\rangle = \begin{pmatrix} 0 \\ \frac{1}{\sqrt{2}} \\ \frac{1}{\sqrt{2}} \\ 0 \end{pmatrix}. \tag{E.3}$$

Proceeding in this way allows us to map the orthonormal matrix basis to an orthonormal vector basis. Effectively, what we are doing with this mapping is to move from a $d$ dimension vector

---

[28]We are using the notation that the set $\{\mathbb{1}, \hat{X}, \hat{Y}, \hat{Z}\}$ corresponds to the usual $2 \times 2$ Pauli matrices.

space to a $d^2$ vector space, so that density matrices are reduced to vectors in the new space. Hence, a density matrix $\rho$ will be mapped onto the vector $|\rho\rangle\rangle$, which may equivalently be expressed in terms of the complete basis vectors[29] as:

$$|\rho\rangle\rangle = \sum_{\sigma_j \in \{\hat{\sigma}_0, \boldsymbol{\sigma}_n\}} \mathrm{Tr}\left(\sigma_j^\dagger \rho\right) |\sigma_j\rangle\rangle. \tag{E.4}$$

In the Liouville representation, the POVM operators are also mapped onto vectors. This allows us to express the probability of measuring a given outcome $a$, simply as:

$$\mathrm{Prob}(a) = \mathrm{Tr}(E_a \rho) = \langle\langle E_a | \rho \rangle\rangle. \tag{E.5}$$

A superoperator mapping density matrices to density matrices needs to be a matrix in this new vector space, since it needs to now map vectors to vectors. Hence, let $\mathcal{E}$ be one of these linear maps:

$$\hat{\mathbb{1}} = \sum_{\sigma_j \in \{\hat{\sigma}_0, \boldsymbol{\sigma}_n\}} |\sigma_i\rangle\rangle\langle\langle\sigma_i| \quad \text{(completeness relation)}$$

$$\implies \quad \mathcal{E} \mapsto \hat{\mathcal{E}} = \hat{\mathbb{1}}\,\hat{\mathcal{E}}\,\hat{\mathbb{1}}$$

$$= \sum_{\sigma_i, \sigma_j \in \{\hat{\sigma}_0, \boldsymbol{\sigma}_n\}} \underbrace{\langle\langle\sigma_i|\hat{\mathcal{E}}|\sigma_j\rangle\rangle}_{=\langle\langle\sigma_i|\mathcal{E}(\sigma_j)\rangle\rangle} |\sigma_i\rangle\rangle\langle\langle\sigma_j| \tag{E.6}$$

$$= \sum_{\sigma_i, \sigma_j \in \{\hat{\sigma}_0, \boldsymbol{\sigma}_n\}} \mathrm{Tr}(\sigma_i^\dagger \mathcal{E}(\sigma_j)) |\sigma_i\rangle\rangle\langle\langle\sigma_j|$$

$$\Leftrightarrow \quad \boxed{\hat{\mathcal{E}} = \sum_{\sigma_i, \sigma_j \in \{\hat{\sigma}_0, \boldsymbol{\sigma}_n\}} \left(\hat{\mathcal{R}}_\mathcal{E}\right)_{i,j} |\sigma_i\rangle\rangle\langle\langle\sigma_j|,}$$

where $\hat{\mathcal{R}}_\mathcal{E}$ is a matrix, known as the Pauli transfer matrix [4], and whose matrix elements we repeat below:

$$\boxed{\left(\hat{\mathcal{R}}_\mathcal{E}\right)_{i,j} \equiv \mathrm{Tr}\left(\sigma_i^\dagger \mathcal{E}\left(\sigma_j\right)\right).} \tag{E.7}$$

The matrix elements of the superoperator $\hat{\mathcal{E}}$ are then given by the matrix elements of the Pauli transfer matrix. One convenient property of working in this basis is that the composition of any two linear maps is transformed into the product of their superoperator counterparts:

$$(\mathcal{E}_1 \circ \mathcal{E}_2)(\rho) \mapsto |\mathcal{E}_1(\mathcal{E}_2(\rho))\rangle\rangle = |\mathcal{E}_1(\rho')\rangle\rangle = \hat{\mathcal{E}}_1|\rho'\rangle\rangle$$

$$= \sum_{\sigma_i, \sigma_j} \left(\hat{\mathcal{R}}_{\mathcal{E}_1}\right)_{i,j} |\sigma_i\rangle\rangle\langle\langle\sigma_j \underbrace{|\rho'\rangle\rangle}_{=\hat{\mathcal{E}}_2|\rho\rangle\rangle}$$

$$= \sum_{\sigma_i, \sigma_j} \left(\hat{\mathcal{R}}_{\mathcal{E}_1}\right)_{i,j} |\sigma_i\rangle\rangle\langle\langle\sigma_j| \left(\sum_{\sigma_l, \sigma_k} \left(\hat{\mathcal{R}}_{\mathcal{E}_2}\right)_{k,l} |\sigma_k\rangle\rangle\langle\langle\sigma_l|\rho\rangle\rangle\right)$$

$$= \sum_{\sigma_i, \sigma_j, \sigma_l, \sigma_k} \left(\hat{\mathcal{R}}_{\mathcal{E}_1}\right)_{i,j}\left(\hat{\mathcal{R}}_{\mathcal{E}_2}\right)_{k,l} |\sigma_i\rangle\rangle \underbrace{\langle\langle\sigma_j|\sigma_k\rangle\rangle}_{=\delta_{j,k}}\langle\langle\sigma_l|\rho\rangle\rangle \tag{E.8}$$

$$= \left(\sum_{\sigma_i, \sigma_j, \sigma_l} \left(\hat{\mathcal{R}}_{\mathcal{E}_1}\right)_{i,j}\left(\hat{\mathcal{R}}_{\mathcal{E}_2}\right)_{j,l} |\sigma_i\rangle\rangle\langle\langle\sigma_l|\right) |\rho\rangle\rangle$$

$$= \hat{\mathcal{E}}_1 \hat{\mathcal{E}}_2 |\rho\rangle\rangle \quad \therefore \quad \boxed{\mathcal{E}_1 \circ \mathcal{E}_2 \mapsto \hat{\mathcal{E}}_1 \hat{\mathcal{E}}_2.}$$

---

[29]If we were expressing a *normal* vector in an orthonormal basis, the coefficients could be equivalently expressed in terms of the inner product with the basis vectors, $|v\rangle = \sum_i c_i |e_i\rangle \implies \langle e_j|v\rangle = \sum_i c_i \langle e_j|e_i\rangle = c_i$. The same principle applies here, except we use the Hilbert-Schmidt inner product.

A channel that is of particular interest is the twirling channel. A natural question then becomes how to express it in the language of superoperators. Following the definition in Eq.(35), the twirling channel can be seen as a linear superposition of map compositions. Since map compositions become matrix products in the Liouville representation, we can immediately write:

$$
\begin{aligned}
\hat{\Lambda}_T &= \frac{1}{|\mathcal{G}|} \sum_{\mathcal{C} \in \mathcal{G}} \hat{\mathcal{C}}^\dagger \hat{\Lambda} \hat{\mathcal{C}} \\
&= \frac{1}{|\mathcal{G}|} \sum_{\mathcal{C} \in \mathcal{G}} \sum_{\sigma_i, \sigma_j, \sigma_k, \sigma_l, \sigma_m, \sigma_n} (\hat{\mathcal{R}}_{\mathcal{C}^\dagger})_{i,j} (\hat{\mathcal{R}}_\Lambda)_{k,l} (\hat{\mathcal{R}}_\mathcal{C})_{m,n} |\sigma_i\rangle\rangle \underbrace{\langle\langle\sigma_j|\sigma_k\rangle\rangle}_{\delta_{j,k}} \underbrace{\langle\langle\sigma_l|\sigma_m\rangle\rangle}_{\delta_{l,m}} \langle\langle\sigma_n| \\
&= \sum_{\sigma_i, \sigma_n} \bigg( \underbrace{\frac{1}{|\mathcal{G}|} \sum_{\mathcal{C} \in \mathcal{G}} \sum_{\sigma_j, \sigma_l} (\hat{\mathcal{R}}_{\mathcal{C}^\dagger})_{i,j} (\hat{\mathcal{R}}_\Lambda)_{j,l} (\hat{\mathcal{R}}_\mathcal{C})_{l,n}}_{\equiv (\hat{\mathcal{R}}_{\Lambda_T})_{i,n}} \bigg) |\sigma_i\rangle\rangle\langle\langle\sigma_n| \\
\Leftrightarrow \quad &\boxed{\hat{\Lambda}_T = \sum_{\sigma_i, \sigma_n} (\hat{\mathcal{R}}_{\Lambda_T})_{i,n} |\sigma_i\rangle\rangle\langle\langle\sigma_n|.}
\end{aligned}
\tag{E.9}
$$

All the important properties of the twirling map are hidden in the matrix elements $(\hat{\mathcal{R}}_{\Lambda_T})_{i,n}$. Note that these are given by of the matrix product of three Pauli transfer matrices, and entail the sum over all (Clifford) group elements. To simplify the matrix elements further, we need to make use of some fundamental results in group theory, as well as some properties of the Clifford group.

### E.0.1 The Clifford group and the implications to the twirling map

Let us then address how the twirling channel is simplified, when the underlying group is the Clifford group. With this goal in mind, we first go over the definitions of the Pauli and the Clifford group. Next, we address how the representations of the Clifford group are expressed in the Liouville representation, and then conclude on the implications to the twirling channel.

Let us start by defining the Pauli group. The $n-$qubit Pauli group, $\mathcal{P}_n$, is a subgroup of the unitary group, whose elements can be represented using the set of $2 \times 2$ Pauli matrices $\{\mathbb{1}, \hat{X}, \hat{Y}, \hat{Z}\}$ [16,56]:

$$
\hat{P} = i^k \, \hat{\mathsf{P}}_1 \otimes \hat{\mathsf{P}}_2 \otimes \cdots \otimes \hat{\mathsf{P}}_n \; \in \mathcal{P}_n, \qquad \text{with} \qquad k = \{0, 1, 2, 3\}, \quad \hat{\mathsf{P}}_i \in \{\mathbb{1}, \hat{X}, \hat{Y}, \hat{Z}\}, \tag{E.10}
$$

with the following properties [56]

1. The elements in the Pauli group square to $\pm\mathbb{1}$.

2. Any two elements in the Pauli group either commute, $[\hat{P}, \hat{P}'] = 0$, or anti-commute, $\{\hat{P}, \hat{P}'\} = 0$. Furthermore, every non-identity element commutes with exact half of the elements in the group, and anti-commutes with the remaining half.

3. Being a subgroup of the unitary group, the unitary condition is satisfied $\hat{P}\hat{P}^\dagger = \mathbb{1}$.

4. Only the elements $\hat{P} \propto \mathbb{1}^{\otimes n}$ have non-zero trace; all the remaining elements have a vanishing trace.

The Pauli group is also the normal subgroup of the Clifford group, and this property is the most defining characteristic of the Clifford group [14]:

$$
\mathcal{C}_n = \{\hat{C} \in U(2^n) \,|\, \hat{C}\hat{P}\hat{C}^\dagger \in \mathcal{P}_n\}, \tag{E.11}
$$

meaning the Clifford group is usually introduced as the subgroup of the unitary group that maps the Pauli group onto itself, via conjugation. In the definition above, $\mathcal{C}_n$ and $U(2^n)$ denote the $n-$qubit Clifford group and the $n-$qubit unitary group, respectively. The way that the Clifford group re-shuffles the elements in $\mathcal{P}_n$ is constrained by the fact that it needs to preserve the commutation relations between Paulis:

$$\left[\hat{C}\hat{P}\hat{C}^\dagger, \hat{C}\hat{P}'\hat{C}^\dagger\right] = \hat{C}\left[\hat{P}, \hat{P}'\right]\hat{C}^\dagger, \quad \text{and} \quad \left\{\hat{C}\hat{P}\hat{C}^\dagger, \hat{C}\hat{P}'\hat{C}^\dagger\right\} = \hat{C}\left\{\hat{P}, \hat{P}'\right\}\hat{C}^\dagger, \tag{E.12}$$

and also the trace:

$$\hat{P}' = \hat{C}\hat{P}\hat{C}^\dagger \quad \implies \quad \text{Tr}\left(\hat{P}'\right) = \text{Tr}\left(\hat{P}\right). \tag{E.13}$$

Thus, given a pair of commuting Paulis, the elements in $\mathcal{C}_n$ are required to map it onto another commuting pair, and likewise for an anti-commuting pair. Consequently, for any two distinct non-identity Paulis ($\hat{P}_k \neq \hat{P}_i$), there will always be a $\hat{C} \in \mathcal{C}_n$ for which $\hat{C}\hat{P}_i\hat{C}^\dagger = \hat{P}_i$ and $\hat{C}\hat{P}_k\hat{C}^\dagger = -\hat{P}_k$ (or vice-versa). Of course, via the same argument, there will also be an element $\hat{C} \in \mathcal{C}_n$ for which $\hat{C}\hat{P}_i\hat{C}^\dagger = \pm\hat{P}_i$ and $\hat{C}\hat{P}_k\hat{C}^\dagger = \pm\hat{P}_k$. Furthermore, since only the identity (and related) has non-zero trace, a non-identity Pauli will never be mapped to the identity.

The set formed by the vectorization of the normalized $n-$qubit Pauli matrices $\{\hat{\sigma}_0, \boldsymbol{\sigma}_n\}$ is an orthornormal basis for the Hilbert space $\mathcal{H}_{d^2}$. Envisioning the Clifford elements as linear operations on density matrices, makes it reasonable to see them as superoperators in $\mathcal{H}_{d^2}$. This allows us to write:

$$\hat{\mathcal{C}} = \sum_{\sigma_i, \sigma_j \in \{\hat{\sigma}_0, \boldsymbol{\sigma}_n\}} \left(\hat{\mathcal{R}}_C\right)_{i,j} |\sigma_i\rangle\rangle\langle\langle\sigma_j|, \tag{E.14}$$

with

$$\begin{aligned}\left(\hat{\mathcal{R}}_C\right)_{i,j} &= \text{Tr}\left(\sigma_i^\dagger C(\sigma_j)\right) \\ &= \text{Tr}\left(\hat{\sigma}_i^\dagger \hat{C}\hat{\sigma}_j\hat{C}^\dagger\right).\end{aligned} \tag{E.15}$$

Note that all the relevant group properties are contained in the Pauli transfer matrix $\hat{\mathcal{R}}_C$; the term $|\sigma_i\rangle\rangle\langle\langle\sigma_j|$ simply supplies a basis for the matrix representation. Thus, we can look at $\hat{\mathcal{R}}_C$ as the effective representation of the Clifford group. If $\hat{\mathcal{R}}_C$ is a suitable representation, then $\hat{\mathcal{R}}_{C^\dagger} = \hat{\mathcal{R}}_C^\dagger$,[30] and if this holds true it should follow that $\hat{\mathcal{R}}_C\hat{\mathcal{R}}_{C^\dagger} = \mathbb{1}$, which we verify below:

$$\begin{aligned}\left(\hat{\mathcal{R}}_C\hat{\mathcal{R}}_{C^\dagger}\right)_{i,l} &= \sum_{\sigma_k}\text{Tr}\left(\sigma_i^\dagger C(\sigma_k)\right)\text{Tr}\left(\sigma_k^\dagger C^\dagger(\sigma_l)\right) \\ &= \sum_{\sigma_k}\text{Tr}\left(\hat{\sigma}_i^\dagger \hat{C}\hat{\sigma}_k\hat{C}^\dagger\right)\text{Tr}\left(\hat{\sigma}_k^\dagger \hat{C}^\dagger\hat{\sigma}_l\hat{C}\right) = \delta_{i,l} \\ &\implies \quad \hat{\mathcal{R}}_C\hat{\mathcal{R}}_{C^\dagger} = \mathbb{1}.\end{aligned} \tag{E.16}$$

The result follows from the fact that the only way the first trace is non-zero is if $\hat{C}\hat{\sigma}_k\hat{C}^\dagger = \hat{\sigma}_i \leftrightarrow \hat{\sigma}_k^\dagger = \hat{C}^\dagger\hat{\sigma}_i^\dagger\hat{C} \implies \text{Tr}\left(\hat{\sigma}_k^\dagger \hat{C}^\dagger\hat{\sigma}_l\hat{C}\right) = \text{Tr}\left(\hat{\sigma}_i^\dagger\hat{\sigma}_l\right) = \delta_{i,l}$. Another property we require from the representation $\hat{\mathcal{R}}_C$ is that $\hat{\mathcal{R}}_{C_2 C_1} = \hat{\mathcal{R}}_{C_2}\hat{\mathcal{R}}_{C_1}$:

$$\begin{aligned}\left(\hat{\mathcal{R}}_{C_2 C_1}\right)_{i,j} &= \text{Tr}\left(\sigma_i^\dagger C_2\left(C_1\sigma_j C_1^\dagger\right)\right) \\ &= \text{Tr}\left(\hat{\sigma}_i^\dagger \hat{C}_2\hat{C}_1\hat{\sigma}_j\hat{C}_1^\dagger\hat{C}_2^\dagger\right) \\ &= \text{Tr}\left(\hat{C}_2^\dagger\hat{\sigma}_i^\dagger \hat{C}_2\hat{C}_1\hat{\sigma}_j\hat{C}_1^\dagger\right),\end{aligned} \tag{E.17}$$

$$\begin{aligned}\left(\hat{\mathcal{R}}_{C_2}\hat{\mathcal{R}}_{C_1}\right)_{i,j} &= \sum_{\sigma_k}\underbrace{\text{Tr}\left(\hat{\sigma}_i^\dagger \hat{C}_2\hat{\sigma}_k\hat{C}_2^\dagger\right)}_{=\delta_{\sigma_k, C_2^\dagger\sigma_i C_2}}\text{Tr}\left(\hat{\sigma}_k^\dagger \hat{C}_1\hat{\sigma}_j\hat{C}_1^\dagger\right) \\ &= \text{Tr}\left(\hat{C}_2^\dagger\hat{\sigma}_i^\dagger \hat{C}_2\hat{C}_1\hat{\sigma}_j\hat{C}_1^\dagger\right) = \left(\hat{\mathcal{R}}_{C_2 C_1}\right)_{i,j}.\end{aligned} \tag{E.18}$$

---

[30]This is just the requirement that the representation should keep the group structure, $\varphi(g^{-1}) = \varphi^{-1}(g)$.

Now that we have motivated the Pauli transfer matrices as representations of the Clifford group, we can more easily establish a relation between $\hat{\mathcal{R}}_{\Lambda_T}$ and Schur's lemma (2) (Eq.(C.7)). Let us first prove that this matrix commutes with any arbitrary representation of the Clifford group, $\hat{\mathcal{R}}_C$:

$$
\begin{aligned}
\hat{\mathcal{R}}_{\Lambda_T} pt &= \frac{1}{|\mathcal{G}|} \sum_{\mathcal{C} \in \mathcal{G}} \hat{\mathcal{R}}_C^\dagger \hat{\mathcal{R}}_\Lambda \hat{\mathcal{R}}_{\mathcal{C}}, \\
\hat{\mathcal{R}}_{\tilde{C}} \hat{\mathcal{R}}_{\Lambda_T} pt &= \frac{1}{|\mathcal{G}|} \sum_{\mathcal{C} \in \mathcal{G}} \underbrace{\hat{\mathcal{R}}_{\tilde{C}} \hat{\mathcal{R}}_C^\dagger}_{=\hat{\mathcal{R}}_{\tilde{C} C^\dagger}} \hat{\mathcal{R}}_\Lambda \hat{\mathcal{R}}_{\mathcal{C}} \\
&\underset{\tilde{C} C^\dagger := (C')^\dagger}{=} \frac{1}{|\mathcal{G}|} \sum_{\mathcal{C}' \in \mathcal{G}} \hat{\mathcal{R}}_{C'}^\dagger \hat{\mathcal{R}}_\Lambda \hat{\mathcal{R}}_{C'\tilde{C}} \\
&= \frac{1}{|\mathcal{G}|} \sum_{\mathcal{C}' \in \mathcal{G}} \hat{\mathcal{R}}_{C'}^\dagger \hat{\mathcal{R}}_\Lambda \hat{\mathcal{R}}_{C'} \, \hat{\mathcal{R}}_{\tilde{C}} = \hat{\mathcal{R}}_{\Lambda_T} \hat{\mathcal{R}}_{\tilde{C}}.
\end{aligned}
\tag{E.19}
$$

Indeed, $\hat{\mathcal{R}}_{\Lambda_T}$ is a matrix that commutes with every representation of the Clifford group. Note that this commutation follows from general group requirements, and not on more strict constraints requiring the representations to be irreducible. Then, assuming the generic case were $\hat{\mathcal{R}}_C$ is a reducible representation, it follows immediately from Eq.(C.7) that $\hat{\mathcal{R}}_{\Lambda_T}$ is a diagonal matrix, of the form:

$$
\hat{\mathcal{R}}_{\Lambda_T} = \frac{1}{|\mathcal{G}|} \sum_{\mathcal{C} \in \mathcal{G}} \hat{\mathcal{R}}_C^\dagger \hat{\mathcal{R}}_\Lambda \hat{\mathcal{R}}_{\mathcal{C}} = \begin{pmatrix} c_1 \mathbb{1}_1 & \hat{0} & \cdots & \hat{0} \\ \hat{0} & c_2 \mathbb{1}_2 & \cdots & \hat{0} \\ \vdots & \vdots & \ddots & \vdots \\ \hat{0} & \hat{0} & \cdots & c_m \mathbb{1}_m \end{pmatrix} = \sum_j c_j \, \mathbb{P}_j.
\tag{E.20}
$$

Recall that $\mathbb{1}_j$ is a short hand notation for a identity matrix on the reduced space of the corresponding irreducible representation $j$. This, in turn, allows the matrix to be written using the projectors onto the corresponding invariant subspaces ($\mathbb{P}_j$). We can re-express the constants $c_j$ in a similar fashion as in Ref. [4], by taking the trace on both sides of Eq.(E.20):

$$
\begin{aligned}
\mathrm{Tr}\left(\hat{\mathcal{R}}_{\Lambda_T}\right) &= \frac{1}{|\mathcal{G}|} \sum_{\mathcal{C} \in \mathcal{G}} \mathrm{Tr}\left(\hat{\mathcal{R}}_\Lambda \hat{\mathcal{R}}_{\mathcal{C}} \hat{\mathcal{R}}_C^\dagger\right) = \sum_j c_j \mathrm{Tr}\left(\mathbb{P}_j\right) \\
\Leftrightarrow \quad \mathrm{Tr}(\hat{\mathcal{R}}_\Lambda \underbrace{\mathbb{1}}_{=\sum_j \mathbb{P}_j}) &= \sum_j c_j \mathrm{Tr}\left(\mathbb{P}_j\right) \\
\Leftrightarrow \quad \sum_j \Big(\mathrm{Tr}\left(\hat{\mathcal{R}}_\Lambda \mathbb{P}_j\right) &- c_j \mathrm{Tr}\left(\mathbb{P}_j\right)\Big) = 0 \\
\implies \quad c_j &= \frac{\mathrm{Tr}\left(\hat{\mathcal{R}}_\Lambda \mathbb{P}_j\right)}{\mathrm{Tr}\left(\mathbb{P}_j\right)}.
\end{aligned}
\tag{E.21}
$$

Hence, the Pauli transfer matrix for the twirling channel, over the Clifford group is given by:

$$
\boxed{\hat{\mathcal{R}}_{\Lambda_T} = \sum_j \frac{\mathrm{Tr}\left(\hat{\mathcal{R}}_\Lambda \mathbb{P}_j\right)}{\mathrm{Tr}\left(\mathbb{P}_j\right)} \mathbb{P}_j.}
\tag{E.22}
$$

## F Action of the single-qubit Clifford group on the Pauli group

In Eq.(46), the twirling operator is defined with respect to the direct product group $\mathcal{I} \times \mathcal{C}_1$. When averaging over all the elements of this group, the only surviving matrix elements for

the Pauli transfer matrix of the twirling channel are those in which the Clifford elements are applied to the same Pauli element (see Eq.(48)). This has to do with the fact that the Cliffords act like a set of signed permutations over the Pauli group. Here we complement this argument by explicitly computing the matrix elements in Eq.(46). To simplify notation, let us define:

$$\text{Tr}\left[\left(\hat{\sigma}_i \otimes \left(\hat{C}\hat{\sigma}_m\hat{C}^\dagger\right)\right)\Lambda\left(\hat{\sigma}_k \otimes \left(\hat{C}\hat{\sigma}_n\hat{C}^\dagger\right)\right)\right] = \text{Tr}\left[\left(\hat{\sigma}_i \otimes \hat{\sigma}_j\right)\Lambda\left(\hat{\sigma}_k \otimes \hat{\sigma}_l\right)\right] = f_{j,l}, \qquad \text{(F.1)}$$

with $\hat{C} \in \mathcal{C}_1$, and taking $\sigma_i$ and $\sigma_k$ to be fixed. We then compute all possible values of the $f_{j,l}$, and list them in the tables below. This requires not only evaluating $f_{j,l}$ for all elements in $\mathcal{C}_1$, but also for all possible pairs of Paulis $(\sigma_m, \sigma_n)$. In the tables below, the last column represents the sum over a subset of Clifford elements, for a fixed pair $(\sigma_m, \sigma_n)$. All the other entries in the tables correspond to the matrix elements arising from the action of specific Cliffords on the specified pair $(\sigma_m, \sigma_n)$. For example, located on the second line and second column of table 6 is the end result of the matrix element $\text{Tr}\left[\left(\hat{\sigma}_i \otimes \left(\hat{H}\sigma_0\hat{H}^\dagger\right)\right)\Lambda\left(\hat{\sigma}_k \otimes \left(\hat{H}\hat{\sigma}_1\hat{H}^\dagger\right)\right)\right]$, where $H$ is the Hadamard gate. This gate maps $\sigma_0$ to itself, while sending $\sigma_1$ to $\sigma_3$, so the resulting matrix element is $f_{0,3}$, which is the result displayed in the table. Additionally, located in the first line and last column of table 6 is the sum of all values of $f_{j,l}$ along this line, evaluated for fixed $\sigma_m = \sigma_n = \sigma_0$ and for the subset of Cliffords $\{I, H, S, HS, SH, SS, HSH, HSS, SHS, SSH, SSS, HSHS\}$. All tables follow this notation. The matrix elements $\left(\hat{\mathcal{R}}_T\right)_{ij,kl}$ are then given in terms of the $f_{j,l}$ as:

$$\left(\hat{\mathcal{R}}_T\right)_{ij,kl} = \frac{1}{|\mathcal{C}_1|}\left(\sum f_{j,l}\right)_{\text{Table A}} + \frac{1}{|\mathcal{C}_1|}\left(\sum f_{j,l}\right)_{\text{Table B}} + \frac{1}{|\mathcal{C}_1|}\left(\sum f_{j,l}\right)_{\text{Table C}}$$

$$= \begin{cases} f_{0,0}, & j = l = 0, \\ \frac{1}{3}\left(f_{1,1} + f_{2,2} + f_{3,3}\right), & (j = l) \neq 0, \\ 0, & \text{otherwise}, \end{cases} \qquad \text{(F.2)}$$

which is precisely Eq.(48).

Table 6: Resulting matrix elements $f_{j,l}$, evaluated for the subset of Cliffords displayed in the columns. Each line corresponds to a different choice for the pair $(\sigma_m, \sigma_n)$.

| | $I$ | $H$ | $S$ | $HS$ | $SH$ | $SS$ | $HSH$ | $HSS$ | $SHS$ | $SSH$ | $SSS$ | $HSHS$ | $\sum f_{j,l}$ |
|---|---|---|---|---|---|---|---|---|---|---|---|---|---|
| $(\sigma_0,\sigma_0)$ | $f_{0,0}$ | $f_{0,0}$ | $f_{0,0}$ | $f_{0,0}$ | $f_{0,0}$ | $f_{0,0}$ | $f_{0,0}$ | $f_{0,0}$ | $f_{0,0}$ | $f_{0,0}$ | $f_{0,0}$ | $f_{0,0}$ | $= 12\,f_{0,0}$ |
| $(\sigma_0,\sigma_x)$ | $f_{0,1}$ | $f_{0,3}$ | $f_{0,2}$ | $-f_{0,2}$ | $f_{0,3}$ | $-f_{0,1}$ | $f_{0,1}$ | $-f_{0,3}$ | $f_{0,1}$ | $f_{0,3}$ | $-f_{0,2}$ | $f_{0,3}$ | $= 3f_{0,3}\ +2f_{0,1}-f_{0,2}$ |
| $(\sigma_0,\sigma_y)$ | $f_{0,2}$ | $-f_{0,2}$ | $-f_{0,1}$ | $-f_{0,3}$ | $f_{0,1}$ | $-f_{0,2}$ | $f_{0,3}$ | $f_{0,2}$ | $-f_{0,3}$ | $f_{0,2}$ | $f_{0,1}$ | $-f_{0,1}$ | $= f_{0,2} - f_{0,3}$ |
| $(\sigma_0,\sigma_z)$ | $f_{0,3}$ | $f_{0,1}$ | $f_{0,3}$ | $f_{0,1}$ | $f_{0,2}$ | $f_{0,3}$ | $-f_{0,2}$ | $f_{0,1}$ | $f_{0,2}$ | $-f_{0,1}$ | $f_{0,3}$ | $-f_{0,2}$ | $= 4f_{0,3} + 2f_{0,1}$ |
| $(\sigma_x,\sigma_0)$ | $f_{1,0}$ | $f_{3,0}$ | $f_{2,0}$ | $-f_{2,0}$ | $f_{3,0}$ | $-f_{1,0}$ | $f_{1,0}$ | $-f_{3,0}$ | $f_{1,0}$ | $f_{3,0}$ | $-f_{2,0}$ | $f_{3,0}$ | $= 3f_{3,0}\ +2f_{1,0}-f_{2,0}$ |
| $(\sigma_x,\sigma_x)$ | $f_{1,1}$ | $f_{3,3}$ | $f_{2,2}$ | $f_{2,2}$ | $f_{3,3}$ | $f_{1,1}$ | $f_{1,1}$ | $f_{3,3}$ | $f_{1,1}$ | $f_{3,3}$ | $f_{2,2}$ | $f_{3,3}$ | $= 4f_{1,1}\ +5f_{3,3}+3f_{2,2}$ |
| $(\sigma_x,\sigma_y)$ | $f_{1,2}$ | $-f_{3,2}$ | $-f_{2,1}$ | $f_{2,3}$ | $f_{3,1}$ | $f_{1,2}$ | $f_{1,3}$ | $-f_{3,2}$ | $-f_{1,3}$ | $f_{3,2}$ | $-f_{2,1}$ | $-f_{3,1}$ | $= 2f_{1,2} - f_{3,2}\ -2f_{2,1}+f_{2,3}$ |
| $(\sigma_x,\sigma_z)$ | $f_{1,3}$ | $f_{3,1}$ | $f_{2,3}$ | $-f_{2,1}$ | $f_{3,2}$ | $-f_{1,3}$ | $-f_{1,2}$ | $-f_{3,1}$ | $f_{1,2}$ | $-f_{3,1}$ | $-f_{2,3}$ | $-f_{3,2}$ | $= -f_{2,1} - f_{3,1}$ |
| $(\sigma_y,\sigma_0)$ | $f_{2,0}$ | $-f_{2,0}$ | $-f_{1,0}$ | $-f_{3,0}$ | $f_{1,0}$ | $-f_{2,0}$ | $f_{3,0}$ | $f_{2,0}$ | $-f_{3,0}$ | $f_{2,0}$ | $f_{1,0}$ | $-f_{1,0}$ | $= f_{2,0} - f_{3,0}$ |
| $(\sigma_y,\sigma_x)$ | $f_{2,1}$ | $-f_{2,3}$ | $-f_{1,2}$ | $f_{3,2}$ | $f_{1,3}$ | $f_{2,1}$ | $f_{3,1}$ | $-f_{2,3}$ | $-f_{3,1}$ | $f_{2,3}$ | $-f_{1,2}$ | $-f_{1,3}$ | $= 2f_{2,1} - 2f_{1,2}\ f_{3,2} - f_{2,3}$ |
| $(\sigma_y,\sigma_y)$ | $f_{2,2}$ | $f_{2,2}$ | $f_{1,1}$ | $f_{3,3}$ | $f_{1,1}$ | $f_{2,2}$ | $f_{3,3}$ | $f_{2,2}$ | $f_{3,3}$ | $f_{2,2}$ | $f_{1,1}$ | $f_{1,1}$ | $= 5f_{2,2} + 4f_{1,1}\ +3f_{3,3}$ |
| $(\sigma_y,\sigma_z)$ | $f_{2,3}$ | $-f_{2,1}$ | $-f_{1,3}$ | $-f_{3,1}$ | $f_{1,2}$ | $-f_{2,3}$ | $-f_{3,2}$ | $f_{2,1}$ | $-f_{3,2}$ | $-f_{2,1}$ | $f_{1,3}$ | $f_{1,2}$ | $= -f_{3,1}\ -f_{2,1}\ +2(f_{1,2} - f_{3,2})$ |
| $(\sigma_z,\sigma_0)$ | $f_{3,0}$ | $f_{1,0}$ | $f_{3,0}$ | $f_{1,0}$ | $f_{2,0}$ | $f_{3,0}$ | $-f_{2,0}$ | $f_{1,0}$ | $f_{2,0}$ | $-f_{1,0}$ | $f_{3,0}$ | $-f_{2,0}$ | $= 2(f_{1,0}\ +2f_{3,0})$ |
| $(\sigma_z,\sigma_x)$ | $f_{3,1}$ | $f_{1,3}$ | $f_{3,2}$ | $-f_{1,2}$ | $f_{2,3}$ | $-f_{3,1}$ | $-f_{2,1}$ | $-f_{1,3}$ | $f_{2,1}$ | $-f_{1,3}$ | $-f_{3,2}$ | $-f_{2,3}$ | $= -f_{1,2}\ -f_{1,3}$ |
| $(\sigma_z,\sigma_y)$ | $f_{3,2}$ | $-f_{1,2}$ | $-f_{3,1}$ | $-f_{1,3}$ | $f_{2,1}$ | $-f_{3,2}$ | $-f_{2,3}$ | $f_{1,2}$ | $-f_{2,3}$ | $-f_{1,2}$ | $f_{3,1}$ | $f_{2,1}$ | $= 2(f_{2,1} - f_{1,2})\ -f_{1,3} - 2f_{2,3}$ |
| $(\sigma_z,\sigma_z)$ | $f_{3,3}$ | $f_{1,1}$ | $f_{3,3}$ | $f_{1,1}$ | $f_{2,2}$ | $f_{3,3}$ | $f_{2,2}$ | $f_{1,1}$ | $f_{2,2}$ | $f_{1,1}$ | $f_{3,3}$ | $f_{2,2}$ | $= 4(f_{1,1} + f_{2,2}\ +f_{3,3})$ |

Table 7: Resulting matrix elements $f_{j,l}$, evaluated for the subset of Cliffords $\{HSSHHSSS, SHSS, SSHS, HSHSS, HSSHS\}$. Each line corresponds to a different choice for the pair $(\sigma_m, \sigma_n)$.

| | $HSSH$ | $HSSS$ | $SHSS$ | $SSHS$ | $HSHSS$ | $HSSHS$ | $\sum f_{j,l}$ |
|---|---|---|---|---|---|---|---|
| $(\sigma_0,\sigma_0)$ | $f_{0,0}$ | $f_{0,0}$ | $f_{0,0}$ | $f_{0,0}$ | $f_{0,0}$ | $f_{0,0}$ | $= 6f_{0,0}$ |
| $(\sigma_0,\sigma_x)$ | $f_{0,1}$ | $f_{0,2}$ | $-f_{0,3}$ | $f_{0,2}$ | $-f_{0,1}$ | $-f_{0,2}$ | $= f_{0,2} - f_{0,3}$ |
| $(\sigma_0,\sigma_y)$ | $-f_{0,1}$ | $f_{0,3}$ | $-f_{0,1}$ | $-f_{0,3}$ | $-f_{0,3}$ | $-f_{0,1}$ | $= -3f_{0,1} - f_{0,3}$ |
| $(\sigma_0,\sigma_z)$ | $-f_{0,3}$ | $f_{0,1}$ | $f_{0,2}$ | $-f_{0,1}$ | $-f_{0,2}$ | $-f_{0,3}$ | $= -2f_{0,3}$ |
| $(\sigma_x,\sigma_0)$ | $f_{1,0}$ | $f_{2,0}$ | $-f_{3,0}$ | $f_{2,0}$ | $-f_{1,0}$ | $-f_{2,0}$ | $= f_{2,0} - f_{3,0}$ |
| $(\sigma_x,\sigma_x)$ | $f_{1,1}$ | $f_{2,2}$ | $f_{3,3}$ | $f_{2,2}$ | $f_{1,1}$ | $f_{2,2}$ | $= 2f_{1,1} + 3f_{2,2} + f_{3,3}$ |
| $(\sigma_x,\sigma_y)$ | $-f_{1,2}$ | $f_{2,3}$ | $f_{3,1}$ | $-f_{2,3}$ | $f_{1,3}$ | $f_{2,1}$ | $= f_{2,1} - f_{1,2} + f_{1,3} + f_{3,1}$ |
| $(\sigma_x,\sigma_z)$ | $-f_{1,3}$ | $f_{2,1}$ | $-f_{3,2}$ | $-f_{2,1}$ | $f_{1,2}$ | $f_{2,3}$ | $= f_{1,2} - f_{1,3} + f_{2,3} - f_{3,2}$ |
| $(\sigma_y,\sigma_0)$ | $-f_{2,0}$ | $f_{3,0}$ | $-f_{1,0}$ | $-f_{3,0}$ | $-f_{3,0}$ | $-f_{1,0}$ | $= -2f_{1,0} - f_{3,0} - f_{2,0}$ |
| $(\sigma_y,\sigma_x)$ | $-f_{2,1}$ | $f_{3,2}$ | $f_{1,3}$ | $-f_{3,2}$ | $f_{3,1}$ | $f_{1,2}$ | $= f_{1,2} - f_{2,1} + f_{1,3} + f_{3,1}$ |
| $(\sigma_y,\sigma_y)$ | $f_{2,2}$ | $f_{3,3}$ | $f_{1,1}$ | $f_{3,3}$ | $f_{3,3}$ | $f_{1,1}$ | $= f_{2,2} + 2f_{1,1} + 3f_{3,3}$ |
| $(\sigma_y,\sigma_z)$ | $f_{2,3}$ | $f_{3,1}$ | $-f_{1,2}$ | $f_{3,1}$ | $f_{3,2}$ | $f_{1,3}$ | $= 2f_{3,1} + f_{2,3} - f_{1,2} + f_{3,2} + f_{1,3}$ |
| $(\sigma_z,\sigma_0)$ | $-f_{3,0}$ | $f_{1,0}$ | $f_{2,0}$ | $-f_{1,0}$ | $-f_{2,0}$ | $-f_{3,0}$ | $= -2f_{3,0}$ |
| $(\sigma_z,\sigma_x)$ | $-f_{3,1}$ | $f_{1,2}$ | $-f_{2,3}$ | $-f_{1,2}$ | $f_{2,1}$ | $f_{3,2}$ | $= -f_{3,1} - f_{2,3} + f_{2,1} + f_{3,2}$ |
| $(\sigma_z,\sigma_y)$ | $f_{3,2}$ | $f_{1,3}$ | $-f_{2,1}$ | $f_{1,3}$ | $-f_{2,1}$ | $f_{3,1}$ | $= -f_{3,1} + f_{2,3} + f_{1,2} - f_{1,3} - f_{2,1} - f_{3,2}$ |
| $(\sigma_z,\sigma_z)$ | $f_{3,3}$ | $f_{1,1}$ | $f_{2,2}$ | $f_{1,1}$ | $f_{2,2}$ | $f_{3,3}$ | $= 2(f_{1,1} + f_{2,2} + f_{3,3})$ |

Table 8: Resulting matrix elements $f_{j,l}$, evaluated for the subset of Cliffords $\{SHSSH, SHSSS, SSHSS, HSHSSH, HSHSSS, HSSHSS\}$. Each line corresponds to a different choice for the pair $(\sigma_m, \sigma_n)$.

| | SHSSH | SHSSS | SSHSS | HSHSSH | HSHSSS | HSSHSS | $\sum f_{j,l}$ |
|---|---|---|---|---|---|---|---|
| $(\sigma_0, \sigma_0)$ | $f_{0,0}$ | $f_{0,0}$ | $f_{0,0}$ | $f_{0,0}$ | $f_{0,0}$ | $f_{0,0}$ | $= 6 f_{0,0}$ |
| $(\sigma_0, \sigma_x)$ | $f_{0,2}$ | $-f_{0,1}$ | $-f_{0,3}$ | $-f_{0,2}$ | $-f_{0,3}$ | $-f_{0,1}$ | $= -2(f_{0,1} + f_{0,3})$ |
| $(\sigma_0, \sigma_y)$ | $f_{0,1}$ | $f_{0,3}$ | $-f_{0,2}$ | $f_{0,3}$ | $f_{0,1}$ | $f_{0,2}$ | $= 2(f_{0,1} + f_{0,3})$ |
| $(\sigma_0, \sigma_z)$ | $-f_{0,3}$ | $f_{0,2}$ | $-f_{0,1}$ | $-f_{0,1}$ | $-f_{0,2}$ | $-f_{0,3}$ | $= -2(f_{0,1} + f_{0,3})$ |
| $(\sigma_x, \sigma_0)$ | $f_{2,0}$ | $-f_{1,0}$ | $-f_{3,0}$ | $-f_{2,0}$ | $-f_{3,0}$ | $-f_{1,0}$ | $= -2(f_{1,0} + f_{3,0})$ |
| $(\sigma_x, \sigma_x)$ | $f_{2,2}$ | $f_{1,1}$ | $f_{3,3}$ | $f_{2,2}$ | $f_{3,3}$ | $f_{1,1}$ | $= 2(f_{1,1} + f_{2,2} + f_{3,3})$ |
| $(\sigma_x, \sigma_y)$ | $f_{2,1}$ | $-f_{1,3}$ | $f_{3,2}$ | $-f_{2,3}$ | $-f_{3,1}$ | $-f_{1,2}$ | $= f_{2,1} - f_{1,2} - f_{1,3} - f_{3,1} + f_{3,2} - f_{2,3}$ |
| $(\sigma_x, \sigma_z)$ | $-f_{2,3}$ | $-f_{1,2}$ | $f_{3,1}$ | $f_{2,1}$ | $f_{3,2}$ | $f_{1,3}$ | $= f_{1,3} + f_{3,1} + f_{2,1} - f_{1,2} + f_{3,2} - f_{2,3}$ |
| $(\sigma_y, \sigma_0)$ | $f_{1,0}$ | $f_{3,0}$ | $-f_{2,0}$ | $f_{3,0}$ | $f_{1,0}$ | $f_{2,0}$ | $= 2(f_{1,0} + f_{3,0})$ |
| $(\sigma_y, \sigma_x)$ | $f_{1,2}$ | $-f_{3,1}$ | $f_{2,3}$ | $-f_{3,2}$ | $-f_{1,3}$ | $-f_{2,1}$ | $= f_{1,2} - f_{2,1} + f_{2,3} - f_{3,2} - f_{3,1} - f_{1,3}$ |
| $(\sigma_y, \sigma_y)$ | $f_{1,1}$ | $f_{3,3}$ | $f_{2,2}$ | $f_{3,3}$ | $f_{1,1}$ | $f_{2,2}$ | $= 2(f_{1,1} + f_{2,2} + f_{3,3})$ |
| $(\sigma_y, \sigma_z)$ | $-f_{1,3}$ | $f_{3,2}$ | $f_{2,1}$ | $-f_{3,1}$ | $-f_{1,2}$ | $-f_{2,3}$ | $= f_{2,1} - f_{1,2} + f_{3,2} - f_{2,3} - f_{3,1} - f_{1,3}$ |
| $(\sigma_z, \sigma_0)$ | $-f_{3,0}$ | $f_{2,0}$ | $-f_{1,0}$ | $-f_{1,0}$ | $-f_{2,0}$ | $-f_{3,0}$ | $= -2(f_{1,0} + f_{3,0})$ |
| $(\sigma_z, \sigma_x)$ | $-f_{3,2}$ | $-f_{2,1}$ | $f_{1,3}$ | $f_{1,2}$ | $f_{2,3}$ | $f_{3,1}$ | $= f_{3,1} + f_{1,3} + f_{2,3} - f_{3,2} + f_{1,2} - f_{2,1}$ |
| $(\sigma_z, \sigma_y)$ | $-f_{3,1}$ | $f_{2,3}$ | $f_{1,2}$ | $-f_{1,3}$ | $-f_{2,1}$ | $-f_{3,2}$ | $= f_{3,1} + f_{1,3} + f_{2,3} - f_{3,2} + f_{1,2} - f_{2,1}$ |
| $(\sigma_z, \sigma_z)$ | $f_{3,3}$ | $f_{2,2}$ | $f_{1,1}$ | $f_{1,1}$ | $f_{2,2}$ | $f_{3,3}$ | $= 2(f_{1,1} + f_{2,2} + f_{3,3})$ |

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
