# Peer review of "Hands-on Introduction to Randomized Benchmarking"

_SciPost Physics Lecture Notes, doi:SciPost Phys. Lect. Notes 97 (2025)_

## Round 1 · Referee Report · Anonymous (Referee 1) · 2025-2-27

Strengths

• The authors have made commendable efforts to ensure the material is accessible. They clearly state their goal of a “pedagogical introduction to RB” and take a “hands-on, user-friendly approach” to the topic . Key concepts are introduced patiently, and all essential mathematical derivations are included and explained in full, as the authors intended to make them “as clear and transparent as possible.” This step by step explanatory style is very effective for readers learning the subject from scratch.
• The paper covers four core RB variants – standard, simultaneous, correlated, and interleaved RB – which were wisely chosen. Standard and interleaved RB are widely used in practice, while simultaneous and correlated RB address crosstalk and correlated errors in multi qubit systems, a pressing issue in current quantum processors . By focusing on these foundational and frequently-used methods , the tutorial gives readers a well rounded understanding of RB’s most important applications. Each variant is presented with its motivation, theoretical model, and practical considerations, providing a comprehensive overview of the RB family of protocols.
•A major strength is the integration of theoretical discussion with practical implementation. The inclusion of interactive Jupyter notebooks for each chapter is a fantastic feature . This allows readers to experiment with RB protocols hands on generating random Clifford sequences, simulating noisy gates, fitting decay curves, etc. which greatly enhances learning. The text points to these notebooks at appropriate times (for example, providing GitLab links for the standard RB examples ), ensuring that readers can seamlessly move from reading theory to running simulations. This dual approach caters to both deep theoretical understanding and practical skills development.
• The paper demonstrates a strong command of the subject and aligns well with established literature. For instance, it correctly highlights the two key features behind RB’s popularity sample efficient error estimation and immunity to state preparation and measurement (SPAM) errors and then clearly states the assumptions required (such as gate independent, time independent errors) that lead to RB’s characteristic exponential decay model . Throughout the tutorial, theoretical results (like the relationship between average sequence fidelity and average error rate) are derived carefully and cross-referenced with standard references (e.g., Refs. [3,12] for standard RB). The authors also connect these results to fundamental concepts like the depolarizing noise channel via twirling arguments.
•The document is well structured and easy to follow. The content is organized into logical chapters and sections, with a clear table of contents and section headers that guide the reader. Each chapter ends with a “Discussion” section that recaps and gives additional insight, which helps reinforce understanding and place the results in context. The writing is straightforward and flows well, with definitions introduced before they are used and symbols clearly explained. The authors also supply helpful figures and diagrams (e.g., circuit diagrams for RB sequences , and conceptual illustrations for simultaneous and correlated RB ) which support the text and provide intuition. Additionally, the inclusion of appendices (such as a “Poor man’s group theory” overview and details on twirling and depolarizing channels ) is a strong point – it ensures the tutorial is self-contained, giving readers the necessary background without assuming extensive prior knowledge.
•While the paper is primarily an overview of known techniques, it adds value by contextualizing RB in current research trends. In particular, Chapter 5 on the gateset shadow protocol introduces readers to a new unified framework that connects RB with classical shadow tomography . This is a novel inclusion that isn’t found in older RB reviews, and it provides insight into the evolving landscape of quantum benchmarking. By doing so, the tutorial not only teaches established methods but also prepares readers to understand and perhaps contribute to ongoing developments in the field. The authors’ extensive references to recent works (for example, they cite very recent methods like leakage RB, crossentropy benchmarking, non-Clifford RB, and shadow process tomography in the conclusions ) further demonstrate the thoroughness and relevance of the material. The paper essentially serves as a bridge between foundational knowledge and cutting-edge techniques.
•The tone of the tutorial is welcoming. The authors explicitly state that the notes are intended for both students and seasoned experimentalists, and they offer two modes of reading: a full indepth study or a quicker practical start . This considerate approach maximizes the paper’s reach. A reader who might be intimidated by the full theoretical treatment can follow the high-level descriptions and jump to the notebooks, whereas another reader who wants a deep understanding is equally catered to. Few papers manage to address such a broad audience range so effectively. Moreover, the code and data are made openly available in a public repository , underscoring the authors’ commitment to reproducibility and hands on learning. This will certainly amplify the paper’s impact as an educational tool.

Weaknesses

• While the paper covers multiple RB techniques, it does not provide a clear summary table comparing their differences, assumptions, and use cases. A concise reference would help readers quickly grasp the key distinctions.
• The paper focuses on theoretical derivations and simulations but lacks discussions or figures showing real-world experimental RB data. Including examples from actual quantum hardware would strengthen the tutorial’s applicability.
• Some sections, particularly on correlated RB and the gateset shadow protocol, are mathematically dense. Adding more intuitive explanations or real-world analogies would help readers unfamiliar with these advanced concepts.
• While the paper contains many derivations, crucial takeaways (e.g., final formulas for error rates or decay parameters) could be more explicitly emphasized, either through boxed summaries or clearer numbering.
• The discussion of this newer benchmarking framework is relatively brief. Expanding on its implications and possible future extensions would provide more value to readers interested in cutting edge developments.

Report

This is a comprehensive tutorial-style introduction to randomized benchmarking (RB) techniques. Its primary objective is to provide a pedagogical overview of the main principles behind RB for newcomers to quantum computing, bridging the gap between detailed theoretical literature and practical implementation . The authors focus on four fundamental RB protocols – standard RB, simultaneous RB, correlated RB, and interleaved RB – which are among the most widely used and relevant methods in the field . Each of these techniques is introduced in its own chapter, with explanations of the protocol’s purpose, underlying assumptions, and theoretical derivations. Notably, the tutorial also connects RB to the modern framework of classical shadow tomography by discussing the recently proposed gateset shadow estimation protocol . This inclusion places RB in a contemporary context, demonstrating how various RB variants can be viewed under a unified framework.

In addition to theoretical explanations, the paper has a strong hands on component. Every chapter is accompanied by a Python notebook that illustrates the essential steps of each RB protocol . This allows readers to actively engage with simulations and numerical examples, reinforcing their understanding of how RB experiments are performed and analyzed in practice. The authors explicitly tailor the tutorial to both students and experimentalists: readers can either delve into the full theoretical derivations or skip directly to practical aspects, depending on their needs . Overall, the paper’s contributions lie in educating the reader it consolidates foundational knowledge of RB, provides step by step derivations of key results, demonstrates implementation techniques, and highlights the relevance of RB in current quantum computing research. By the end of the tutorial, a reader should understand how to apply RB methods to quantify quantum gate errors and appreciate how these methods fit into the broader landscape of quantum characterization tools.

Requested changes

While the paper is already strong, there are a few suggestions that could further enhance its quality and usefulness:
• It would benefit the reader to include a concise comparative summary table that lists the four main RB variants (standard, simultaneous, correlated, interleaved) and highlights their key features. Such a table could include each protocol’s primary purpose (e.g., “measure average error rate of a gate set” for standard RB, “detect crosstalk errors” for simultaneous RB, etc.), the main assumptions (e.g., gate-independent errors, etc.), and the figure of merit it produces (average error rate r, crosstalk error metric, specific gate error rate, etc.). This would serve as a handy reference for readers to quickly recall differences and use-cases. It can be placed at the end of the tutorial or at the beginning of the conclusion section as a capstone. Given the rich content presented, a summary table would reinforce understanding by allowing a side-by-side comparison of the protocols.
• Some of the more advanced parts (particularly the correlated RB chapter and the gate-set shadow protocol chapter) introduce complex ideas that might be challenging for a novice reader. The authors might consider adding a few more sentences of intuitive explanation or a simple example to those sections. For instance, when introducing the correlators in correlated RB or the sequence correlation functions in the shadow protocol, a short intuitive description of what these quantities mean physically would complement the mathematical definition. The text as written is correct, but an extra bit of intuition (perhaps in the Discussion subsections) could help readers who are less mathematically inclined. Essentially, ensure that for every new parameter or function introduced, the reader has a mental picture of what it represents in terms of errors or circuits.
•The tutorial currently relies on simulated data (via the notebooks) to illustrate the protocols. It could be inspiring to include or discuss briefly an example of real experimental RB data from the literature. For example, when talking about standard RB, the authors could reference a specific experiment (perhaps from Refs. [7,8] which they cite as uses of RB in practice) and mention the typical values or outcomes (e.g., “a two-qubit device achieving an average error per gate of X%”). Similarly, for interleaved RB, referencing a real benchmarking of a particular gate (like a CNOT gate fidelity from a superconducting qubit experiment) would show how the theory translates to practice. Even without adding new data, a short description or figure showing an actual experimental decay curve and how it fits to extract r would connect the tutorial to hands-on lab work. This addition would underscore the “hands-on” aspect by demonstrating real-world relevance and could motivate readers by showing actual results achieved with these methods.
• As mentioned, there are a few minor typos (e.g., “randomzied”→“randomized” page 41) and formatting inconsistencies that should be corrected.
• The paper contains many derivations and equations. It might help readers if the authors highlight the most important formulas in some way (either by numbering them and referencing them in the text or by explicitly stating in words that “this equation is the central result of the protocol”). For example, the final expression relating the average sequence fidelity to the depolarizing parameter p in standard RB, or the equation giving the interleaved error rate in terms of two decay parameters, are crucial takeaways. Making sure these stand out – perhaps by referencing them in the Conclusions or Discussion as the “key results” – can aid a reader doing a quick review of the material. In a tutorial context, explicitly summarizing “what you should remember” is very useful. The authors do much of this in discussions, but a little more emphasis on formula labeling could help.

Recommendation

Publish (meets expectations and criteria for this Journal)

---

## Round 2 · Author Response

Dear Editor,
Thank you for the opportunity to revise our manuscript entitled “Hands-on Introduction to Randomized Benchmarking” to be considered for publication in SciPost Lecture Notes. We would also like to thank the reviewer for taking the time to carefully read our manuscript and for their detailed comments and valuable suggestions for improvement.
A point-by-point response to the reviewer comments has been made and the revised manuscript is provided, along with an extra PDF where every change in the text has been tracked in blue color. We have also changed the template to the corresponding SciPost Lecture Notes template and we have added all the DOIs for the references listed in our bibliography.
Thank you for your consideration.
Yours sincerely,
Ana Silva and Eliska Greplova
==== Response to Reviewer #1 ====
We greatly thank the reviewer for their careful reading of our manuscript and for their very helpful remarks. We are glad that the reviewer finds our tutorial relevant and pedagogical. We will address the remarks of the reviewer point by point below.
"It would benefit the reader to include a concise comparative summary table that lists the four main RB variants (standard, simultaneous, correlated, interleaved) and highlights their key features. Such a table could include each protocol’s primary purpose (e.g., “measure average error rate of a gate set” for standard RB, “detect crosstalk errors” for simultaneous RB, etc.), the main assumptions (e.g., gate-independent errors, etc.), and the figure of merit it produces (average error rate r, crosstalk error metric, specific gate error rate, etc.). This would serve as a handy reference for readers to quickly recall differences and use-cases. It can be placed at the end of the tutorial or at the beginning of the conclusion section as a capstone. Given the rich content presented, a summary table would reinforce understanding by allowing a side-by-side comparison of the protocols."
We thank the reviewer for this important suggestion. It is indeed helpful to include a summary table that allows a quick comparison between the different RB protocols. This not only reinforces the message of each chapter and improves the clarity of the manuscript, but also allows the reader to more quickly browse the document and refresh their memory on the topic.
We have added a comparison table at the end of the lecture notes (Chapter 6). This table provides a side-by-side comparison of the four RB protocols discussed in the lecture notes: standard RB, simultaneous RB, correlated RB and interleaved RB. The table allows the reader to quickly recall the main purpose of each protocol, the main assumptions under which each protocol operates, and the figure of merit that each protocol aims to estimate.
"Some of the more advanced parts (particularly the correlated RB chapter and the gate-set shadow protocol chapter) introduce complex ideas that might be challenging for a novice reader. The authors might consider adding a few more sentences of intuitive explanation or a simple example to those sections. For instance, when introducing the correlators in correlated RB or the sequence correlation functions in the shadow protocol, a short intuitive description of what these quantities mean physically would complement the mathematical definition. The text as written is correct, but an extra bit of intuition (perhaps in the Discussion subsections) could help readers who are less mathematically inclined. Essentially, ensure that for every new parameter or function introduced, the reader has a mental picture of what it represents in terms of errors or circuits."
We thank the reviewer for raising this issue. We agree that some of the more technical aspects might be more difficult for a first-time reader. We have tried to make all the steps involved in the mathematical derivations clear, so that the reader can follow them line by line if they wish. We also note that one of the purposes of the supplementary Python notebooks is to provide clear examples of the protocols, so that the reader has a good sense of how the mathematical formulae enter the protocol in practice.
However, we agree with the reviewer that our previous version of the manuscript lacked a proper pedagogical introduction to correlation functions. The correlation functions are key mathematical objects for both the correlated RB and the shadow protocol. We have now added more explanation onto the meaning of correlation functions, particularly in Section 3.1, where we give a general description of the correlated RB protocol. We also refer to the introduction of correlated functions in Section 3.1 later when we introduce the shadow protocol in Section 5.1, and provide more clarity on what the correlation functions are in the shadow protocol.
"The tutorial currently relies on simulated data (via the notebooks) to illustrate the protocols. It could be inspiring to include or discuss briefly an example of real experimental RB data from the literature. For example, when talking about standard RB, the authors could reference a specific experiment (perhaps from Refs. [7,8] which they cite as uses of RB in practice) and mention the typical values or outcomes (e.g., “a two-qubit device achieving an average error per gate of X%”). Similarly, for interleaved RB, referencing a real benchmarking of a particular gate (like a CNOT gate fidelity from a superconducting qubit experiment) would show how the theory translates to practice. Even without adding new data, a short description or figure showing an actual experimental decay curve and how it fits to extract r would connect the tutorial to hands-on lab work. This addition would underscore the “hands-on” aspect by demonstrating real-world relevance and could motivate readers by showing actual results achieved with these methods."
We thank the reviewer for this suggestion. We have now added a new section to the standard RB protocol chapter, providing an example, with real experimental data, of how to use the standard RB protocol to benchmark single-qubit operations in a real quantum device. We guide the reader through the main steps of the fitting procedure and provide actual experimental decay curves for the single-qubit fidelities. We also put the estimated values for the fidelities in the context of previously documented single-qubit fidelities for the same device. We also provide an accompanying Python notebook (see here: https://gitlab.com/QMAI/papers/rb-tutorial/-/blob/
main/StandardRB/Benchmarking_a_real_device.ipynb ), so that the readers can try out the fitting procedure for themselves on the same data.
"As mentioned, there are a few minor typos (e.g., “randomzied”→“randomized” page 41) and formatting inconsistencies that should be corrected."
We thank the reviewer for drawing our attention to the presence of typographical errors in the manuscript. We have made every effort to correct all typographical errors in the revised version of our manuscript.
"The paper contains many derivations and equations. It might help readers if the authors highlight the most important formulas in some way (either by numbering them and referencing them in the text or by explicitly stating in words that “this equation is the central result of the protocol”). For example, the final expression relating the average sequence fidelity to the depolarizing parameter p in standard RB, or the equation giving the interleaved error rate in terms of two decay parameters, are crucial takeaways. Making sure these stand out – perhaps by referencing them in the Conclusions or Discussion as the “key results” – can aid a reader doing a quick review of the material. In a tutorial context, explicitly summarizing “what you should remember” is very useful. The authors do much of this in discussions, but a little more emphasis on formula labeling could help."
We thank the reviewer for this suggestion. We have now added a summary table to each chapter of the manuscript, which lists the key findings for each protocol. In addition, we have made it clearer which formulae are actually the most important in each protocol by explicitly stating this in the text when these formulae appear for the first time.
Thank you for the opportunity to revise our manuscript entitled “Hands-on Introduction to Randomized Benchmarking” to be considered for publication in SciPost Lecture Notes. We would also like to thank the reviewer for taking the time to carefully read our manuscript and for their detailed comments and valuable suggestions for improvement.
A point-by-point response to the reviewer comments has been made and the revised manuscript is provided, along with an extra PDF where every change in the text has been tracked in blue color. We have also changed the template to the corresponding SciPost Lecture Notes template and we have added all the DOIs for the references listed in our bibliography.
Thank you for your consideration.
Yours sincerely,
Ana Silva and Eliska Greplova
==== Response to Reviewer #1 ====
We greatly thank the reviewer for their careful reading of our manuscript and for their very helpful remarks. We are glad that the reviewer finds our tutorial relevant and pedagogical. We will address the remarks of the reviewer point by point below.
"It would benefit the reader to include a concise comparative summary table that lists the four main RB variants (standard, simultaneous, correlated, interleaved) and highlights their key features. Such a table could include each protocol’s primary purpose (e.g., “measure average error rate of a gate set” for standard RB, “detect crosstalk errors” for simultaneous RB, etc.), the main assumptions (e.g., gate-independent errors, etc.), and the figure of merit it produces (average error rate r, crosstalk error metric, specific gate error rate, etc.). This would serve as a handy reference for readers to quickly recall differences and use-cases. It can be placed at the end of the tutorial or at the beginning of the conclusion section as a capstone. Given the rich content presented, a summary table would reinforce understanding by allowing a side-by-side comparison of the protocols."
We thank the reviewer for this important suggestion. It is indeed helpful to include a summary table that allows a quick comparison between the different RB protocols. This not only reinforces the message of each chapter and improves the clarity of the manuscript, but also allows the reader to more quickly browse the document and refresh their memory on the topic.
We have added a comparison table at the end of the lecture notes (Chapter 6). This table provides a side-by-side comparison of the four RB protocols discussed in the lecture notes: standard RB, simultaneous RB, correlated RB and interleaved RB. The table allows the reader to quickly recall the main purpose of each protocol, the main assumptions under which each protocol operates, and the figure of merit that each protocol aims to estimate.
"Some of the more advanced parts (particularly the correlated RB chapter and the gate-set shadow protocol chapter) introduce complex ideas that might be challenging for a novice reader. The authors might consider adding a few more sentences of intuitive explanation or a simple example to those sections. For instance, when introducing the correlators in correlated RB or the sequence correlation functions in the shadow protocol, a short intuitive description of what these quantities mean physically would complement the mathematical definition. The text as written is correct, but an extra bit of intuition (perhaps in the Discussion subsections) could help readers who are less mathematically inclined. Essentially, ensure that for every new parameter or function introduced, the reader has a mental picture of what it represents in terms of errors or circuits."
We thank the reviewer for raising this issue. We agree that some of the more technical aspects might be more difficult for a first-time reader. We have tried to make all the steps involved in the mathematical derivations clear, so that the reader can follow them line by line if they wish. We also note that one of the purposes of the supplementary Python notebooks is to provide clear examples of the protocols, so that the reader has a good sense of how the mathematical formulae enter the protocol in practice.
However, we agree with the reviewer that our previous version of the manuscript lacked a proper pedagogical introduction to correlation functions. The correlation functions are key mathematical objects for both the correlated RB and the shadow protocol. We have now added more explanation onto the meaning of correlation functions, particularly in Section 3.1, where we give a general description of the correlated RB protocol. We also refer to the introduction of correlated functions in Section 3.1 later when we introduce the shadow protocol in Section 5.1, and provide more clarity on what the correlation functions are in the shadow protocol.
"The tutorial currently relies on simulated data (via the notebooks) to illustrate the protocols. It could be inspiring to include or discuss briefly an example of real experimental RB data from the literature. For example, when talking about standard RB, the authors could reference a specific experiment (perhaps from Refs. [7,8] which they cite as uses of RB in practice) and mention the typical values or outcomes (e.g., “a two-qubit device achieving an average error per gate of X%”). Similarly, for interleaved RB, referencing a real benchmarking of a particular gate (like a CNOT gate fidelity from a superconducting qubit experiment) would show how the theory translates to practice. Even without adding new data, a short description or figure showing an actual experimental decay curve and how it fits to extract r would connect the tutorial to hands-on lab work. This addition would underscore the “hands-on” aspect by demonstrating real-world relevance and could motivate readers by showing actual results achieved with these methods."
We thank the reviewer for this suggestion. We have now added a new section to the standard RB protocol chapter, providing an example, with real experimental data, of how to use the standard RB protocol to benchmark single-qubit operations in a real quantum device. We guide the reader through the main steps of the fitting procedure and provide actual experimental decay curves for the single-qubit fidelities. We also put the estimated values for the fidelities in the context of previously documented single-qubit fidelities for the same device. We also provide an accompanying Python notebook (see here: https://gitlab.com/QMAI/papers/rb-tutorial/-/blob/
main/StandardRB/Benchmarking_a_real_device.ipynb ), so that the readers can try out the fitting procedure for themselves on the same data.
"As mentioned, there are a few minor typos (e.g., “randomzied”→“randomized” page 41) and formatting inconsistencies that should be corrected."
We thank the reviewer for drawing our attention to the presence of typographical errors in the manuscript. We have made every effort to correct all typographical errors in the revised version of our manuscript.
"The paper contains many derivations and equations. It might help readers if the authors highlight the most important formulas in some way (either by numbering them and referencing them in the text or by explicitly stating in words that “this equation is the central result of the protocol”). For example, the final expression relating the average sequence fidelity to the depolarizing parameter p in standard RB, or the equation giving the interleaved error rate in terms of two decay parameters, are crucial takeaways. Making sure these stand out – perhaps by referencing them in the Conclusions or Discussion as the “key results” – can aid a reader doing a quick review of the material. In a tutorial context, explicitly summarizing “what you should remember” is very useful. The authors do much of this in discussions, but a little more emphasis on formula labeling could help."
We thank the reviewer for this suggestion. We have now added a summary table to each chapter of the manuscript, which lists the key findings for each protocol. In addition, we have made it clearer which formulae are actually the most important in each protocol by explicitly stating this in the text when these formulae appear for the first time.

---

## Round 2 · List of Changes

To summarise the changes:
In section 1.3: lines 286-287.
Added new section 1.4.
In section 2.3: lines 582-583, lines 593-596, lines 661-662.
Added new section 2.5.
In section 3.1: lines 793-836, lines 890-892.
In section 3.2: lines 903-904.
Added new section 3.4.
In section 4.1: lines 1016-1017.
Added new section 4.3.
In section 5.1: lines 1096-1098, lines 1103-1119, lines 1160-1162.
Added new section 5.4.
Added new table in chapter 6.
In section 1.3: lines 286-287.
Added new section 1.4.
In section 2.3: lines 582-583, lines 593-596, lines 661-662.
Added new section 2.5.
In section 3.1: lines 793-836, lines 890-892.
In section 3.2: lines 903-904.
Added new section 3.4.
In section 4.1: lines 1016-1017.
Added new section 4.3.
In section 5.1: lines 1096-1098, lines 1103-1119, lines 1160-1162.
Added new section 5.4.
Added new table in chapter 6.

---

## Editorial Decision

published